# Active Tabular Augmentation via Policy-Guided Diffusion Inpainting

**Zheyu Zhang** [1,2]  **Shuo Yang** [1,2]  **Bardh Prenkaj** [3]  **Gjergji Kasneci** [1,2]

## Abstract

Generative tabular augmentation is appealing in data-scarce domains, yet the prevailing focus on distributional fidelity does not reliably translate into better downstream models. We formalize a *fidelity-utility gap*: common generative objectives prioritize distributional plausibility, whereas augmentation succeeds only when injected samples reduce the current learner's held-out evaluation loss. This gap motivates learning not just how to generate, but what to generate and when to inject as training evolves. We propose **TAP (Tabular Augmentation Policy)**, which couples diffusion inpainting with a lightweight, learner-conditioned policy to steer generation toward high-utility regions and controls safe injection via explicit gating and conservative windowed commitment. Under severe data scarcity, TAP consistently outperforms strong generative baselines on seven real-world datasets, improving classification accuracy by up to 15.6 percentage points and reducing regression RMSE by up to 32%.

## 1. Introduction

Tabular data drives decisions in healthcare, finance, science, and operations (Fatima et al., 2017; Dastile et al., 2020; Shwartz-Ziv & Armon, 2022; Baldi et al., 2014). In exactly these domains, labeled data is often scarce due to privacy constraints, annotation costs, and distribution shift across institutions (Levin et al., 2023; Bansal et al., 2022). While data augmentation is widely adopted as a remedy for limited data, its application to tabular data is particularly fragile. The heterogeneity of tabular features and the presence of strong inter-column dependencies that encode domain-specific constraints imply that even minor perturbations may invalidate samples or introduce spurious relationships (Cui et al., 2024; Borisov et al., 2022). A central challenge is therefore to

---

[1]Technical University of Munich [2]Munich Center for Machine Learning (MCML) [3]Sapienza University of Rome. Correspondence to: Zheyu Zhang <zheyu.zhang@tum.de>.

*Proceedings of the 43$^{rd}$ International Conference on Machine Learning*, Seoul, South Korea. PMLR 306, 2026. Copyright 2026 by the author(s).

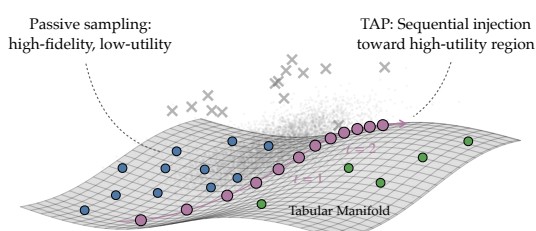

*Figure 1.* **Fidelity-utility gap in tabular augmentation.** Fidelity-oriented generators sample high-density regions of $P(X, Y)$, yielding plausible records that can be redundant and may offer limited downstream gain. Utility is state-dependent and tied to real-query loss. TAP learns what to generate and when to inject using conservative, feasibility-aware decisions under scarcity.

generate valid records that respect domain constraints while achieving high utility for downstream tasks.

Theoretically, valid records are not scattered uniformly across feature space. Instead, they concentrate on a structured manifold shaped by inter-column dependencies and domain constraints (Jiang et al., 2025; Mumuni & Mumuni, 2022). To match its distribution, modern generative models, including GANs, VAEs, flows, and diffusion models, often model either the joint distribution $P(X, Y)$ or a conditional variant, achieving strong statistical fidelity (Jiang et al., 2026). However, prior work shows that high-fidelity synthetic data do not necessarily improve downstream performance (Onishi & Meguro, 2023). Utility depends on task-specific relevance rather than solely on statistical similarity to the observed distribution. In contrast, classic methods such as SMOTE (Chawla et al., 2002) expand minority regions through simple neighbor interpolation, producing samples that are easily distinguishable from real data yet often effectively *improve* classifiers. These observations expose a fundamental tension that traditional generators are typically trained to maximize fidelity to the joint distribution $P(X, Y)$, whereas successful augmentation should make the learner approximate $P(Y|X)$ better. Therefore, high fidelity is not a sufficient condition for high utility, and under scarcity, it can be misaligned with the learner's needs.

**The fidelity-utility gap.** This disconnect reflects a misalignment between *how* generators are trained, and *how* augmentation is evaluated. Distribution-matching objectives encourage sampling from high-density regions of $P(X, Y)$, where the learner is already confident. Augmentation, how-

ever, is judged in downstream usage (Van Dyk & Meng, 2001; Onishi & Meguro, 2023). Under scarcity, the most impactful samples often lie where the model is uncertain, such as decision boundaries or under-covered subpopulations, although uncertainty alone is not sufficient (Bansal et al., 2022; Alzubaidi et al., 2023). Because these regions correspond to low-density areas under $P(X, Y)$, generators trained for distribution-matching tend to under-explore them even when the generator achieves high fidelity.

**Our View.** We view an effective generator as a controllable proposal mechanism for augmentation. Given a training set, it induces a family of feasible, anchor-conditioned proposal kernels over the tabular manifold. After that, we study how to control this proposal family to reduce the downstream error. Since the learner changes after each commitment, the utility of the same generation choice changes in training. This feedback makes augmentation an *active* process and motivates a state-conditioned policy that adapts generation conditions to the evolving learner.

**Our Method.** We instantiate this view in TAP[1] (short for Tabular Augmentation Policy), as illustrated in Figure 1. TAP uses diffusion inpainting to produce manifold-local proposals by fixing part of an anchor record and regenerating the remaining columns. A set of explicit quality gates enforces hard feasibility by filtering candidates that violate constraints. A lightweight policy then selects generation conditions based on a compact summary of the learner's state, thereby improving expected downstream utility. To improve robustness under noisy utility estimates, TAP employs windowed commitment, accumulating admitted candidates in a pool and committing the pool only when its estimated joint benefit exceeds a threshold.

**Contributions.** We turn a strong tabular diffusion generator into a controllable proposal mechanism for augmentation by learning both what to generate and when to inject. Concretely, we (1) formalize augmentation as a sequential control problem under an evolving learner, revealing the fidelity-utility gap and the necessity of state-dependent steering; (2) propose TAP, a state-conditioned policy that steers diffusion inpainting through target, template, and exploration controls, with safe injection enforced by hard gating and windowed commitment; (3) demonstrate consistent improvements across seven real-world datasets and five scarcity levels, with accuracy gains up to 15.6 percentage points and RMSE reductions up to 32% over strong generative baselines. Diagnostic analyses confirm that utility concentrates in informative yet learnable regions, validating our design principles.

---

[1]We make our code publicly available at https://github.com/oooranz/TAP.

## 2. Background & Motivation

The central question on the fidelity-utility gap is *what to inject* to maximize downstream benefit rather than *how to generate* more realistic samples.

### 2.1. Formalizing the Fidelity-Utility Gap

Let $P$ denote the distribution over feature-label pairs $(x, y)$. Augmentation is evaluated on a labeled query set $Q_{\text{real}} \sim P$, implemented as a validation split or cross-validation folds. An end-to-end augmentation pipeline induces an injection distribution $Q$ over the proposed and injected synthetic samples. This distribution is determined jointly by the generator and the injection rule, and it may evolve as the training set changes.

Let $f_\theta$ denote a predictor parameterized by $\theta$, and let $\theta(D)$ be the parameters obtained by training on $D$. Downstream performance is measured by loss on real queries,

$$L(\theta(D)) := \frac{1}{|Q_{\text{real}}|} \sum_{(x,y) \in Q_{\text{real}}} \ell(f_{\theta(D)}(x), y), \quad (1)$$

where $\ell$ is the task-appropriate loss. The value of injecting a candidate set $S$ is the marginal utility,

$$\Delta U(D, S) := L(\theta(D)) - L(\theta(D \cup S)). \quad (2)$$

Fidelity judges plausibility under $P$, but it does not determine which samples an end-to-end pipeline injects. Utility depends on whether injected samples complement $D$ and reduce loss on $Q_{\text{real}}$. Under scarcity, high-density samples are often redundant, whereas gains tend to come from informative yet learnable regions that are under-covered by $D$. This mismatch motivates learning *what to generate and when to inject*.

Directly optimizing Equation (2) is intractable because evaluating $\Delta U$ would retrain the learner for each candidate set. We therefore treat a fixed generator as a proposal family and learn a policy that steers admission and commitment toward positive marginal utility as the learner evolves.

**A first-order diagnostic of utility.** To motivate what information a control policy must track, we view injection through a smooth surrogate learner trained by regularized empirical risk minimization. Influence functions (Cook et al., 1982; Koh & Liang, 2017) describe the first-order effect of adding a training example. Combining this with a Taylor expansion of the real-query loss around $\theta(D)$ yields the diagnostic approximation

$$\Delta U(D, \{z\}) \approx \frac{1}{|D|} \nabla_\theta L(\theta(D))^\top \\ \times H_D^{-1} \nabla_\theta \ell(f_{\theta(D)}(x), y), \quad (3)$$

where $H_D$ is the Hessian of the training objective at $\theta(D)$. A derivation is provided in Appendix B.5. We use Equation (3) only as a diagnostic and do not compute it in the algorithm. Its role is to connect utility to the learner's current errors while highlighting the need to enforce feasibility and avoid redundant injection, which directly motivates our plug-in evaluator, gating, and diversity-aware policy.

## 2.2. Design Principles

The formalization above suggests that effective augmentation requires explicit optimization of downstream utility, while maintaining feasibility and robustness to noisy estimates. We identify three guiding principles.

**Principle 1: Two-stage feasibility.** Tabular records must satisfy soft and hard constraints. Soft feasibility requires candidates to lie near the data manifold, respecting inter-column dependencies learned from data. Hard feasibility requires compliance with domain rules such as valid categorical values, value ranges, and logical consistency. A proposal mechanism should encourage soft feasibility by design, while hard feasibility requires explicit enforcement.

**Principle 2: Utility-driven selection.** Not all feasible samples are equally valuable (Bengio et al., 2009). Equation (3) suggests that utility depends on how injected samples interact with the learner's current errors. Under scarcity, uncertainty can indicate where gains are possible, but it is not sufficient. Boundary-adjacent samples can be inherently ambiguous and may degrade performance when injected. Selection should therefore target samples that are informative yet learnable, adapting to the learner as it evolves.

**Principle 3: Conservative sequential injection.** Marginal utility depends on the current learner and is noisy to estimate, especially under scarcity where a few harmful samples can have outsized impact. We therefore inject conservatively by accumulating admitted samples in a window and updating the committed buffer only when the pooled gain is large enough:

$$B_{t+K} = \begin{cases} B_t \cup P_t^{(K)}, & \Delta U(D_t, P_t^{(K)}) > \tau, \\ B_t, & \text{otherwise.} \end{cases} \quad (4)$$

Here $P_t^{(K)}$ is the pool collected over a window of length $K$, and $\tau$ is a minimum required gain that controls the conservativeness of commitment. In practice, we commit based on a plug-in estimate of the pooled gain with an explicit safety margin.

Diffusion inpainting supports soft feasibility, explicit gating enforces hard feasibility, policy-guided selection targets high-utility regions, and windowed commitment prevents harmful injections. Section 3 instantiates these principles as a controlled proposal and admission process.

## 3. Methodology

The fidelity-utility gap reveals that effective augmentation requires injecting samples that help the learner, not merely samples that resemble real data. In this section, we formalize this view and introduce `TAP`.

### 3.1. Sequential Augmentation as a Controlled Process

Recall the real-query loss $L(\theta(D))$ and marginal utility $\Delta U(D, S)$ defined in Equations (1) and (2), respectively. Our objective is to reduce $L(\theta(D))$ through safe injection rather than to match the data distribution.

**Committed dataset and decision horizon.** Augmentation proceeds for $T$ decision steps. We maintain a committed buffer $B$ and a temporary pool $P$, and denote their values at step $t$ as $B_t$ and $P_t$. The training set at step $t$ is

$$D_t := D_0 \cup B_t. \quad (5)$$

At each step, the policy selects an action that induces a batch proposal distribution. Pointwise feasibility gates reject invalid candidates, and the remaining samples are added to $P_t$. The buffer $B_t$ changes only at commitment times, which is the only mechanism by which augmentation affects the learner. Since $D_t$ changes only at commitment times, we cache $\widehat{L}_\psi(D_t)$ within each window and evaluate $\widehat{L}_\psi(D_t \cup S)$ by forward passes for candidate sets, where $\psi$ is the plug-in evaluator introduced in Section 3.3. Full pseudocode is provided in Appendix C.1.

**Trajectory objective.** Let $\pi$ be a policy that maps the learner state $s_t$ to a distribution over actions. We define the trajectory-level objective as the expected final utility

$$J(\pi) := \mathbb{E}_\pi [L(\theta(D_0)) - L(\theta(D_T))]. \quad (6)$$

**Utility telescopes over commitments.** We evaluate pooled utility every $K$ steps and update the training set when the pool passes the commitment rule. Let $0 = t_0 < t_1 < \cdots < t_M = T$ be commitment times, with $M \leq \lceil T/K \rceil$, and let $P_i$ denote the pool committed at time $t_{i+1}$. Because the committed buffer changes only at these times,

$$L(\theta(D_0)) - L(\theta(D_T)) = \sum_{i=0}^{M-1} \Delta U(D_{t_i}, P_i). \quad (7)$$

A short proof is included in Appendix B. When training is stochastic, the same identity holds in expectation over learner randomness.

### 3.2. Manifold-Constrained Proposals via Diffusion Inpainting

Principle 1 requires a proposal mechanism that respects manifold structure while allowing controlled exploration.

We use diffusion inpainting as a proposal operator, adapting techniques originally developed for image completion (Lugmayr et al., 2022) to the tabular setting. It produces locally coherent candidates by conditioning on a real anchor and regenerating a subset of columns.

Let $q_\phi$ be a diffusion model trained on the real training split and frozen during policy learning. We train it on labeled tables and treat the label as fixed during inpainting, so we never sample labels separately. Given an anchor record from $D_t$, a binary mask $m \in \{0,1\}^d$ over feature columns, and a target condition $c$, diffusion inpainting samples

$$x^{\mathrm{syn}} \sim q_\phi(x_m \mid x_{\bar{m}}, c), \tag{8}$$

where $x_{\bar{m}}$ denotes the fixed columns. This can be viewed as sampling a conditional marginal of the learned joint table distribution, with the label held fixed by the condition.

We implement inpainting by overwriting fixed columns at each reverse diffusion step. Let $x^{(s)}$ denote the sample at reverse step $s$. After proposing $x^{(s-1)}$, we replace the fixed coordinates with the corresponding forward noised anchor values:

$$x_{\bar{m}}^{(s-1)} \leftarrow \sqrt{\bar{\alpha}_{s-1}}\, x_{\bar{m}} + \sqrt{1 - \bar{\alpha}_{s-1}}\, \epsilon, \qquad \epsilon \sim \mathcal{N}(0, I), \tag{9}$$

where $\bar{\alpha}_s$ is the cumulative noise schedule. This yields stable conditional generation without retraining the backbone.

**Action space.** We parameterize generation by three complementary controls,

$$a := (c, \eta, \rho), \tag{10}$$

where $c$ selects a target condition, $\eta$ selects a mask template that controls locality, and $\rho \in [0,1]$ controls exploration strength. Concretely, $c$ indexes a class label for classification or a target quantile bin for regression, $\eta$ selects between an explore template and a conservative template, and $\rho$ adjusts how many feature columns are regenerated within the chosen template. Larger $\rho$ yields more diverse proposals, while smaller $\rho$ produces near-anchor samples. Exact templates and the sampling rule are given in Appendix C.2.

**A controlled proposal kernel.** An action $a = (c, \eta, \rho)$ induces a proposal distribution through anchor selection, mask construction, and diffusion randomness. We write the induced proposal family as

$$Q_a(\cdot \mid D_t) = \mathbb{E}_{x \sim p_{\mathrm{anc}}(\cdot \mid D_t, c)} \mathbb{E}_{m \sim p_{\mathrm{mask}}(\cdot \mid \eta, \rho)} \\ \times [q_\phi(\cdot \mid x, m, c)], \tag{11}$$

where $p_{\mathrm{anc}}$ selects anchors from $D_t$ and $p_{\mathrm{mask}}$ samples regeneration patterns from the template indexed by $\eta$.

We draw a candidate batch by sampling independently from $Q_{a_t}(\cdot \mid D_t)$, and we denote this batch level sampling by

$$\widetilde{S}_t \sim \mathcal{K}_\phi(\cdot \mid D_t, a_t). \tag{12}$$

Actions change $Q_a$ and thus the explored regions of the data manifold. Only committed pools update $B$ and therefore modify the learner through Equation (6).

### 3.3. Utility-Aligned Selection by Policy Optimization

Principle 2 requires selection that adapts to the learner and targets high-utility regions. Directly optimizing $\Delta U(D_t, S_t)$ is intractable because it requires training the downstream learner for each candidate set. We instead use a plug-in evaluator that supports fast conditioning and repeated forward passes on a focused query set.

**Focused plug-in loss and plug-in utility.** Let $Q_{\mathrm{hard}}(D_t) \subseteq Q_{\mathrm{real}}$ be a subset of informative queries selected using the current evaluator. For classification, we select high-entropy queries. For regression, we select high-uncertainty queries. We define the focused plug-in loss

$$\widehat{L}_\psi(D_t) := \frac{1}{|Q_{\mathrm{hard}}(D_t)|} \sum_{(x,y) \in Q_{\mathrm{hard}}(D_t)} \ell(f_{\psi(D_t)}(x), y). \tag{13}$$

Here $\psi$ is an online evaluation procedure and $\psi(D_t)$ denotes the evaluator conditioned on the current training set $D_t$. We use TabPFN (Hollmann et al., 2025) as the default evaluator because it supports fast in-context conditioning and repeated forward passes. $\widehat{L}_\psi$ is used only as a ranking signal for candidate pools during policy learning, and the reported gains are measured by retraining standard downstream predictors on the committed augmented set. Under cross-validation, each query fold is evaluated using a context that excludes that fold. We use this loss to define a plug-in estimate of the marginal utility of injecting a candidate set $S$,

$$\widehat{\Delta U}_\psi(D_t, S) := \widehat{L}_\psi(D_t) - \widehat{L}_\psi(D_t \cup S). \tag{14}$$

Under a uniform accuracy condition on $\widehat{L}_\psi$, the induced error on $\widehat{\Delta U}_\psi$ is bounded. Details are in Appendix B.

**Admission and expected gain.** At step $t$, the policy samples $a_t \sim \pi(\cdot \mid s_t)$ and proposals are generated by $\widetilde{S}_t \sim \mathcal{K}_\phi(\cdot \mid D_t, a_t)$. An admission rule yields the admitted set $S_t$. Let $G = 1$ denote the event that a proposal passes the gates. By the law of total expectation, the expected plug-in gain factors into feasibility and conditional utility,

$$\mathbb{E}\left[\widehat{\Delta U}_\psi(D_t, S_t) \mid s_t, a_t\right] = \Pr(G = 1 \mid s_t, a_t) \\ \cdot \mathbb{E}\left[\widehat{\Delta U}_\psi(D_t, S_t) \mid s_t, a_t, G = 1\right]. \tag{15}$$

**State design.** We encode the learner state as

$$s_t := (\delta_t, u_t, g_t, d_t). \tag{16}$$

Here $\delta_t$ tracks under-covered targets, $u_t$ summarizes predictive uncertainty on focused queries, $g_t$ estimates recent gate pass statistics, and $d_t$ summarizes redundancy relative to the committed buffer and the current pool to mitigate diminishing returns. These components are motivated by the diagnostic in Equation (3) and are designed to help rank actions under Equation (15). Appendix B provides further rationale and Appendix F validates the design via state ablations.

**Preference-based regularized improvement.** Utility estimates are noisy and heavy-tailed, so we use a KL-regularized policy update against a conservative reference $\pi_{\text{ref}}$. Let $\widehat{A}_t$ be a baseline-corrected advantage derived from $\widehat{\Delta U}_\psi$. We optimize

$$\max_\pi \ \mathbb{E}\left[\widehat{A}_t\right] - \beta \operatorname{KL}(\pi(\cdot \mid s_t) \| \pi_{\text{ref}}(\cdot \mid s_t)). \tag{17}$$

In our implementation, we optimize this objective with a KL-regularized preference-style update using pairwise comparisons derived from $\widehat{\Delta U}_\psi$. Implementation details and default hyperparameters are provided in Appendix C.2.

### 3.4. Safe Admission and Conservative Commitment

Principle 3 requires robustness to noisy utility estimation and protection against harmful injection under scarcity. We enforce this through pointwise feasibility gates and windowed commitment.

**Pointwise gating.** We use an acceptance function $G(x; D_t) \in \{0, 1\}$ that enforces hard feasibility constraints, such as valid categorical values and range checks. Candidates that violate any hard constraint are rejected. The admitted set is

$$S_t := \{x \in \widetilde{S}_t \mid G(x; D_t) = 1\}. \tag{18}$$

Implementation details of the gates are provided in Appendix C.2.

**Windowed commitment.** We accumulate admitted samples into a pool over a window of length $K$. Let $P_t^{(K)}$ denote the pool formed within the current window, with a formal definition in Appendix C.1. At commitment times, we evaluate pooled plug-in utility

$$\widehat{\Delta U}_{K,\psi}(D_t, P_t^{(K)}) := \widehat{L}_\psi(D_t) - \widehat{L}_\psi(D_t \cup P_t^{(K)}), \tag{19}$$

and commit only when $\widehat{\Delta U}_{K,\psi}(D_t, P_t^{(K)}) > \tau + \epsilon_t$. Within a window, $D_t$ remains fixed and $\widehat{L}_\psi(D_t)$ is computed once and reused. After each commitment check, we discard the pool and start a new window.

**Theorem 3.1** (Commitment safety with calibrated plug-in uncertainty). *Fix a commitment time $t$ with pool $P_t^{(K)}$. Assume we can compute an error bar $\epsilon_t \geq 0$ such that*

$$\Pr\left(\left|\widehat{\Delta U}_{K,\psi}(D_t, P_t^{(K)}) - \Delta U(D_t, P_t^{(K)})\right| \leq \epsilon_t\right) \geq 1 - \alpha. \tag{20}$$

*If* TAP *commits only when $\widehat{\Delta U}_{K,\psi}(D_t, P_t^{(K)}) > \tau + \epsilon_t$, then the committed pool satisfies $\Delta U(D_t, P_t^{(K)}) \geq \tau$ with probability at least $1 - \alpha$.*

Theorem 3.1 turns commitment into a certified decision rule once plug-in uncertainty is calibrated. We estimate $\epsilon_t$ using the focused query set and report calibration results in Appendix E.3.

## 4. Experiments

We organize our experiments around three questions: (1) whether TAP yields reliable downstream gains under scarcity, where augmentation is often fragile; (2) where high-utility injected samples lie and which injection mechanism best identifies them; and (3) whether gating and conservative windowed commitment reduce harmful injection.

### 4.1. Setup

**Datasets and scarcity simulation.** Dataset statistics are provided in Appendix D.1. Following Margeloiu et al. (2024), we subsample training data to simulate scarcity, i.e., $n_{\text{real}} \in \{20, 50, 100, 200, 500\}$, with a 4:1 train-validation split. All experiments are repeated over 5 random splits. For TAP, plug-in utility is estimated using $M$-fold cross-validation splits constructed from the real training split. The evaluator is conditioned on $D_t = D_0 \cup B_t$ and never accesses validation or test labels.

**Baselines.** We compare against seven augmentation methods: the interpolation method SMOTE (Chawla et al., 2002), the VAE-based TVAE (Xu et al., 2019), the GAN-based CTGAN (Xu et al., 2019), the tree-based ARF (Watson et al., 2023), the flow-based SPADA (Yang et al., 2025), and diffusion-based TabDDPM (Kotelnikov et al., 2023) and TabDiff (Shi et al., 2025). We also report "Real", which trains on real data only.

**Downstream predictors.** We evaluate on six downstream predictors: Logistic Regression (Cox, 1958), KNN (Fix, 1985), MLP (Gorishniy et al., 2021), Random Forest (Breiman, 2001), LightGBM (Ke et al., 2017), and XGBoost (Chen, 2016). During injection, TAP estimates plug-in utility with TabPFN (Hollmann et al., 2025) as a default online evaluator, which supports frequent evaluations without retraining. We report results only on the downstream predictors above so that the final evaluation is independent

*Table 1.* **Overall downstream utility under data scarcity.** Classification accuracy (%) is averaged over six classifiers (LR, KNN, MLP, RF, LightGBM, XGBoost), and regression RMSE is averaged over four regressors (KNN, RF, LightGBM, XGBoost). The anomalous entries of TabDDPM on Ailerons are explained in Appendix D.3.

| Dataset | $N_{real}$ | Real | SMOTE | TVAE | CTGAN | ARF | SPADA | TabDDPM | TabDiff | TAP |
|---|---|---|---|---|---|---|---|---|---|---|
| | | | | | Classification (Accuracy ↑) | | | | | |
| MiceProtein | 20 | $36.21_{\pm3.96}$ | $41.34_{\pm4.22}$ | $36.93_{\pm4.29}$ | $32.35_{\pm3.24}$ | $36.91_{\pm4.92}$ | $37.59_{\pm4.83}$ | $34.05_{\pm4.44}$ | $37.85_{\pm2.78}$ | $\mathbf{44.60_{\pm5.04}}$ |
| | 50 | $59.39_{\pm2.98}$ | $59.40_{\pm3.31}$ | $50.64_{\pm3.58}$ | $49.51_{\pm4.13}$ | $55.53_{\pm3.73}$ | $55.78_{\pm2.99}$ | $54.05_{\pm4.48}$ | $54.15_{\pm2.77}$ | $\mathbf{61.78_{\pm3.08}}$ |
| | 100 | $71.96_{\pm2.30}$ | $71.27_{\pm2.04}$ | $63.59_{\pm2.88}$ | $65.13_{\pm2.63}$ | $65.01_{\pm1.72}$ | $68.86_{\pm1.91}$ | $66.95_{\pm3.00}$ | $67.21_{\pm2.99}$ | $\mathbf{73.06_{\pm3.83}}$ |
| | 200 | $86.00_{\pm1.99}$ | $85.98_{\pm1.83}$ | $78.48_{\pm2.08}$ | $80.93_{\pm1.83}$ | $81.09_{\pm2.27}$ | $84.03_{\pm2.26}$ | $81.10_{\pm2.15}$ | $80.97_{\pm2.48}$ | $\mathbf{86.51_{\pm1.77}}$ |
| | 500 | $96.44_{\pm0.86}$ | $\mathbf{96.65_{\pm0.88}}$ | $93.75_{\pm1.23}$ | $93.71_{\pm1.20}$ | $94.56_{\pm1.15}$ | $96.13_{\pm0.88}$ | $93.81_{\pm1.34}$ | $94.99_{\pm1.49}$ | $96.11_{\pm0.99}$ |
| Credit-G | 20 | $66.37_{\pm4.73}$ | $59.06_{\pm9.70}$ | $65.79_{\pm2.89}$ | $65.48_{\pm4.02}$ | $64.25_{\pm3.48}$ | $57.58_{\pm5.70}$ | $63.99_{\pm2.40}$ | $62.84_{\pm3.16}$ | $\mathbf{68.13_{\pm2.75}}$ |
| | 50 | $65.29_{\pm3.52}$ | $67.23_{\pm1.92}$ | $68.21_{\pm1.46}$ | $60.43_{\pm5.29}$ | $66.49_{\pm2.07}$ | $65.53_{\pm4.12}$ | $62.79_{\pm3.26}$ | $67.55_{\pm1.89}$ | $\mathbf{69.72_{\pm0.51}}$ |
| | 100 | $67.53_{\pm2.43}$ | $68.27_{\pm1.23}$ | $68.65_{\pm1.63}$ | $67.26_{\pm1.68}$ | $67.27_{\pm1.50}$ | $66.09_{\pm3.05}$ | $64.07_{\pm2.19}$ | $68.15_{\pm2.11}$ | $\mathbf{70.73_{\pm1.66}}$ |
| | 200 | $67.85_{\pm3.81}$ | $69.33_{\pm1.38}$ | $69.30_{\pm1.52}$ | $64.41_{\pm2.34}$ | $67.22_{\pm4.42}$ | $66.07_{\pm2.90}$ | $62.77_{\pm5.06}$ | $70.13_{\pm1.73}$ | $\mathbf{71.35_{\pm1.67}}$ |
| | 500 | $71.17_{\pm0.64}$ | $71.07_{\pm0.74}$ | $71.50_{\pm0.89}$ | $68.06_{\pm2.18}$ | $69.86_{\pm1.15}$ | $69.53_{\pm1.55}$ | $66.25_{\pm3.14}$ | $72.31_{\pm1.62}$ | $\mathbf{74.21_{\pm0.77}}$ |
| Electricity | 20 | $66.09_{\pm5.58}$ | $61.99_{\pm5.89}$ | $64.74_{\pm4.89}$ | $59.70_{\pm4.95}$ | $67.81_{\pm6.74}$ | $66.75_{\pm8.45}$ | $62.23_{\pm4.98}$ | $60.17_{\pm7.94}$ | $\mathbf{69.28_{\pm9.21}}$ |
| | 50 | $69.05_{\pm4.10}$ | $64.71_{\pm4.11}$ | $69.09_{\pm4.95}$ | $63.64_{\pm3.07}$ | $70.81_{\pm5.21}$ | $69.61_{\pm4.60}$ | $66.11_{\pm4.36}$ | $68.05_{\pm4.68}$ | $\mathbf{71.55_{\pm4.50}}$ |
| | 100 | $72.73_{\pm3.81}$ | $68.21_{\pm3.71}$ | $72.15_{\pm4.62}$ | $67.21_{\pm3.03}$ | $74.02_{\pm3.40}$ | $72.83_{\pm3.77}$ | $70.97_{\pm2.95}$ | $69.49_{\pm2.94}$ | $\mathbf{74.73_{\pm3.24}}$ |
| | 200 | $74.95_{\pm3.09}$ | $70.61_{\pm3.02}$ | $74.50_{\pm3.75}$ | $72.16_{\pm3.41}$ | $75.19_{\pm3.15}$ | $74.63_{\pm3.12}$ | $72.07_{\pm2.60}$ | $72.69_{\pm3.60}$ | $\mathbf{75.87_{\pm2.51}}$ |
| | 500 | $76.37_{\pm2.17}$ | $73.25_{\pm2.11}$ | $76.45_{\pm2.27}$ | $75.33_{\pm2.49}$ | $76.41_{\pm2.44}$ | $76.37_{\pm2.31}$ | $74.51_{\pm2.07}$ | $76.06_{\pm1.95}$ | $\mathbf{77.77_{\pm2.04}}$ |
| Fourier | 20 | $38.69_{\pm4.10}$ | $40.09_{\pm4.90}$ | $40.63_{\pm5.62}$ | $28.77_{\pm4.14}$ | $38.87_{\pm5.07}$ | $38.69_{\pm4.10}$ | $30.95_{\pm5.12}$ | $31.63_{\pm6.42}$ | $\mathbf{41.67_{\pm7.43}}$ |
| | 50 | $59.03_{\pm2.19}$ | $60.71_{\pm1.87}$ | $52.31_{\pm2.56}$ | $43.67_{\pm3.25}$ | $60.75_{\pm2.62}$ | $59.03_{\pm2.19}$ | $49.57_{\pm3.15}$ | $46.23_{\pm3.12}$ | $\mathbf{62.91_{\pm1.96}}$ |
| | 100 | $67.23_{\pm1.83}$ | $68.35_{\pm1.77}$ | $61.43_{\pm1.75}$ | $55.08_{\pm1.70}$ | $68.44_{\pm1.41}$ | $67.23_{\pm1.83}$ | $61.54_{\pm2.75}$ | $58.25_{\pm3.18}$ | $\mathbf{68.89_{\pm2.55}}$ |
| | 200 | $72.99_{\pm1.31}$ | $73.97_{\pm1.21}$ | $69.13_{\pm2.35}$ | $67.78_{\pm2.63}$ | $73.48_{\pm1.57}$ | $72.99_{\pm1.31}$ | $69.54_{\pm1.23}$ | $67.61_{\pm2.51}$ | $\mathbf{75.01_{\pm1.39}}$ |
| | 500 | $77.58_{\pm1.51}$ | $\mathbf{78.02_{\pm1.52}}$ | $75.09_{\pm1.67}$ | $75.19_{\pm1.59}$ | $77.17_{\pm1.91}$ | $77.58_{\pm1.51}$ | $76.21_{\pm1.29}$ | $75.09_{\pm1.73}$ | $77.82_{\pm1.58}$ |
| Steel | 20 | $70.81_{\pm2.66}$ | $72.30_{\pm4.11}$ | $67.11_{\pm2.94}$ | $65.83_{\pm2.91}$ | $68.37_{\pm2.75}$ | $70.81_{\pm2.66}$ | $69.59_{\pm3.54}$ | $75.26_{\pm3.41}$ | $\mathbf{77.19_{\pm4.55}}$ |
| | 50 | $78.63_{\pm2.71}$ | $84.42_{\pm3.30}$ | $73.24_{\pm2.51}$ | $72.92_{\pm3.27}$ | $72.48_{\pm3.22}$ | $78.63_{\pm2.71}$ | $81.17_{\pm5.77}$ | $87.81_{\pm3.31}$ | $\mathbf{94.27_{\pm2.39}}$ |
| | 100 | $87.75_{\pm2.34}$ | $95.73_{\pm1.21}$ | $77.21_{\pm2.03}$ | $78.69_{\pm2.16}$ | $83.71_{\pm2.44}$ | $87.75_{\pm2.34}$ | $90.13_{\pm3.21}$ | $95.36_{\pm3.03}$ | $\mathbf{98.47_{\pm0.72}}$ |
| | 200 | $94.92_{\pm1.31}$ | $98.29_{\pm0.45}$ | $83.97_{\pm1.41}$ | $88.23_{\pm1.96}$ | $94.91_{\pm1.81}$ | $94.92_{\pm1.31}$ | $95.51_{\pm1.24}$ | $98.43_{\pm0.63}$ | $\mathbf{98.55_{\pm0.54}}$ |
| | 500 | $98.67_{\pm0.47}$ | $99.25_{\pm0.13}$ | $93.44_{\pm1.24}$ | $95.73_{\pm1.84}$ | $98.92_{\pm0.35}$ | $98.67_{\pm0.47}$ | $98.02_{\pm1.24}$ | $99.21_{\pm0.24}$ | $\mathbf{99.27_{\pm0.36}}$ |
| | | | | | Regression (RMSE ↓) | | | | | |
| Ailerons | 20 | $1.042_{\pm0.19}$ | $1.077_{\pm0.22}$ | $1.046_{\pm0.23}$ | $1.107_{\pm0.21}$ | $0.926_{\pm0.19}$ | $1.065_{\pm0.15}$ | $1.035_{\pm0.21}$ | $1.015_{\pm0.18}$ | $\mathbf{0.919_{\pm0.19}}$ |
| | 50 | $0.790_{\pm0.15}$ | $0.833_{\pm0.15}$ | $0.898_{\pm0.19}$ | $0.914_{\pm0.16}$ | $0.762_{\pm0.16}$ | $0.833_{\pm0.12}$ | $0.808_{\pm0.16}$ | $0.920_{\pm0.19}$ | $\mathbf{0.737_{\pm0.15}}$ |
| | 100 | $0.596_{\pm0.08}$ | $0.646_{\pm0.08}$ | $0.683_{\pm0.09}$ | $0.756_{\pm0.06}$ | $0.587_{\pm0.09}$ | $0.641_{\pm0.09}$ | $108.419_{\pm215.59}$ | $0.673_{\pm0.08}$ | $\mathbf{0.570_{\pm0.08}}$ |
| | 200 | $0.569_{\pm0.06}$ | $0.590_{\pm0.04}$ | $0.620_{\pm0.06}$ | $0.667_{\pm0.07}$ | $0.557_{\pm0.06}$ | $0.580_{\pm0.06}$ | $83.927_{\pm108.43}$ | $0.604_{\pm0.08}$ | $\mathbf{0.551_{\pm0.06}}$ |
| | 500 | $0.501_{\pm0.04}$ | $0.522_{\pm0.04}$ | $0.543_{\pm0.04}$ | $0.587_{\pm0.08}$ | $0.499_{\pm0.04}$ | $0.506_{\pm0.04}$ | $262.262_{\pm225.86}$ | $0.516_{\pm0.05}$ | $\mathbf{0.493_{\pm0.04}}$ |
| Insurance | 20 | $0.971_{\pm0.34}$ | $0.952_{\pm0.24}$ | $1.017_{\pm0.24}$ | $1.267_{\pm0.24}$ | $1.145_{\pm0.23}$ | $1.435_{\pm0.19}$ | $0.979_{\pm0.34}$ | $1.067_{\pm0.20}$ | $\mathbf{0.885_{\pm0.41}}$ |
| | 50 | $0.937_{\pm0.31}$ | $0.834_{\pm0.28}$ | $0.900_{\pm0.30}$ | $1.248_{\pm0.21}$ | $1.039_{\pm0.21}$ | $1.458_{\pm0.17}$ | $0.944_{\pm0.35}$ | $0.841_{\pm0.27}$ | $\mathbf{0.632_{\pm0.22}}$ |
| | 100 | $0.616_{\pm0.09}$ | $0.609_{\pm0.09}$ | $0.702_{\pm0.10}$ | $1.319_{\pm0.17}$ | $0.758_{\pm0.12}$ | $1.265_{\pm0.16}$ | $0.628_{\pm0.09}$ | $0.607_{\pm0.07}$ | $\mathbf{0.502_{\pm0.10}}$ |
| | 200 | $0.593_{\pm0.04}$ | $0.611_{\pm0.04}$ | $0.654_{\pm0.04}$ | $1.166_{\pm0.12}$ | $0.674_{\pm0.05}$ | $1.101_{\pm0.14}$ | $0.661_{\pm0.13}$ | $0.567_{\pm0.06}$ | $\mathbf{0.459_{\pm0.03}}$ |
| | 500 | $0.593_{\pm0.01}$ | $0.611_{\pm0.03}$ | $0.620_{\pm0.06}$ | $1.099_{\pm0.07}$ | $0.685_{\pm0.05}$ | $0.714_{\pm0.15}$ | $0.709_{\pm0.12}$ | $0.472_{\pm0.10}$ | $\mathbf{0.468_{\pm0.01}}$ |

of the online evaluator. Appendix F reports results with alternative online evaluators.

**Protocol.** For each split, each augmentation method is fit on the real training split. For fair comparison, all methods inject the same budget of $n_{syn} = 500$ synthetic samples into the real training set. We use TabDiff as the diffusion backbone for TAP, trained once and fixed throughout. Downstream predictors are then trained on the augmented set, with the real validation set used for early stopping and the real test set for final evaluation. Appendix E.4 reports computational cost.

### 4.2. Utility Gains Across Scarcity Levels

We first evaluate whether TAP delivers reliable downstream gains under scarcity, where tabular augmentation is most valuable and most fragile.

Table 1 reports classification accuracy and regression RMSE averaged over six downstream predictors (see Appendix G.2 for per-predictor results). Across scarcity levels, TAP achieves the best or near-best average performance, with the largest gains at $n_{real}=20$. This is the regime where each injected record has an outsized influence, so inconsistent samples can easily degrade performance.

The results are also consistent with the fidelity–utility gap in Section 2.1. Several generators underperform Real on a non-trivial fraction of datasets and scarcity levels, which indicates that synthetic samples can harm rather than help. No single baseline dominates across settings, and the relative ranking varies with dataset and scarcity. In contrast, TAP yields consistently positive improvements, which aligns with optimizing a surrogate of downstream utility rather than relying on distributional fidelity alone.

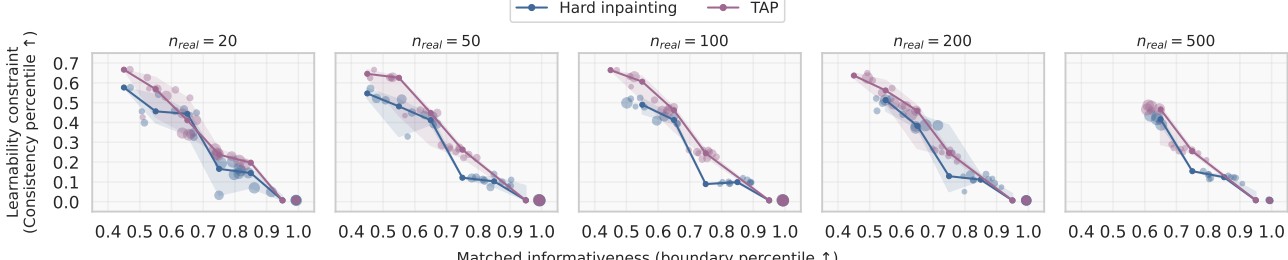

*Figure 2.* **Learnability under matched informativeness.** Runs are bucketed by decision-boundary percentile and the vertical axis reports learnability percentile. `TAP` achieves better learnability at comparable levels of informativeness.

### 4.3. Where High-Utility Samples Lie

The results above establish that `TAP` outperforms baselines, but they do not reveal what makes injected samples useful. Principle 2 suggests that high-utility samples are informative yet learnable. We test this hypothesis through controlled mechanism comparisons and post-hoc diagnostics.

**The injection mechanism matters.** To disentangle *where/when to inject* from backbone quality, we fix the same diffusion model and vary only the injection rule: (i) *Global sampling* draws synthetic records from the conditional generator without anchoring to a real example. (ii) *Random inpainting* selects an anchor uniformly from the current dataset and regenerates a random subset of columns. (iii) *Hard inpainting* selects high-uncertainty anchors, applies a fixed inpainting configuration, and generates the full budget in one shot. The configuration is specified in Appendix D.3. (iv) `TAP` learns a state-conditioned policy over targets and inpainting templates, and commits only when pooled utility is reliably positive.

*Hard inpainting* is a deliberately strong reference because it already anchors proposals and targets uncertain regions. The remaining gap to `TAP` highlights the benefit of sequential feedback and conservative commitment. Figure 3 shows a consistent ordering, with utility improving from global sampling to `TAP`.

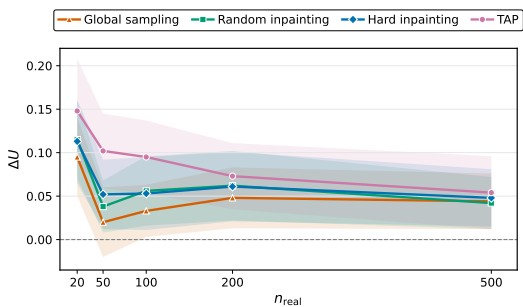

*Figure 3.* **Utility gain across injection methods with a shared diffusion backbone.** Shaded regions denote 95% CIs.

**Operationalizing informative yet learnable samples.** The ladder shows that `TAP` outperforms *Hard inpainting*,

yet both anchor on uncertain samples. To explain this gap, we introduce two post-hoc diagnostics that are used only for analysis and are not available to the policy during training. We measure *informativeness* as proximity to the decision boundary:

$$s_{\text{bnd}}(x) = \begin{cases} H(p_\theta(\cdot \mid x)), & \text{classification} \\ \text{Var}(f_\theta(\text{kNN}(x))), & \text{regression} \end{cases} \quad (21)$$

and *learnability* as label consistency:

$$s_{\text{con}}(x, y) = \begin{cases} -\log p_\theta(y \mid x), & \text{classification} \\ (y - f_\theta(x))^2, & \text{regression} \end{cases} \quad (22)$$

Higher $s_{\text{bnd}}$ indicates more uncertain regions, while lower $s_{\text{con}}$ indicates more learnable samples.

Figure 2 compares methods under matched informativeness. At comparable informativeness, `TAP` achieves better learnability, supporting the view that high-utility samples are not merely near the boundary but are informative and learnable.

**Interventional check.** The diagnostics above are correlational. To provide an interventional check, we partition candidates into five learnability bins and inject from each bin in turn. Figure 4 shows that utility concentrates in the middle bins where samples are informative yet learnable. The least learnable tail yields negative utility, which supports Principle 2, which states that boundary proximity alone is insufficient. This pattern is consistent with the observation that anchored inpainting produces fewer severely inconsistent samples than unanchored global draws, especially in the least learnable region.

### 4.4. Conservative Commitment Prevents Harm

The preceding analysis reveals a harmful tail that must be avoided. Principle 3 argues for conservative commitment via gating and windowed commitment. We test whether these mechanisms are necessary.

**Setup.** We compare `TAP` with two ablations that keep all other components fixed. The first removes point-wise gating and admits all candidates. The second removes windowed commitment and commits at every step. We focus on

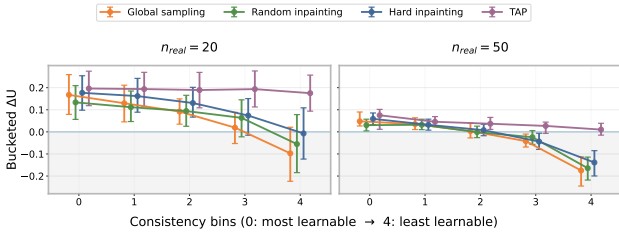

*Figure 4.* **Bucketed injection.** Utility by learnability bin where 0 is most learnable, and 4 is least learnable. Gains concentrate in the middle bins, and the harmful tail degrades performance.

$n_{\text{real}} \in \{20, 50, 100\}$ where robustness matters most. We report utility gain $\Delta\mathcal{U}$, win-rate as the fraction of runs with positive gain, and tail risk as the mean $s_{\text{con}}$ over the 20% injected samples with the largest $s_{\text{con}}$.

**Results.** Table 2 shows that both mechanisms contribute to reliability. Removing gating sharply increases tail risk and reduces win rate, consistent with its role in enforcing hard feasibility. Removing windowed commitment has a smaller but consistent negative effect, which is consistent with its role in filtering noisy utility estimates. Together, the two components address distinct failure modes: gating removes individual candidates that are invalid, while windowed commitment rejects pools whose joint gain is uncertain. In addition, plug-in uncertainty calibration achieves near-nominal coverage across datasets, which supports using $\epsilon_t$ as a conservative margin; details are in Appendix E.3.

*Table 2.* **Ablation on safe injection under extreme scarcity.** Utility gain $\Delta\mathcal{U}$, win-rate (% runs with $\Delta\mathcal{U} > 0$), and tail risk (mean inconsistency percentile over the worst 20% injected samples) for `TAP` and ablations removing gating or commitment.

| Method | $n_{\text{real}}{=}20$ | $n_{\text{real}}{=}50$ | $n_{\text{real}}{=}100$ | Win-rate ↑ | Tail risk ↓ |
|---|---|---|---|---|---|
| TAP | $0.140_{\pm 0.057}$ | $0.095_{\pm 0.039}$ | $0.097_{\pm 0.043}$ | 100.0% | 46.7% |
| – Gate | $0.108_{\pm 0.049}$ | $0.037_{\pm 0.052}$ | $0.052_{\pm 0.052}$ | 57.3% | 61.9% |
| – Commit | $0.134_{\pm 0.051}$ | $0.083_{\pm 0.043}$ | $0.085_{\pm 0.043}$ | 85.3% | 47.2% |

`TAP` achieves consistent utility gains by selecting informative yet learnable samples, rather than relying on fidelity or boundary proximity alone. Conservative commitment via gating and windowed commitment prevents harmful injection, making augmentation reliable under scarcity.

## 5. Related Work

**Generative modeling for tabular data.** A variety of approaches model the joint distribution of heterogeneous tabular features. Early methods such as TVAE and CTGAN (Xu et al., 2019) transform mixed-type features into continuous space, while tree-based estimators (Watson et al., 2023) and normalizing flows (Durkan et al., 2019; Yang et al., 2025) operate on preprocessed representations. Among diffusion models, TabDDPM (Kotelnikov et al., 2023) embeds categorical features before applying Gaussian diffusion, Tab-Syn (Zhang et al., 2024) operates in a VAE latent space,

and TabDiff (Shi et al., 2025) learns separate processes for continuous and categorical features. Language model-based approaches (Borisov et al., 2023; Yang et al., 2024; Zhang et al., 2025) leverage pretrained knowledge but face scalability challenges. These methods optimize distributional fidelity, yet high fidelity does not guarantee downstream utility. Under scarcity, passive sampling often produces redundant samples in already-covered regions.

**Data augmentation and sample selection.** Tabular augmentation differs from images or text because features are heterogeneous and tightly constrained (Cui et al., 2024; Jiang et al., 2025). Classic methods such as SMOTE (Chawla et al., 2002) can remain effective despite low fidelity, while Mixup variants (Zhang et al., 2018; Darabi et al., 2021) require adaptation to mixed-type columns. Recent generative approaches include TabEBM (Margeloiu et al., 2024) and LLM-based methods (Seedat et al., 2024), which often rely on post-hoc filtering. On the selection side, AutoAugment (Cubuk et al., 2019) and RandAugment (Cubuk et al., 2020) learn policies over fixed transforms, curriculum learning (Hacohen & Weinshall, 2019) adapts to learner state, and active learning (Gal et al., 2017; Ash et al., 2020) selects informative instances with oracle labels. Most augmentation methods lack learner feedback (Manousakas & Aydöre, 2023), and most selection methods do not control generation. In contrast, `TAP` treats the generator as a controllable proposal mechanism and steers synthesis conditioned on learner state, rather than passively sampling from a fixed model or injecting task signals into the diffusion trajectory (Jia et al., 2024). It further adopts conservative commitment to inject samples only when utility is consistently indicated, bridging generation and selection for reliable augmentation under data scarcity.

## 6. Conclusion

In this work, we introduced `TAP`, a utility-aligned framework for tabular augmentation that directly addresses the fidelity–utility gap. Instead of sampling synthetic records to mimic the joint distribution, `TAP` casts augmentation as sequential control of a proposal kernel. It couples diffusion inpainting with a learned policy that adapts generation to the learner's state, while hard gating and windowed commitment make injection reliable under noisy utility estimates. Across diverse tasks and scarcity regimes, `TAP` consistently outperforms strong generative baselines, improving accuracy by up to 15.6 percentage points, reducing RMSE by up to 32%, and reducing tail risk from harmful samples. A limitation is that sequential injection introduces additional overhead compared to one-shot generation. We mitigate this cost through window-level caching and training-free evaluators, and we view more efficient utility estimation as an important direction for future work.

## Acknowledgements

This work was partially supported by the Verband der Vereine Creditreform e.V..

## Impact Statement

Data scarcity in machine learning remains a practical barrier for many real-world deployments, due to cost, privacy, and domain constraints (Alzubaidi et al., 2023). While data augmentation can improve robustness (Rebuffi et al., 2021), it can be fragile in low-data regimes. Each injected record may have outsized influence, and harmful synthetic samples can degrade downstream performance. We believe TAP advances reliable augmentation for low-data tabular settings by reducing harmful injections through conservative selection and commitment. This can benefit deployment in domains with limited data, such as finance and healthcare (Alami et al., 2020), and in settings involving underrepresented subgroups (Suresh & Guttag, 2021).

At the same time, synthetic data augmentation should be applied carefully in high-stakes settings. Tabular data often reflects measurement noise, institutional practices, and historical inequities, and augmentation can propagate these effects if they are present in the training data. TAP is designed to reduce the risk of harmful injections under scarcity by combining manifold-local diffusion inpainting with hard feasibility gates and conservative windowed commitment, so that samples are injected only when utility is consistently indicated. We recommend validating gains on real held-out data, reporting subgroup metrics when available, and documenting intended use and known limitations following established dataset documentation practices (Gebru et al., 2021).

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

## A. Workflow

To give a clear view of how `TAP` works, we give a visualisation of the general workflow in Figure 5.

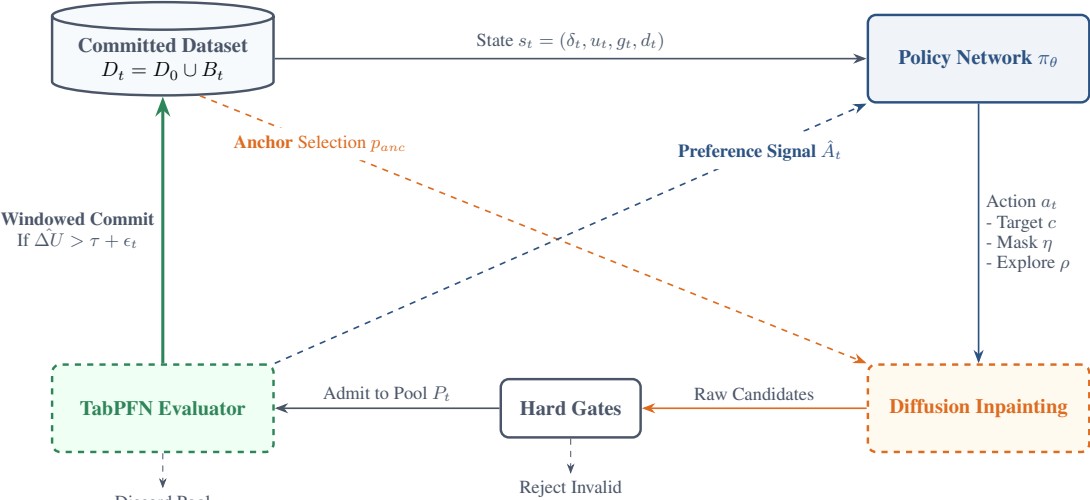

*Figure 5.* Overview of the `TAP` framework. `TAP` frames data augmentation as a sequential control process. At each step, a learnable policy observes the learner's state to guide a frozen diffusion inpainting kernel. Proposed candidates undergo hard feasibility gating and are accumulated in a temporary pool. A frozen online evaluator assesses the pool's utility, providing advantage signals for preference-based policy optimization and triggering safe windowed commitment to the downstream dataset.

## B. Theoretical Details

This section gathers the proofs and supporting results referenced in the main text. We first prove the telescoping decomposition in Equation (7), then derive an error bound relating plug-in utility to true utility under a sufficient accuracy condition, and finally establish Theorem 3.1. We also provide a sufficient condition that explains the design rationale for our state summary in action ranking. Together, these results offer safety guarantees and design intuition rather than a global optimality guarantee.

### B.1. Proof of Utility Telescoping (Equation (7))

*Proof.* Let $D_t = D_0 \cup B_t$ and let $0 = t_0 < t_1 < \cdots < t_M = T$ denote the commitment times. For each $i = 0, \ldots, M-1$, let $\widetilde{P}_i$ denote the pool accumulated in window $i$ and evaluated at time $t_{i+1}$. Let $A_i \in \{0, 1\}$ indicate whether the commitment rule accepts at time $t_{i+1}$, and define the committed pool

$$P_i := \begin{cases} \widetilde{P}_i, & A_i = 1, \\ \emptyset, & A_i = 0. \end{cases} \tag{23}$$

The committed dataset therefore updates as

$$D_{t_{i+1}} = D_{t_i} \cup P_i. \tag{24}$$

By the definition of marginal utility and the update above,

$$L(\theta(D_{t_i})) - L(\theta(D_{t_{i+1}})) = \Delta U(D_{t_i}, P_i). \tag{25}$$

Summing over $i = 0, \ldots, M-1$ yields

$$L(\theta(D_0)) - L(\theta(D_T)) = \sum_{i=0}^{M-1} \Delta U(D_{t_i}, P_i), \tag{26}$$

which proves Equation (7). $\square$

## B.2. Plug-in Utility Error Bound

We state a sufficient condition under which the plug-in objective approximates the true utility.

**Assumption B.1** (A sufficient plug-in accuracy condition). There exists $\varepsilon_L \geq 0$ such that for any dataset $D$ reachable by the algorithm,

$$\left| \widehat{L}_\psi(D) - L(\theta(D)) \right| \leq \varepsilon_L. \tag{27}$$

In practice, we replace a uniform $\varepsilon_L$ with a step-dependent error bar estimated from focused queries, which is used in the commitment rule.

**Lemma B.2** (Induced plug-in utility error). *Under Assumption B.1, for any candidate set S,*

$$\left| \widehat{\Delta U}_\psi(D, S) - \Delta U(D, S) \right| \leq 2\varepsilon_L. \tag{28}$$

*Proof.* By definition,

$$\widehat{\Delta U}_\psi(D, S) - \Delta U(D, S) = \left( \widehat{L}_\psi(D) - L(\theta(D)) \right) - \left( \widehat{L}_\psi(D \cup S) - L(\theta(D \cup S)) \right). \tag{29}$$

Applying the triangle inequality and Assumption B.1 to both terms yields the bound. $\square$

## B.3. Proof of Theorem 3.1

*Proof.* Let $E_t$ be the event in Equation (20). On $E_t$,

$$\Delta U(D_t, P_t^{(K)}) \geq \widehat{\Delta U}_{K,\psi}(D_t, P_t^{(K)}) - \epsilon_t. \tag{30}$$

If `TAP` commits only when $\widehat{\Delta U}_{K,\psi}(D_t, P_t^{(K)}) > \tau + \epsilon_t$, then on $E_t$ we have $\Delta U(D_t, P_t^{(K)}) \geq \tau$. Since $\Pr(E_t) \geq 1 - \alpha$, the claim follows. $\square$

## B.4. A Sufficient Condition for Action Ranking

This subsection connects Equation (15) to the components of $s_t$. Here, we provide a heuristic justification for the action ranking design rather than a formal optimality guarantee.

**A practical scoring heuristic for action ranking.** Equation (15) decomposes expected gain into a feasibility term and a conditional utility term. In practice, we approximate feasibility using recent gate pass rates and approximate conditional utility using state signals that capture deficits, uncertainty, and redundancy. We combine these signals multiplicatively to form an action score. This heuristic is motivated by the factorized structure of Equation (15), but it does not provide a formal guarantee that score ordering matches the true expected-gain ordering for all action pairs. We therefore treat it as design intuition and validate its effectiveness empirically through state ablations in Appendix F.

**Gate rates.** The term $\Pr(G = 1 \mid s_t, a_t)$ depends on how often each template passes validity constraints under the current dataset and scarcity regime. Tracking recent per-template pass rates provides a simple empirical proxy for this factor that is stable in practice.

**Target deficits and uncertainty.** The conditional component $\mathbb{E}[\widehat{\Delta U}_\psi(D_t, S_t) \mid s_t, a_t, G = 1]$ is controlled by which targets are under-covered and where the current plug-in evaluator is uncertain. The deficit statistic $\delta_t$ measures miscoverage relative to the desired target mixture, while $u_t$ summarizes predictive uncertainty over targets or bins.

**Diversity.** When admitted samples are near-duplicates of committed points, marginal gains diminish. A diversity score $d_t$ summarizes novelty relative to $B_t$ and the current pool, and it is computed from information available before sampling $a_t$.

## B.5. Derivation of Equation (3)

This approximation is local and is most accurate when the parameter shift induced by adding one example is small. In the severe-scarcity regime, it can be crude numerically. We therefore use it as a qualitative guide for state and action design,

while relying on empirical utility estimation and conservative commitment for reliable decisions. We derive a first-order approximation for the marginal utility of injecting a single example.

Let $D = \{z_i\}_{i=1}^n$ with $z_i = (x_i, y_i)$. Consider regularized empirical risk minimization:

$$F_D(\theta) := \frac{1}{n} \sum_{i=1}^n \ell(f_\theta(x_i), y_i) + \lambda R(\theta), \qquad \theta(D) \in \arg\min_\theta F_D(\theta). \tag{31}$$

Let $H_D := \nabla_\theta^2 F_D(\theta(D))$.

**Lemma B.3.** *Assume $F_D$ is twice differentiable and locally strongly convex at $\theta(D)$. We ignore the $1/(n+1)$ versus $1/n$ difference for notational simplicity and drop higher-order terms, which does not affect the qualitative first-order dependence on the added example. Let $D' = D \cup \{z\}$ with $z = (x, y)$. Then,*

$$\theta(D') - \theta(D) \approx -\frac{1}{n} H_D^{-1} \nabla_\theta \ell(f_{\theta(D)}(x), y). \tag{32}$$

*Proof.* The first-order optimality conditions give $\nabla_\theta F_D(\theta(D)) = 0$ and $\nabla_\theta F_{D'}(\theta(D')) = 0$. Apply a first-order Taylor expansion of $\nabla_\theta F_{D'}$ around $\theta(D)$ and solve for $\theta(D') - \theta(D)$, which yields Equation (32) up to higher-order terms. □

**Proposition B.4.** *Assume $L(\theta)$ in Equation (1) is differentiable. Then the marginal utility of injecting $z$ admits the approximation*

$$\Delta U(D, \{z\}) \approx \frac{1}{n} \nabla_\theta L(\theta(D))^\top H_D^{-1} \nabla_\theta \ell(f_{\theta(D)}(x), y), \tag{33}$$

*which matches Equation (3).*

*Proof.* Apply a first-order Taylor expansion of $L(\theta)$ around $\theta(D)$:

$$L(\theta(D')) \approx L(\theta(D)) + \nabla_\theta L(\theta(D))^\top (\theta(D') - \theta(D)). \tag{34}$$

Substitute Lemma B.3 into Equation (34) and rearrange. □

## B.6. A Pareto view of informativeness and learnability

This subsection provides an explanatory lens for Figure 2. The diagnostics are not optimized directly by the algorithm. Instead, we show that under a mild monotonicity condition, utility maximization within a matched informativeness slice prefers more learnable samples.

**Diagnostics.** Let $s_{\mathrm{bnd}}(x)$ denote the decision boundary score used to bucket informativeness in Section 4.3. Let $s_{\mathrm{con}}(x, y)$ denote the post hoc inconsistency score used as a proxy for learnability. Smaller $s_{\mathrm{con}}$ indicates higher learnability.

**Matched informativeness slices.** Fix a percentile bucket $\mathcal{B}$ induced by $s_{\mathrm{bnd}}$. We consider candidate pools whose elements fall in the same bucket $\mathcal{B}$, which matches the evaluation protocol in Figure 2.

**Assumption B.5** (Utility monotonicity within a bucket)**.** For a fixed dataset state $D$ and a fixed bucket $\mathcal{B}$, consider two candidate pools $S$ and $S'$ whose elements lie in $\mathcal{B}$. If the average inconsistency in $S$ is no larger than that in $S'$, then $\Delta U(D, S) \geq \Delta U(D, S')$.

**Proposition B.6** (Pareto efficiency under matched informativeness)**.** *Fix $D$ and a bucket $\mathcal{B}$. Let $\mathcal{S}_\mathcal{B}$ be the family of candidate pools with elements in $\mathcal{B}$. Under Assumption B.5, any maximizer of $\Delta U(D, S)$ over $\mathcal{S}_\mathcal{B}$ also minimizes average inconsistency over $\mathcal{S}_\mathcal{B}$. Equivalently, it is Pareto efficient in the plane defined by informativeness and learnability when restricted to the bucket.*

*Proof.* The assumption states that within $\mathcal{B}$, ordering by utility agrees with ordering by negative inconsistency. Therefore any utility maximizer must achieve the smallest inconsistency among feasible pools in $\mathcal{S}_\mathcal{B}$. □

---

**Algorithm 1** `TAP`: Policy-Guided Tabular Augmentation

---

**Require:** Initial labeled set $D_0$, diffusion backbone $q_\phi$, horizon $T$, window size $K$, threshold $\tau$
1: Initialize policy $\pi$
2: Initialize committed buffer $B \leftarrow \emptyset$            // cumulative across windows
3: Initialize pool $P \leftarrow \emptyset$             // temporary within a window
4: **for** $t = 0$ **to** $T - 1$ **do**
5:      Set $D_t \leftarrow D_0 \cup B$
6:      Compute state summary $s_t$ from $(D_t, P)$
7:      Sample action $a_t \sim \pi(\cdot \mid s_t)$
8:      Propose candidates $\widetilde{S}_t \sim \mathcal{K}_\phi(\cdot \mid D_t, a_t)$
9:      Apply pointwise feasibility gates $G(\cdot; D_t)$ and admit
10:       $S_t \leftarrow \{x \in \widetilde{S}_t \mid G(x; D_t) = 1\}$
11:     Compute a preference signal using $\widehat{\Delta U}_\psi(D_t, S_t)$
12:       (cache $\widehat{L}_\psi(D_t)$ within a window and use forward passes for $\widehat{L}_\psi(D_t \cup S_t)$)
13:     Update $\pi$ by regularized preference optimization
14:     Update pool $P \leftarrow P \cup S_t$
15:     **if** $(t + 1) \bmod K = 0$ **then**
16:       Compute error bar $\epsilon_t$
17:       **if** $\widehat{\Delta U}_{K,\psi}(D_t, P) > \tau + \epsilon_t$ **then**
18:         Commit $B \leftarrow B \cup P$
19:       **end if**
20:       Reset pool $P \leftarrow \emptyset$           // discard if not committed
21:     **end if**
22: **end for**
23: **return** $D_T = D_0 \cup B$

---

**Implication for `TAP`.** Our policy is trained to maximize a plug-in estimate of $\Delta U$ and our commitment rule filters out pools whose estimated gain is uncertain. When the plug-in error is bounded as in Equation (20), pools with larger estimated gain also tend to have larger true gain up to the estimation error. This supports reading Figure 2 as evidence that `TAP` selects more learnable samples at comparable informativeness. We view this as an explanatory result rather than a global optimality guarantee for the learned policy.

## C. Additional Method Details

### C.1. Complete `TAP` Procedure

We use pointwise feasibility gates $G(x; D_t) \in \{0, 1\}$ to filter invalid candidates before they enter the pool. Within a window of length $K$, the running pool variable $P$ equals the pooled set $P_t^{(K)}$ at the commitment check in the main text.

### C.2. `TAP` Mechanism Settings

This subsection documents the `TAP` mechanism at the level needed for reproducibility. We describe the state vector, the action parameterization, the induced proposal kernel, and the step-wise control flow. We also clarify how each action component affects the inpainting distribution.

#### C.2.1. STATE CONSTRUCTION

We encode the learner state using the compact summary

$$s_t := (\delta_t, u_t, g_t, d_t), \tag{35}$$

which mirrors Equation (16) in the main text. Each component is computed from the current committed dataset $D_t = D_0 \cup B_t$ and the current pool $P$ (i.e., $P_t$) that has not yet been committed.

**Target deficit** $\delta_t$. $\delta_t$ measures coverage mismatch between the desired target mixture and the current realized mixture from real plus committed synthetic data. For classification, targets correspond to class labels. For regression, targets correspond to quantile bins.

**Uncertainty proxy** $u_t$. $u_t$ aggregates predictive uncertainty of the plug-in evaluator over the target partition. This quantity is aligned with the focused query set construction, which selects informative queries using the current predictor.

**Gate statistic** $g_t$. $g_t$ stores recent gate pass rates per mask template. It estimates feasibility for each template, which appears as $\Pr(G = 1 \mid s_t, a_t)$ in Equation (15).

**Diversity score** $d_t$. To keep the state definition temporally consistent, $d_t$ is computed before sampling $a_t$ using only the committed buffer $B_t$ and the current pool $P$. It measures redundancy by a nearest-neighbor distance between pooled samples and the committed buffer.

### C.2.2. ACTION PARAMETERIZATION AND SAMPLING

We parameterize generation by the compact action

$$a_t := (c_t, \eta_t, \rho_t), \tag{36}$$

where $c_t$ selects a target condition, $\eta_t$ selects a mask template, and $\rho_t \in [0, 1]$ controls exploration strength.

**Policy factorization.** We use a factorized stochastic policy

$$\pi_\omega(a_t \mid s_t) = \pi_\omega(c_t \mid s_t)\,\pi_\omega(\eta_t \mid s_t)\,\pi_\omega(\rho_t \mid s_t).$$

The discrete components $(c_t, \eta_t)$ are sampled from categorical distributions parameterized by logits produced by an MLP on $s_t$. The continuous component $\rho_t$ is sampled from a Gaussian whose mean and scale are also produced by the same MLP, and the sampled value is clamped to $[0, 1]$.

### C.2.3. CONTROLLED PROPOSAL KERNEL INDUCED BY THE ACTION

An action $a = (c, \eta, \rho)$ induces a proposal distribution through anchor selection, mask construction, and diffusion sampling randomness. We write the induced proposal family as

$$Q_a(\cdot \mid D_t) = \mathbb{E}_{x \sim p_{\mathrm{anc}}(\cdot \mid D_t, c)} \mathbb{E}_{m \sim p_{\mathrm{mask}}(\cdot \mid \eta, \rho)} \left[ q_\phi(\cdot \mid x, m, c) \right], \tag{37}$$

and we sample a candidate batch by $\widetilde{S}_t \sim \mathcal{K}_\phi(\cdot \mid D_t, a_t)$.

**Anchor selection** $p_{\mathrm{anc}}(\cdot \mid D_t, c)$. We select anchor records from the current dataset restricted to the target condition $c$. For classification, $c$ indexes a class and anchors are drawn from that class. For regression, $c$ indexes a quantile bin and anchors are drawn from that bin. Within the restricted set, anchors are chosen using a mixture of hard-sample preference and uniform sampling. Hardness is measured by the plug-in evaluator's uncertainty or error.

**Mask template** $p_{\mathrm{mask}}(\cdot \mid \eta, \rho)$. A mask $m \in \{0, 1\}^d$ indicates regenerated coordinates, where $m_j = 1$ means regenerate feature $j$ and $m_j = 0$ means keep it fixed. We implement two templates.

- **Explore template.** This template fixes only the label condition and allows regeneration across all non-label features.

- **Conservative template.** This template fixes the label condition and fixes an additional set of important columns, determined by a combination of mutual information with the target and bootstrap stability of feature statistics.

**Exploration strength** $\rho$. The scalar $\rho$ controls locality through mask construction. When $\rho < 1$, we additionally fix a random subset of numeric features that would otherwise be regenerated. Smaller $\rho$ therefore yields more conservative near-anchor proposals.

C.2.4. HARD FEASIBILITY GATES AND ADMISSION

We apply pointwise gating to every proposed sample $x \in \widetilde{S}_t$. The gate function $G(x; D_t) \in \{0, 1\}$ checks domain constraints and rejects candidates that violate hard feasibility. The admitted set is

$$S_t := \{x \in \widetilde{S}_t \mid G(x; D_t) = 1\}. \tag{38}$$

In our implementation, $G$ includes at least the following checks.

- **Type validity.** All categorical features must take values within the set observed in the real training data, and all numeric features must be finite.

- **Range validity.** Numeric features must lie within a conservative range derived from real training statistics. In all experiments, numeric features are clipped to the quantile range $[q_{\min}, q_{\max}]$ computed on the real training split, with $q_{\min} = 0.01$ and $q_{\max} = 0.99$ in all experiments.

- **Task-dependent logical checks.** If the dataset includes known logical relations, we enforce them as deterministic constraints.

C.2.5. CONSERVATIVE WINDOWED COMMITMENT

Let $P$ denote the running pool variable at a commitment check, which corresponds to $P_t^{(K)}$ in the main text. We compute pooled plug-in utility

$$\widehat{\Delta U}_{K,\psi}(D_t, P) := \widehat{L}_\psi(D_t) - \widehat{L}_\psi(D_t \cup P), \tag{39}$$

and we commit the pool only when $\widehat{\Delta U}_{K,\psi}(D_t, P) > \tau + \epsilon_t$. This implements a confidence-based selection rule that reduces tail risk from noisy per-step estimates and captures complementarities among admitted candidates.

## C.3. Preference-Based Policy Optimization Details

We implement the regularized improvement objective in Equation (17) using a preference-based update. At each step we compute a scalar advantage $\widehat{A}_t$ from the plug-in utility and convert it into binary feedback with abstention.

**From KL regularized improvement to preference learning.** In the main text we optimize a KL regularized improvement objective against a conservative reference policy $\pi_{\mathrm{ref}}$, namely

$$\max_\pi \; \mathbb{E}\left[\widehat{A}_t\right] - \beta \, \mathrm{KL}(\pi(\cdot \mid s_t) \, \| \, \pi_{\mathrm{ref}}(\cdot \mid s_t)), \tag{40}$$

where $\widehat{A}_t$ is a baseline-corrected advantage computed from the plug-in utility.

Direct regression on $\widehat{A}_t$ can be unstable because utility estimates may be heavy tailed and their scale can drift across scarcity regimes. We therefore instantiate Equation (40) using a preference-based update that only requires *binary* feedback.

**Binary feedback construction.** At each step $t$ we map the scalar advantage $\widehat{A}_t$ to a preference signal $z_t \in \{+1, -1, \emptyset\}$,

$$z_t := \begin{cases} +1, & \widehat{A}_t > \kappa_t, \\ -1, & \widehat{A}_t < -\kappa_t, \\ \emptyset, & \text{otherwise}, \end{cases} \tag{41}$$

where $\kappa_t$ is an adaptive threshold based on a running scale estimate. Samples with $z_t = \emptyset$ are skipped, which reduces gradient noise when the estimated advantage magnitude is small.

**Log ratio parameterization.** Define the per-sample log ratio

$$r_\omega(s_t, a_t) := \log \pi_\omega(a_t \mid s_t) - \log \pi_{\mathrm{ref}}(a_t \mid s_t). \tag{42}$$

This quantity is a control variable that measures how strongly the learned policy deviates from the conservative reference on the sampled action. It also appears naturally in the solution structure of KL regularized policy improvement, where the optimal policy has the form $\pi^\star(a \mid s) \propto \pi_{\mathrm{ref}}(a \mid s) \exp(\widehat{A}(s, a)/\beta)$.

**KTO style objective.** We instantiate KTO (Ethayarajh et al., 2024) as a *direct* optimizer over the policy that pushes $r_\theta$ upward on desirable actions and downward on undesirable actions, with asymmetric weighting that reflects loss aversion. Concretely, for a minibatch $\mathcal{B}$ of labeled tuples $(s_t, a_t, z_t)$ with $z_t \neq \emptyset$, we minimize

$$\mathcal{L}_{\text{KTO}}(\omega) = -\lambda_D \, \mathbb{E}_{(s,a,z)\sim\mathcal{B}}[\mathbb{1}\{z = +1\} \log \sigma(r_\omega(s,a))] - \lambda_U \, \mathbb{E}_{(s,a,z)\sim\mathcal{B}}[\mathbb{1}\{z = -1\} \log \sigma(-r_\omega(s,a))], \quad (43)$$

where $\sigma(\cdot)$ is the logistic sigmoid, and $\lambda_D, \lambda_U > 0$ are adaptive weights.

Equation (43) is a stable preference objective that avoids value function fitting. It increases $\pi_\theta(a \mid s)$ relative to $\pi_{\text{ref}}(a \mid s)$ when $z = +1$, and it decreases it when $z = -1$. This operationalizes Equation (40) using sign information from $\widehat{A}_t$ rather than its raw magnitude.

**Adaptive weighting.** We set $\lambda_D$ and $\lambda_U$ using the observed ratio of desirable versus undesirable samples in the minibatch. The goal is to avoid regimes where nearly all feedback is of one type, which would otherwise cause either overly aggressive deviation from $\pi_{\text{ref}}$ or overly conservative updates. A simple instantiation is

$$\lambda_D \propto \frac{1}{\max(1, |\{z = +1\}|)} \qquad \text{and} \qquad \lambda_U \propto \frac{1}{\max(1, |\{z = -1\}|)}.$$

Any equivalent normalization that balances the two terms is acceptable.

**Reference policy.** The reference policy $\pi_{\text{ref}}$ is heuristic and conservative. It prioritizes actions that reduce target deficits and it reduces exploration when feasibility is low, which stabilizes learning early in training. This choice makes the KL regularizer in Equation (40) operationally meaningful because it anchors learning to a safe default.

**Implementation note for mixed action types.** Our action $a = (c, \eta, \rho)$ includes discrete components $(c, \eta)$ and a continuous component $\rho \in [0, 1]$. We factorize the policy as

$$\pi_\theta(a \mid s) = \pi_\theta(c \mid s) \, \pi_\theta(\eta \mid s) \, \pi_\theta(\rho \mid s), \quad (44)$$

so $\log \pi_\theta(a \mid s)$ decomposes additively. For $\rho$ we use a Gaussian policy and clamp its sampled value to $[0, 1]$. When computing $\log \pi_\theta(\rho \mid s)$, we use the pre-clamp density evaluation, which yields stable gradients in practice.

## D. Experimental Details

### D.1. Datasets

We evaluate on seven real world tabular datasets that span healthcare, finance, science, and operations. The first six datasets are publicly available on OpenML (Bischl et al., 2017), and the seventh dataset Insurance is sourced from Kaggle. Dataset statistics, task types, and feature compositions are reported in Table 3.

*Table 3.* **Statistics of the real-world datasets used in our experiments.** # Samples, # Features and # Classes denote the numbers of samples, features and classes in tabular datasets, respectively.

| Dataset | Domain | # Samples | # Features | Task | # Classes |
|---|---|---|---|---|---|
| MiceProtein (Higuera et al., 2015) | Medical | 1,080 | 77 | Classification | 8 |
| Credit-G (Asuncion et al., 2007) | Finance | 16,087 | 20 | Classification | 2 |
| Electricity (Gama et al., 2004) | Energy | 45,312 | 8 | Classification | 2 |
| Fourier (Asuncion et al., 2007) | Synthetic | 2,000 | 76 | Classification | 10 |
| Steel (Asuncion et al., 2007) | Manufacturing | 1,941 | 33 | Classification | 2 |
| Ailerons (Grinsztajn et al., 2022) | Engineering | 13,750 | 40 | Regression | - |
| Insurance (From Kaggle[1]) | Finance | 1,338 | 6 | Regression | - |

[1] https://www.kaggle.com/datasets/mirichoi0218/insurance

**Data splitting.** We follow the data splitting protocol used in TabEBM (Margeloiu et al., 2024). Given a dataset of size $N$, we first construct a held-out test set of size $N_{\text{test}} = \min\big(\big\lfloor \frac{N}{2} \big\rfloor, 500\big)$, and we denote the remaining examples as the oracle set, with size $N_{\text{oracle}} = N - N_{\text{test}}$. The oracle set is used only for simulating data scarcity. For each scarcity level $n_{\text{real}} \in \{20, 50, 100, 200, 500\}$, we sample a real labeled subset of size $n_{\text{real}}$ from the oracle set and split it into a real training split and a real validation split with an $80\%$ to $20\%$ ratio. The generator and the augmentation policy are trained using only the real training split. Synthetic samples are injected only into the downstream predictor training set. The real validation set is used for model selection and early stopping when applicable. The real test set is used only for final evaluation. We repeat this procedure over five random splits for each dataset and each $n_{\text{real}}$. When computing plug-in utility for policy learning, we construct cross-validation folds from the real training split to define real query examples. The evaluator is conditioned on the current committed dataset $D_t = D_0 \cup B_t$ during policy learning.

## D.2. Implementation Details

We implement `TAP` in PyTorch (Paszke et al., 2019) and follow Algorithm 1 in Appendix C.1. At decision step $t$, the policy observes the state summary $s_t$ and samples an action $a_t = (c, \eta, \rho)$ as in Equation (10). The action instantiates a proposal kernel $\mathcal{K}_\phi(\cdot \mid D_t, a_t)$ through anchor selection, mask construction, and diffusion inpainting. Candidates are filtered by pointwise feasibility gates, accumulated into a pool, and committed with windowed evaluation every $K$ steps. Policy updates use a KL-regularized preference objective, with details in Appendix C.3. `TAP` employs the same hyperparameter setting for all datasets and scarcity levels unless otherwise stated. All experiments were conducted on an NVIDIA A100 GPU with 80GB memory.

### D.2.1. DIFFUSION BACKBONE

Tabular datasets are typically mixed-type, combining continuous and categorical fields that obey strong cross-column constraints. We use TabDiff (Shi et al., 2025) as the diffusion backbone $q_\phi$ for all `TAP` experiments because it models mixed-type tables with type-aware noise schedules, which yields strong generative fidelity and provides a reliable backbone for inpainting proposals.

Given an anchor record $x$ and a binary regeneration mask $m \in \{0, 1\}^d$, inpainting samples a synthetic record by conditioning on the fixed coordinates,

$$x^{\text{syn}} \sim q_\phi(x_m \mid x_{\bar{m}}, c), \tag{45}$$

where $c$ denotes the target condition and $x_{\bar{m}}$ denotes the fixed columns. We implement inpainting by overwriting fixed coordinates at every reverse diffusion step as in Equation (9).

We adopt TabDiff with its default training and sampling configuration. Specifically, we train TabDiff for 8000 steps with learning rate $10^{-3}$ and batch size 4096, using EMA decay 0.997. For generation, we use stochastic sampling with second-order correction. The backbone is trained on the real training split and frozen during policy learning.

### D.2.2. ONLINE UTILITY ESTIMATION.

For policy learning, we estimate plug-in utility via $M_{\text{cv}}$-fold cross-validation folds constructed from the real training split, so the policy does not access validation labels. For each fold, the evaluator is conditioned on the remaining folds together with $B_t$, and the loss is computed only on the held-out fold. The same fold-wise construction is used when evaluating $\widehat{L}_\psi(D_t)$ and $\widehat{L}_\psi(D_t \cup S)$. Within each fold, we evaluate loss reduction on a focused subset of query examples given by the top-$\alpha$ fraction ranked by informativeness. We use predictive entropy for classification and predictive residual magnitude for regression.

We instantiate the plug-in evaluator $\psi$ with TabPFN (Hollmann et al., 2025), a prior-data fitted network that performs in-context learning without requiring gradient updates. This training-free property is essential for our iterative utility evaluation, as each decision step requires multiple forward passes through the evaluator. Traditional models would need retraining at each step, making the approach computationally prohibitive. TabPFN provides stable few-shot predictions by conditioning on the training set as context, which aligns well with the severe scarcity regime we target.

### D.2.3. HYPERPARAMETERS

Table 4 lists the key hyperparameters for `TAP`.

*Table 4.* **Key** `TAP` **hyperparameters used in all experiments.** We keep the same setting across all datasets and scarcity levels. The table summarizes the policy network and optimizer, the KTO-style preference update, the cross-validated plug-in utility estimation used for policy learning, and the generation, gating, and windowed commitment settings that control safe injection.

| Category | Hyperparameter | Value |
|---|---|---|
| Policy | MLP layers / hidden width | 2 / 128 |
| | Optimizer / learning rate | AdamW / $3 \times 10^{-4}$ |
| | Max grad norm | 0.5 |
| Preference update | KL coefficient $\beta$ | 3.0 |
| | Feedback quantile / window | 0.6 / 200 |
| Utility estimation | CV folds $M$ | 5 |
| | Focused query ratio $\alpha$ | 0.2 |
| Generation | Candidates per step | 16 |
| | Synthetic budget $n_{\mathrm{syn}}$ | 500 |
| Commitment | Commit interval $K$ | 20 |
| | Commit threshold $\tau$ | 0 |
| Gating | Numeric clipping quantiles | $[0.01, 0.99]$ |
| | Classification $p_{\min}$ / margin | 0.3 / 0.1 |
| | Regression residual percentile | 95 |
| | Diversity threshold | 0.1 |
| Regression only | Number of target bins | 7 |

### D.3. Baseline Configurations

We compare against SMOTE, TVAE, CTGAN, ARF, SPADA, TabDDPM, and TabDiff. We use the official implementations with recommended defaults. We use the same synthetic budget $n_{\mathrm{syn}} = 500$ and the same data splits for all methods. In addition, we evaluate several injection rules under a shared diffusion backbone for mechanism analysis.

**Hard inpainting.** *Hard inpainting* is used only in the mechanism analysis. It shares the same diffusion backbone and pointwise feasibility gates as `TAP`, but removes policy learning, sequential feedback, and windowed commitment. It selects anchors from the current dataset with high plug-in uncertainty, using predictive entropy for classification and residual magnitude for regression. We then generate the full synthetic budget in one shot using the conservative template and set $\rho = 0.3$ in all experiments.

**Adapting SMOTE to regression and low-sample regimes.** Standard SMOTE is designed for classification and oversamples minority classes by interpolating in feature space. Following Zhang et al. (2024), we adapt SMOTE to regression by operating in the joint space $(X, y)$. We construct a binary discrimination problem in which real samples are labeled as 0 and randomly generated Gaussian noise is labeled as 1. We then apply SMOTE to interpolate within the real class in the joint space and retain only synthetic samples classified as real by the discriminator. This procedure produces continuous target values while preserving feature-target correlations.

In low-sample regimes, the default $k = 5$ nearest neighbors can exceed the available samples. We therefore set $k = \min(5, n_{\min} - 1)$, where $n_{\min}$ is the minimum class size for classification or the total sample count for regression. If $k < 1$, we skip SMOTE and fall back to bootstrap resampling.

**Anomalous entries in Table 1.** TabDDPM on Ailerons is unstable under scarce training in our runs, leading to high variance and occasional failure cases. We use the official implementation with its recommended defaults and report the observed outcomes under the same splits and synthetic budget as other methods. Per-predictor tables in Appendix G.2 confirm that our main conclusions are not driven by a single downstream model.

### D.4. Evaluation Protocol

**Metrics.** For classification tasks, we report Accuracy and Macro F1. For regression tasks, we report RMSE and MAE. Following Zhang et al. (2024), we standardize continuous targets on the real training split when training regressors for RMSE based evaluation. We apply the same transformation to validation and test targets, and we report metrics on the standardized scale for consistent aggregation across datasets.

**Downstream predictors.** We evaluate augmentation quality using six classifiers and four regressors. The classifiers are Logistic Regression, KNN, MLP, Random Forest, XGBoost, and LightGBM. The regressors are KNN, Random Forest, XGBoost, and LightGBM. All ensemble methods use 100 estimators. KNN uses $k = 5$ neighbors. MLP uses a single hidden layer with 100 hidden units and a maximum of 500 iterations. Logistic Regression uses a maximum of 1000 iterations. All models use `random_state=42` for reproducibility.

**Aggregation and repetitions.** For each dataset and each scarcity level, we repeat experiments over five random splits. We report the mean and standard deviation across splits. When presenting aggregate tables, we further average performance across downstream predictors within each task type.

# E. Additional Analyses

## E.1. Policy Learning Dynamics

To verify that the policy learns meaningful behavior, we compare `TAP` against a **No-Learn** baseline that freezes the policy (skipping preference optimization in Algorithm 1) while keeping all other components identical.

**Evaluation protocol.** For each decision step, we compute a proxy reward using a held-out validation signal not used during training:

$$r_t^{\text{proxy}} = L(\theta(D_t), Q_{\text{proxy}}) - L(\theta(D_t \cup S_t), Q_{\text{proxy}}), \tag{46}$$

where $Q_{\text{proxy}}$ is reserved for diagnostics only. An action is *desirable* if $r_t^{\text{proxy}} > 0$. We report desirable rate per commitment window, aggregated over all datasets at $n_{\text{real}} = 50$, a scarcity level where augmentation is impactful yet sufficient signal exists for policy learning.

**Results.** Figure 6 shows desirable rates across commitment windows. In early windows (0-4), both methods perform similarly because `TAP` initializes from a conservative reference policy and KL regularization constrains gradual deviation. As training progresses, `TAP` increasingly outperforms No-Learn. The gap is most pronounced in windows 5-8, where `TAP` achieves desirable rates around 0.50-0.60 compared to 0.20-0.45 for No-Learn. This pattern confirms that the policy learns to improve upon the reference as preference feedback accumulates.

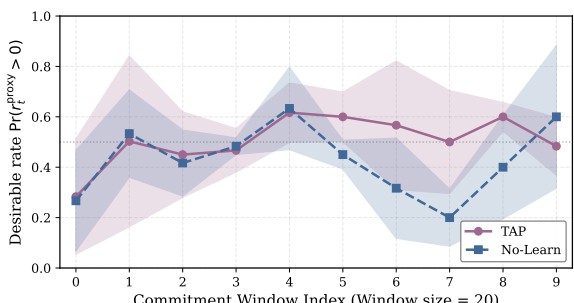

*Figure 6.* **Desirable rate (proxy reward $> 0$) across commitment windows, aggregated over all datasets at $n_{\text{real}} = 50$.** `TAP` learns to outperform the frozen baseline as training progresses.

## E.2. Sensitivity to Commitment Hyperparameters

We analyze sensitivity to the commitment window size $K$ and the threshold $\tau$ used in the conservative commitment rule.

To summarize robustness without listing all per-dataset sweeps, we report the worst-case accuracy drop from the default setting over the tested grid

$$\text{WorstDrop} = \max_{h \in \mathcal{H}} \Big( \text{Acc}(h_{\text{def}}) - \text{Acc}(h) \Big). \tag{47}$$

Table 5 shows aggregate WorstDrop statistics. The drops are small, indicating that performance is stable across a broad range of $K$ and $\tau$ values. On stable datasets such as Steel, accuracy is nearly unchanged across the grid. On more challenging settings such as MiceProtein, $\tau$ can provide a modest improvement at intermediate values, while the overall sensitivity remains mild.

*Table 5.* **Robustness to commitment hyperparameters.** We report the worst-case accuracy drop from the default setting over the tested grid, aggregated over representative settings.

| Hyperparameter | Grid | Default | WorstDrop in accuracy (mean / 90th / max) |
|---|---|---|---|
| $K$ | $\{1, 5, 10, 20, 50\}$ | 20 | 0.011 / 0.015 / 0.015 |
| $\tau$ | $\{0, 0.02, 0.05, 0.1, 0.2\}$ | 0 | 0.002 / 0.004 / 0.004 |

### E.3. Plug-in Utility Calibration

Theorem 3.1 assumes an error bar $\epsilon_t$ such that the pooled plug-in estimate $\widehat{\Delta U}_{K,\psi}(D_t, P_t^{(K)})$ provides a conservative approximation to the realized utility improvement $\Delta U(D_t, P_t^{(K)})$. We assess this empirically by comparing $\Delta \hat{U}_{K,\psi}$ against a retraining-based proxy.

**Practical note.** The plug-in evaluator is used only to rank candidate pools and to implement conservative commitment checks. Final results are reported by retraining standard downstream predictors on the committed augmented set, so the evaluator is not used for final reporting. Here we evaluate whether the estimated margin $\epsilon_t$ behaves as a reasonable conservative bound for commitment decisions under scarcity.

**Setup.** At each commitment check (every $K$ steps), we record the pooled plug-in estimate $\Delta \hat{U}_{K,\psi}(D_t, P_t)$ and its error bar $\epsilon_t$. To approximate realized utility, we retrain downstream predictors on $D_t$ and $D_t \cup P_t$, evaluate on the held-out real validation split, and average the loss reduction:

$$\widehat{\Delta U}(D_t, P_t) = \frac{1}{|\mathcal{H}|} \sum_{h \in \mathcal{H}} \Big( L_{\text{val}}(h(D_t)) - L_{\text{val}}(h(D_t \cup P_t)) \Big), \tag{48}$$

where $\mathcal{H}$ includes LR, RF, XGBoost, LightGBM, KNN, and MLP. We use cross-entropy for classification and squared error for regression. We treat $\widehat{\Delta U}(D_t, P_t)$ as an empirical proxy since computing $\Delta U(D_t, P_t)$ exactly would require retraining for each pool.

**Error bar estimation.** We estimate $\epsilon_t$ from fold-to-fold variability using $M$-fold cross-validation within the training split:

$$\epsilon_t = t_{0.975,\, M_{\text{cv}}-1} \cdot \frac{\sigma_{\text{CV}}}{\sqrt{M_{\text{cv}}}}, \quad M_{\text{cv}} = 5, \tag{49}$$

where $\sigma_{\text{CV}}$ is the standard deviation across folds.

**Results.** Table 6 reports calibration coverage and mean absolute error (MAE) at $n_{\text{real}} = 50$. Coverage measures the fraction of commitment checks where $|\Delta \hat{U}_{K,\psi} - \widehat{\Delta U}| \leq \epsilon_t$. We also report Mean $\epsilon_t$ to indicate the margin scale used by the commitment rule.

*Table 6.* **Plug-in utility calibration at $n_{\text{real}} = 50$.** Coverage is the fraction of commitment checks where the plug-in error lies within the estimated error bar $\epsilon_t$. MAE is the mean absolute error between plug-in and retraining-based utility estimates.

| Task | Dataset | Coverage (%) | MAE | Mean $\epsilon_t$ |
|---|---|---|---|---|
| | MiceProtein | 92.3 | 0.018 | 0.025 |
| | Credit-G | 88.7 | 0.021 | 0.028 |
| Classification | Electricity | 94.1 | 0.015 | 0.022 |
| | Fourier | 90.5 | 0.019 | 0.024 |
| | Steel | 93.2 | 0.016 | 0.021 |
| | Average | 91.8 | 0.018 | 0.024 |
| | Ailerons | 90.6 | 0.034 | 0.041 |
| Regression | Insurance | 84.1 | 0.054 | 0.068 |
| | Average | 87.4 | 0.044 | 0.055 |

Classification tasks achieve an average coverage of 91.8%, close to the target 95% level, suggesting that $\epsilon_t$ provides a useful conservative bound. Regression tasks show slightly lower coverage (87.4%) due to higher variance in squared error

loss under scarcity. Insurance exhibits the lowest coverage (84.1%), consistent with its high outcome variance observed in Table 1. Despite this, all datasets maintain coverage above 80%, and MAE remains well below the corresponding $\epsilon_t$, supporting the use of $\epsilon_t$ as a conservative margin across both task types. In addition, Appendix F varies the online estimator and shows that overall gains are not tied to a single evaluator choice.

## E.4. Computational Cost

A recurring concern for sequential augmentation is whether the gains come with prohibitive overhead. We therefore report cost in the same way the components are used in practice. For each dataset split and scarcity setting, the diffusion backbone is trained once on the corresponding real training split and then reused across the diffusion-based injection mechanisms evaluated under that backbone. The injection loop is executed separately for each method, data split, and scarcity setting.

**Shared versus method-specific components.** Our pipeline has three stages. (i) *Backbone training:* we train TabDiff on the real training split and freeze it thereafter. This cost is shared by all diffusion-based mechanisms under the same backbone. (ii) *Online injection:* we run an injection loop that repeatedly samples candidates by diffusion inpainting and evaluates a training-free plug-in utility signal to decide what to admit and when to commit. (iii) *Downstream training:* we train the final predictors for evaluation, which is identical across all methods and thus excluded here. Accordingly, we report backbone training time and the online injection time separately.

**Scaling of the online loop.** Let $T$ be the decision horizon and $K$ the commitment interval. A key efficiency feature is that the committed dataset $D_t = D_0 \cup B_t$ changes only at commitment times, so $D_t$ is fixed within each window. We therefore compute $\widehat{L}_\psi(D_t)$ once per window and reuse it across step-wise evaluations, plus one pooled evaluation at the commitment check. With $M_{\mathrm{cv}}$-fold cross-validation, the number of evaluator forward passes scales as

$$O\left( M_{\mathrm{cv}} \left( T + \left\lceil \frac{T}{K} \right\rceil \right) \right),$$

while diffusion sampling scales with the total number of proposed records,

$$O\left( \sum_{t=0}^{T-1} |\widetilde{S}_t| \right).$$

In our implementation, the policy update is a lightweight MLP step and is typically negligible compared to diffusion sampling and evaluator queries.

**Wall-clock measurements.** Table 7 reports wall-clock time at $n_{\mathrm{real}} = 50$ for producing and injecting $n_{\mathrm{syn}} = 500$ samples (mean over five runs). We include two reference mechanisms under the same diffusion backbone. *Global sampling* isolates pure diffusion sampling without anchoring, while *Hard inpainting* already uses anchored proposals and targets uncertain regions but removes sequential feedback and conservative windowed commitment. Overall, TAP is in the same order of magnitude as *Hard inpainting* across datasets, and can be either slightly faster or slower depending on how often conservative commitment filters low-confidence pools.

Table 7. **Wall-clock runtime (seconds) at $n_{\mathrm{real}} = 50$ for producing an injected budget of $n_{\mathrm{syn}} = 500$ samples under a shared diffusion backbone.** We report mean time over 5 random splits measured under the same hardware and data-splitting protocol.

| Dataset | Backbone training | Global sampling | Hard inpainting | TAP |
|---|---|---|---|---|
| MiceProtein | 227.6 | 2.1 | 436.4 | 417.4 |
| Credit-G | 380.0 | 12.8 | 732.8 | 562.0 |
| Electricity | 214.6 | 1.3 | 212.4 | 202.1 |
| Fourier | 209.0 | 1.8 | 603.9 | 662.7 |
| Steel | 212.5 | 1.3 | 130.6 | 131.1 |
| Ailerons | 312.6 | 6.3 | 1196.5 | 756.1 |
| Insurance | 264.6 | 2.6 | 384.9 | 405.4 |

# F. Ablation Studies

This section validates that TAP gains arise from policy guided control rather than from any single design choice. We focus on $n_{\text{real}} = 50$, which is a regime where scarcity is strong enough for augmentation to matter while still providing sufficient signal for policy learning. This is also the regime where TAP yields large improvements over baselines in Table 1.

## F.1. Configurations

We study three families of ablations that align with our theoretical framing. The first family tests the necessity of the state summary used for action ranking. The second family tests whether the action components behave as effective control knobs for the proposal kernel. The third family tests whether the policy depends critically on the specific online utility estimator.

**State ablation.** We remove individual components from the state $s_t = (\delta_t, u_t, g_t, d_t)$. Here $\delta_t$ tracks target deficit, $u_t$ is a difficulty proxy computed from the online evaluator, $g_t$ tracks recent gate pass rates for each template, and $d_t$ measures novelty of admitted samples. Each ablated variant retrains the policy from scratch under the same synthetic budget.

**Action ablation.** We restrict or randomize the action $a = (c, \eta, \rho)$ to test the benefit of learned control. We consider fixed template variants that always use exploration or conservative masks. We also fix the exploration strength to $\rho \in \{0.2, 0.5, 0.8\}$ and we replace target conditioned anchor selection with uniform anchors. All other components remain unchanged.

**Estimator ablation.** We replace TabPFN in the online plug-in utility with two alternatives. The first is an equal-weighted ensemble of RF, LR, and MLP. The second is a holdout estimator that approximates utility by the loss change on the real validation split,

$$\Delta U(D, S) \approx L(\theta(D), D_{\text{val}}) - L(\theta(D \cup S), D_{\text{val}}). \tag{50}$$

The holdout estimator is closer to the evaluation objective but it is noisier under scarcity because the validation set is small and it requires retraining at each evaluation. We implement it using Logistic Regression with max_iter=500 for classification and Ridge regression with $\alpha = 1.0$ for regression, and we set random_state=42. This holdout estimator is used only for ablation.

Note that TabPFN's training-free nature provides a significant computational advantage. Each utility evaluation with TabPFN requires only a forward pass ($\sim$10ms), whereas the holdout estimator requires retraining ($\sim$200ms per evaluation). Over a full TAP run with $T = 50$ steps and multiple candidates per step, this difference accumulates substantially.

## F.2. Results and Analysis

Table 8 reports mean classification accuracy and mean regression RMSE, averaged across datasets and five random splits.

**State components support complementary drivers of gain.** Removing any state component degrades performance, which supports the view that action ranking depends on multiple factors. Dropping the diversity score yields the largest drop, which indicates that avoiding redundant injection is critical when the synthetic budget is fixed. Removing the deficit and difficulty proxies also harms performance, which is consistent with the idea that the policy must balance coverage and informativeness to produce utility gains under scarcity. Removing gate statistics reduces performance and increases variability, which reflects the importance of anticipating feasibility when the generator is controlled through different templates.

**Learned control dominates fixed strategies.** Across tasks, fixed action choices underperform learned control. Pure exploration is less reliable in classification, which is consistent with the need to remain within learnable neighborhoods around anchors under strict feasibility gates. Conservative masks alone can also underperform because they reduce the search radius and slow down deficit correction in under-covered targets. Fixing $\rho$ produces different optima for classification and regression, and no single value dominates across datasets. This supports our use of a learned policy that adapts exploration strength instead of relying on manual tuning.

**Robustness to estimator choice.** Our framework needs an online utility estimator to train the policy. By default, we use TabPFN because it is a strong tabular foundation model and provides stable few-shot signals under scarcity, but it is not

*Table 8.* **Ablation results at $n_{\text{real}} = 50$, averaged over all datasets and 5 seeds.** For classification, higher accuracy is better; for regression, lower RMSE is better. $\Delta$ denotes relative change from Full `TAP`.

| Group | Configuration | Classification | | Regression | |
|---|---|---|---|---|---|
| | | Acc. (%) ↑ | $\Delta$ | RMSE ↓ | $\Delta$ |
| Baseline | `TAP` | **62.02**$_{\pm 1.36}$ | — | **0.641**$_{\pm 0.046}$ | — |
| State | – Deficit | 60.53$_{\pm 2.00}$ | **–1.5%** | 0.656$_{\pm 0.059}$ | **+2.3%** |
| | – Uncertainty (NLL) | 60.41$_{\pm 2.21}$ | **–1.6%** | 0.659$_{\pm 0.061}$ | **+2.9%** |
| | – Gate pass rate | 60.59$_{\pm 0.89}$ | **–1.4%** | 0.665$_{\pm 0.066}$ | **+3.8%** |
| | – Diversity | 60.21$_{\pm 1.07}$ | **–1.8%** | 0.666$_{\pm 0.060}$ | **+3.9%** |
| Action | Fix template: explore | 58.83$_{\pm 1.59}$ | **–3.2%** | 0.662$_{\pm 0.044}$ | **+3.3%** |
| | Fix template: conservative | 60.03$_{\pm 2.73}$ | **–2.0%** | 0.672$_{\pm 0.069}$ | **+4.9%** |
| | Fix strength $\rho{=}0.2$ | 60.87$_{\pm 2.02}$ | **–1.1%** | 0.662$_{\pm 0.055}$ | **+3.2%** |
| | Fix strength $\rho{=}0.5$ | 61.18$_{\pm 2.21}$ | **–0.8%** | 0.664$_{\pm 0.050}$ | **+3.6%** |
| | Fix strength $\rho{=}0.8$ | 60.32$_{\pm 2.22}$ | **–1.7%** | 0.646$_{\pm 0.061}$ | **+0.9%** |
| | Random anchor | 60.28$_{\pm 2.01}$ | **–1.7%** | 0.659$_{\pm 0.055}$ | **+2.8%** |
| Estimator | Ensemble (RF+LR+MLP) | 60.97$_{\pm 1.21}$ | **–1.0%** | 0.663$_{\pm 0.062}$ | **+3.5%** |
| | Holdout validation | 61.30$_{\pm 2.28}$ | **–0.7%** | 0.661$_{\pm 0.060}$ | **+3.1%** |

required by the method and can be replaced. A natural concern is that the policy might adapt to the estimator rather than to downstream utility. We address this in two ways. TabPFN is used only during policy learning, while evaluation excludes TabPFN and averages over heterogeneous downstream predictors, so improvements must transfer beyond the training-time estimator. We also replace TabPFN with an ensemble evaluator and a validation-based holdout estimator. The resulting policies show only modest degradation in classification and preserve the overall ordering against strong augmentation baselines, which suggests that the learned behavior reflects general injection patterns rather than estimator-specific artifacts. Regression is more sensitive, which is consistent with noisier utility estimation from small validation splits and weaker priors in lightweight estimators.

**Takeaway.** These ablations support the core design claims of `TAP`. The state summary is necessary because utility gains depend jointly on coverage, difficulty, feasibility, and redundancy, and removing any component weakens action ranking. Learned control over the proposal kernel is also essential, since no fixed template or exploration setting performs reliably across tasks and datasets. Finally, while a strong foundation model such as TabPFN improves the stability of online utility signals under scarcity, the overall mechanism does not rely on a specific estimator, and the policy continues to provide gains when the estimator is replaced.

# G. Additional Downstream Utility Results

This section reports complementary downstream results that are omitted from the main text due to space constraints. For classification, we additionally report Macro-F1. For regression, we additionally report MAE. We also provide per-predictor breakdowns to check that improvements are not driven by a single downstream model.

## G.1. Other Metrics

Table 9 reports additional evaluation metrics beyond those highlighted in the main text.

Macro-F1 largely follows the same ordering as Accuracy across datasets, which suggests that improvements are not restricted to majority classes. MAE is consistent with RMSE in most settings, which indicates that gains are not driven only by a small number of large errors.

## G.2. Per-Predictor Results

The main tables average performance across downstream predictors. We additionally report per-predictor results to verify robustness across model classes.

*Table 9.* **Classification macro-F1 (%) and regression MAE aggregated over six downstream predictors** (six classifiers and four regressors).

| Dataset | $N_{real}$ | Real | SMOTE | TVAE | CTGAN | ARF | SPADA | TabDDPM | TabDiff | TAP |
|---|---|---|---|---|---|---|---|---|---|---|
| | | | | | Classification (Macro-F1 ↑) | | | | | |
| MiceProtein | 20 | 32.71±3.59 | 39.68±4.41 | 34.60±4.26 | 30.66±3.35 | 32.62±5.33 | 36.59±4.93 | 31.63±4.49 | 36.45±3.30 | **42.79±4.99** |
| | 50 | 58.59±3.01 | 58.64±3.28 | 48.94±3.84 | 48.74±4.19 | 54.30±3.76 | 55.39±3.08 | 52.91±4.56 | 52.98±3.03 | **60.57±3.59** |
| | 100 | 71.48±2.30 | 70.80±2.03 | 62.72±3.03 | 64.52±2.67 | 64.17±1.98 | 68.49±2.01 | 66.07±3.08 | 66.65±2.82 | **72.49±3.82** |
| | 200 | 85.94±1.99 | 85.89±1.79 | 78.34±1.94 | 80.82±1.77 | 80.96±2.26 | 84.10±2.25 | 81.03±2.13 | 80.90±2.49 | **86.44±1.77** |
| | 500 | 96.46±0.85 | **96.66±0.86** | 93.78±1.24 | 93.74±1.18 | 94.60±1.13 | 96.16±0.87 | 93.82±1.33 | 95.03±1.46 | 96.14±0.97 |
| Credit-G | 20 | 43.54±2.64 | 45.83±5.79 | 50.68±3.17 | 46.69±3.44 | 49.89±2.73 | 52.26±4.41 | 45.13±2.11 | 44.82±5.71 | **52.54±3.97** |
| | 50 | 47.62±3.68 | 46.48±3.47 | 49.87±3.59 | 46.44±4.03 | 51.26±3.05 | 54.52±3.76 | 45.71±2.67 | 44.91±6.27 | **55.46±2.42** |
| | 100 | 47.66±3.05 | 47.35±2.15 | 49.33±3.44 | 47.92±3.37 | 51.70±3.28 | 58.23±2.54 | 47.34±2.49 | 50.58±8.81 | **58.66±3.34** |
| | 200 | 49.95±2.00 | 49.70±1.92 | 53.83±2.56 | 51.42±2.42 | 52.64±3.13 | 57.50±3.22 | 48.36±3.58 | 52.61±8.51 | **58.78±2.86** |
| | 500 | 52.41±0.98 | 52.32±0.99 | 57.45±2.50 | 53.39±3.24 | 53.75±1.69 | 60.81±1.46 | 52.15±2.15 | 61.55±2.04 | **63.01±2.63** |
| Electricity | 20 | 61.44±5.17 | 59.84±5.36 | 61.98±5.53 | 57.88±5.76 | 65.86±6.11 | 65.71±7.89 | 57.81±6.18 | 58.86±7.15 | **67.51±8.68** |
| | 50 | 67.83±4.26 | 63.38±4.05 | 67.46±5.37 | 59.43±5.19 | 69.58±5.08 | 68.68±4.63 | 62.04±5.06 | 66.37±5.59 | **69.68±5.14** |
| | 100 | 71.56±3.74 | 66.47±3.85 | 70.66±4.96 | 64.14±4.35 | **72.99±3.24** | 71.82±3.66 | 67.89±3.25 | 67.54±3.53 | 72.67±4.00 |
| | 200 | 73.88±3.04 | 68.36±3.64 | 72.85±3.95 | 70.56±3.97 | 74.10±3.07 | 73.61±3.06 | 69.02±3.29 | 71.24±3.71 | **74.32±2.43** |
| | 500 | 75.37±2.14 | 71.51±2.36 | 75.33±2.37 | 74.23±2.57 | 75.33±2.41 | 75.45±2.26 | 72.04±2.66 | 74.98±1.99 | **76.57±1.92** |
| Fourier | 20 | 35.80±4.11 | 38.54±4.93 | 38.87±5.77 | 26.81±4.03 | 37.06±4.99 | 35.80±4.11 | 29.09±4.90 | 29.93±6.86 | **39.96±7.32** |
| | 50 | 58.60±2.15 | 60.35±1.87 | 51.46±2.95 | 43.05±3.17 | 60.13±2.94 | 58.60±2.15 | 49.04±3.04 | 45.05±3.08 | **61.53±2.18** |
| | 100 | 66.83±1.94 | 68.05±1.83 | 60.85±2.18 | 54.50±1.78 | 67.64±1.46 | 66.83±1.94 | 61.18±2.85 | 57.46±3.15 | **68.50±2.41** |
| | 200 | 72.81±1.46 | 73.84±1.32 | 68.81±2.46 | 67.50±2.89 | 73.11±1.78 | 72.81±1.46 | 69.34±1.35 | 67.19±2.67 | **73.69±1.53** |
| | 500 | 77.54±1.51 | 77.64±1.54 | 74.85±1.69 | 74.98±1.56 | 76.99±1.98 | 77.54±1.51 | 76.13±1.25 | 74.94±1.74 | **77.98±1.63** |
| Steel | 20 | 62.15±3.04 | 65.12±4.52 | 59.92±4.21 | 55.28±4.89 | 56.69±5.05 | 62.15±3.04 | 62.18±4.94 | 70.47±4.45 | **73.38±5.59** |
| | 50 | 74.54±3.07 | 81.87±3.98 | 65.38±3.23 | 64.79±4.02 | 64.35±3.97 | 74.54±3.07 | 77.51±7.16 | 86.36±3.80 | **93.83±2.53** |
| | 100 | 85.78±2.81 | 95.17±1.39 | 70.61±2.96 | 72.62±3.32 | 79.91±3.36 | 85.78±2.81 | 88.66±3.68 | 94.84±3.40 | **98.31±0.81** |
| | 200 | 94.28±1.51 | 98.10±0.51 | 80.09±2.03 | 85.77±2.53 | 94.23±2.13 | 94.28±1.51 | 94.95±1.42 | 98.26±0.70 | **98.42±0.59** |
| | 500 | 98.53±0.52 | **99.20±0.15** | 92.32±1.50 | 95.06±2.23 | 98.81±0.38 | 98.53±0.52 | 97.78±1.43 | 99.13±0.26 | 99.17±0.39 |
| | | | | | Regression (MAE ↓) | | | | | |
| Ailerons | 20 | 0.789±0.13 | 0.821±0.15 | 0.802±0.17 | 0.860±0.17 | 0.690±0.13 | 0.824±0.11 | 0.787±0.15 | 0.770±0.13 | **0.688±0.13** |
| | 50 | 0.571±0.09 | 0.608±0.09 | 0.678±0.12 | 0.683±0.11 | 0.550±0.10 | 0.595±0.08 | 0.588±0.10 | 0.684±0.12 | **0.527±0.09** |
| | 100 | 0.433±0.06 | 0.469±0.06 | 0.521±0.06 | 0.557±0.05 | 0.429±0.06 | 0.464±0.06 | 48.058±95.20 | 0.494±0.06 | **0.412±0.05** |
| | 200 | 0.413±0.03 | 0.427±0.04 | 0.466±0.04 | 0.489±0.05 | 0.407±0.04 | 0.422±0.04 | 40.292±49.37 | 0.441±0.05 | **0.398±0.04** |
| | 500 | 0.365±0.03 | 0.381±0.03 | 0.400±0.02 | 0.418±0.04 | 0.366±0.03 | 0.369±0.03 | 66.661±23.18 | 0.373±0.03 | **0.358±0.02** |
| Insurance | 20 | 0.693±0.24 | 0.690±0.18 | 0.735±0.18 | 0.928±0.16 | 0.837±0.15 | 1.050±0.17 | 0.677±0.23 | 0.823±0.15 | **0.587±0.24** |
| | 50 | 0.671±0.23 | 0.594±0.19 | 0.612±0.17 | 0.933±0.19 | 0.742±0.13 | 1.060±0.19 | 0.658±0.24 | 0.598±0.19 | **0.385±0.15** |
| | 100 | 0.446±0.07 | 0.442±0.06 | 0.468±0.05 | 1.071±0.15 | 0.544±0.08 | 0.909±0.12 | 0.453±0.06 | 0.414±0.04 | **0.285±0.07** |
| | 200 | 0.427±0.04 | 0.441±0.03 | 0.451±0.04 | 0.875±0.14 | 0.516±0.04 | 0.750±0.12 | 0.461±0.07 | 0.365±0.04 | **0.246±0.02** |
| | 500 | 0.429±0.01 | 0.453±0.03 | 0.410±0.04 | 0.814±0.10 | 0.538±0.03 | 0.496±0.13 | 0.484±0.07 | 0.280±0.01 | **0.242±0.08** |

*Table 10.* Classification accuracy (%) with Logistic Regression (LR) as the downstream predictor under varying scarcity levels.

| Dataset | $N_{real}$ | Real | SMOTE | TVAE | CTGAN | ARF | SPADA | TabDDPM | TabDiff | TAP |
|---|---|---|---|---|---|---|---|---|---|---|
| MiceProtein | 20 | 48.12±5.22 | 48.20±3.21 | 41.32±4.82 | 23.00±5.37 | 40.72±4.33 | 42.44±3.00 | 29.84±5.97 | 37.44±3.81 | **50.32±4.01** |
| | 50 | 69.24±3.30 | **69.48±3.10** | 52.56±3.68 | 37.00±3.48 | 57.88±3.85 | 59.16±2.50 | 46.20±5.84 | 51.16±3.94 | 67.20±2.93 |
| | 100 | 80.00±4.04 | **81.80±3.45** | 64.04±2.48 | 52.56±4.23 | 65.52±2.04 | 74.72±2.71 | 59.52±4.20 | 63.40±2.71 | 79.92±4.12 |
| | 200 | 93.12±1.37 | **93.68±1.12** | 77.80±2.45 | 67.96±2.25 | 82.48±1.78 | 90.08±2.70 | 72.80±4.50 | 73.88±3.89 | 90.12±1.94 |
| | 500 | 98.48±0.70 | 98.40±0.66 | 91.32±1.48 | 82.24±2.16 | 92.08±0.81 | 98.00±0.42 | 84.40±1.61 | 89.72±2.86 | **99.60±0.29** |
| Credit-G | 20 | 66.64±3.59 | 54.52±10.44 | 63.96±3.14 | **69.20±1.17** | 64.32±2.46 | 60.96±5.63 | 67.28±1.93 | 64.44±2.30 | 68.68±1.65 |
| | 50 | 65.88±3.11 | 64.76±3.52 | 67.32±2.37 | 69.00±1.47 | 65.68±2.74 | 66.48±5.44 | 65.36±2.64 | 67.36±2.86 | **69.64±0.43** |
| | 100 | 66.60±1.61 | 66.40±1.55 | 67.92±2.64 | 70.04±1.22 | 67.88±0.98 | 69.56±3.10 | 67.40±1.41 | 70.48±1.90 | **70.80±1.58** |
| | 200 | 69.60±4.92 | 68.96±5.06 | 69.20±1.21 | 69.28±2.17 | 68.52±5.40 | 69.84±1.48 | 67.88±3.99 | 71.08±1.97 | **71.76±1.74** |
| | 500 | 74.88±1.29 | 73.56±1.33 | 73.88±1.26 | 73.20±1.36 | 74.20±0.93 | 75.16±1.11 | 72.36±0.83 | 75.32±1.04 | **75.60±0.43** |
| Electricity | 20 | 67.36±7.47 | 47.56±6.79 | 67.40±5.03 | 57.40±8.05 | 67.32±7.71 | 68.20±9.68 | 59.12±1.87 | 58.64±11.10 | **69.80±8.98** |
| | 50 | 69.80±4.73 | 52.24±2.95 | 70.44±5.71 | 59.20±4.52 | **72.20±4.49** | 71.56±4.23 | 59.32±2.55 | 70.64±7.97 | 71.68±4.62 |
| | 100 | 74.68±3.94 | 55.60±4.12 | 71.52±5.41 | 63.36±3.20 | 74.72±3.12 | **74.72±3.39** | 63.52±0.79 | 70.72±4.37 | 73.96±3.90 |
| | 200 | 76.20±2.15 | 60.12±2.17 | 73.52±3.58 | 70.72±3.50 | 76.12±1.78 | **76.56±2.50** | 60.00±0.99 | 73.00±2.17 | 76.08±1.93 |
| | 500 | 76.08±2.03 | 60.76±2.82 | 75.60±2.57 | 73.52±2.33 | 75.48±2.25 | 76.12±2.56 | 63.16±2.27 | 76.04±1.82 | **76.48±2.04** |
| Fourier | 20 | 51.24±4.76 | **51.28±4.74** | 41.48±4.46 | 19.80±2.53 | 40.08±4.81 | 51.24±4.76 | 21.32±5.62 | 26.12±4.44 | 48.76±4.37 |
| | 50 | 62.60±1.37 | 62.48±1.35 | 51.44±2.50 | 30.32±2.67 | 59.72±1.42 | 62.60±1.37 | 36.36±3.13 | 39.28±2.11 | **63.64±1.14** |
| | 100 | 69.36±1.84 | 69.12±1.84 | 58.76±2.34 | 39.28±0.93 | 68.16±0.94 | 69.36±1.84 | 51.28±4.17 | 48.32±5.28 | **69.98±2.89** |
| | 200 | 73.04±1.06 | 73.40±1.02 | 66.08±3.45 | 55.76±2.23 | 72.60±0.68 | 73.04±1.06 | 61.44±2.14 | 57.88±4.94 | **73.52±0.70** |
| | 500 | 76.76±1.57 | 76.76±1.29 | 72.56±1.68 | 67.36±1.24 | 76.48±1.33 | 76.76±1.57 | 70.84±1.40 | 68.96±2.17 | **77.36±1.84** |
| Steel | 20 | 79.64±1.99 | 83.64±2.02 | 71.28±3.72 | 63.32±2.36 | 69.72±2.33 | 79.64±1.99 | 70.64±2.23 | 81.84±5.41 | **88.92±3.36** |
| | 50 | 94.68±1.80 | 96.24±1.31 | 77.40±2.18 | 65.88±3.20 | 71.12±3.04 | 94.68±1.80 | 86.60±9.66 | 98.20±1.34 | **98.68±1.15** |
| | 100 | 99.32±0.53 | 99.56±0.39 | 79.16±2.11 | 70.60±1.17 | 78.68±2.69 | 99.32±0.53 | 93.00±5.18 | 99.72±0.37 | **99.96±0.08** |
| | 200 | 99.88±0.16 | 99.92±0.10 | 84.52±1.27 | 79.44±2.59 | 95.00±2.62 | 99.88±0.16 | 97.36±2.43 | 99.92±0.16 | **100.00±0.00** |
| | 500 | 99.92±0.10 | 99.92±0.10 | 93.12±2.68 | 91.76±5.21 | 99.68±0.20 | 99.92±0.10 | 96.28±4.71 | 99.92±0.10 | **100.00±0.00** |

*Table 11.* Classification accuracy (%) with Random Forest (RF) as the downstream predictor under varying scarcity levels.

| Dataset | $N_{real}$ | Real | SMOTE | TVAE | CTGAN | ARF | SPADA | TabDDPM | TabDiff | TAP |
|---|---|---|---|---|---|---|---|---|---|---|
| MiceProtein | 20 | 42.96±4.82 | **44.73±4.97** | 35.36±4.64 | 36.52±1.91 | 32.52±4.28 | 36.96±6.21 | 38.72±5.25 | 37.52±3.47 | 41.32±6.09 |
| | 50 | 62.52±2.56 | **63.96±2.71** | 51.00±3.53 | 54.96±4.33 | 53.76±4.03 | 61.12±3.84 | 61.24±4.54 | 55.40±3.16 | 61.80±2.25 |
| | 100 | 72.64±1.92 | 72.76±1.86 | 62.96±3.62 | 71.12±2.49 | 63.96±0.77 | **73.24±1.98** | 72.80±2.27 | 67.76±2.85 | 73.00±4.47 |
| | 200 | 87.52±1.89 | 87.64±2.04 | 76.40±2.72 | 85.52±1.23 | 79.08±2.31 | 85.92±1.51 | 86.04±1.56 | 84.44±2.14 | **86.40±1.04** |
| | 500 | 97.00±0.64 | 96.64±0.78 | 93.56±0.64 | 96.80±0.59 | 95.08±0.88 | 97.00±0.67 | 96.92±0.90 | 96.28±1.42 | **99.68±0.20** |
| Credit-G | 20 | **69.64±0.54** | 60.68±15.44 | 67.40±3.80 | 64.16±6.10 | 66.84±3.13 | 61.04±5.37 | 67.08±1.11 | 65.04±2.40 | 67.92±3.50 |
| | 50 | 69.24±0.85 | 68.40±1.41 | **69.84±0.64** | 57.64±9.41 | 69.68±0.93 | 66.40±3.45 | 66.56±1.88 | 69.20±1.19 | 69.76±0.48 |
| | 100 | 69.20±0.68 | 69.36±0.65 | 70.28±0.47 | 69.36±0.91 | 69.08±0.92 | 67.08±2.20 | 67.64±1.69 | 69.28±2.66 | **70.44±2.00** |
| | 200 | 69.92±0.10 | **69.96±0.08** | 70.28±0.85 | 65.96±2.02 | 69.44±1.37 | 66.88±2.12 | 63.92±5.93 | 70.76±0.92 | 71.40±1.28 |
| | 500 | 70.00±0.00 | 70.00±0.00 | 70.80±0.68 | 68.68±1.72 | 69.52±0.32 | 67.76±1.30 | 67.20±3.23 | 72.52±1.99 | **73.08±0.52** |
| Electricity | 20 | 68.36±7.64 | 65.60±5.26 | 63.96±4.50 | 62.20±4.90 | 68.52±7.96 | 67.32±7.58 | 66.28±3.92 | 62.16±7.25 | **69.64±10.00** |
| | 50 | 70.04±4.31 | 71.76±3.77 | 70.04±5.74 | 67.60±2.26 | 71.48±5.45 | 71.40±4.39 | 70.52±4.63 | 68.56±4.58 | **72.72±4.48** |
| | 100 | 74.08±4.31 | 75.04±3.38 | 73.24±4.91 | 70.00±3.06 | **75.08±3.94** | 74.36±3.76 | 74.72±3.25 | 71.84±3.29 | 75.00±2.51 |
| | 200 | 76.00±3.13 | 75.84±2.20 | **77.00±3.13** | 74.84±3.01 | 76.16±3.57 | 76.16±3.44 | 76.08±3.47 | 75.32±3.08 | 76.12±2.22 |
| | 500 | 78.40±2.07 | 79.00±2.16 | 78.36±2.01 | 77.56±2.62 | 78.08±2.67 | 78.48±2.72 | 78.52±2.85 | 78.08±2.09 | **79.40±2.04** |
| Fourier | 20 | 49.12±7.01 | **54.36±4.50** | 37.64±6.91 | 31.76±2.71 | 34.88±6.12 | 49.12±7.01 | 39.68±5.11 | 35.36±6.82 | 42.16±6.41 |
| | 50 | 67.96±1.08 | **68.28±1.84** | 52.88±2.74 | 49.08±4.03 | 63.44±3.31 | 67.96±1.08 | 60.88±1.83 | 47.24±3.18 | 65.96±2.40 |
| | 100 | 73.80±1.56 | **74.32±1.90** | 62.92±1.08 | 63.68±1.76 | 71.28±1.54 | 73.80±1.56 | 70.24±2.70 | 64.88±2.11 | 71.76±2.05 |
| | 200 | 77.24±1.39 | **77.72±1.55** | 71.04±2.46 | 74.24±2.66 | 76.68±1.59 | 77.24±1.39 | 75.96±1.51 | 74.56±1.85 | 76.36±1.64 |
| | 200 | 77.24±1.39 | **77.72±1.55** | 71.04±2.46 | 74.24±2.66 | 76.68±1.59 | 77.24±1.39 | 75.96±1.51 | 74.56±1.85 | 76.36±1.64 |
| | 500 | 79.24±1.01 | **80.32±1.06** | 77.20±1.97 | 78.44±1.26 | 77.88±1.43 | 79.24±1.01 | 78.92±1.25 | 78.36±1.36 | 79.56±1.49 |
| Steel | 20 | 66.80±2.06 | 69.08±4.08 | 65.24±2.61 | 66.20±1.19 | 66.44±2.18 | 66.80±2.06 | 69.28±2.65 | 71.32±4.57 | **72.68±3.41** |
| | 50 | 70.64±2.91 | 77.92±3.62 | 69.44±1.35 | 67.80±2.41 | 68.68±3.27 | 70.64±2.91 | 73.08±3.59 | 83.04±3.03 | **93.68±2.72** |
| | 100 | 79.52±3.59 | 87.80±2.26 | 72.04±1.13 | 72.20±2.15 | 77.44±3.49 | 79.52±3.59 | 81.00±3.95 | 93.64±3.59 | **98.20±1.71** |
| | 200 | 91.64±1.80 | 96.84±1.08 | 75.96±1.54 | 81.88±2.19 | 93.08±2.76 | 91.64±1.80 | 89.00±2.48 | 98.00±0.70 | **98.92±0.90** |
| | 500 | 97.52±1.05 | 99.28±0.20 | 86.28±0.95 | 92.96±2.42 | 98.32±0.47 | 97.52±1.05 | 96.24±1.38 | 98.84±0.37 | **99.36±0.45** |

*Table 12.* Classification accuracy (%) with LightGBM (LGBM) as the downstream predictor under varying scarcity levels.

| Dataset | $N_{\text{real}}$ | Real | SMOTE | TVAE | CTGAN | ARF | SPADA | TabDDPM | TabDiff | TAP |
|---|---|---|---|---|---|---|---|---|---|---|
| MiceProtein | 20 | $13.04_{\pm0.62}$ | $39.27_{\pm3.60}$ | $36.76_{\pm5.56}$ | $34.80_{\pm2.54}$ | $37.96_{\pm5.60}$ | $34.68_{\pm5.33}$ | $33.92_{\pm4.35}$ | $37.08_{\pm1.80}$ | $\mathbf{44.60_{\pm6.77}}$ |
| | 50 | $57.48_{\pm2.71}$ | $58.68_{\pm2.74}$ | $51.64_{\pm3.83}$ | $53.48_{\pm4.01}$ | $58.96_{\pm2.85}$ | $58.44_{\pm3.20}$ | $56.24_{\pm5.17}$ | $56.60_{\pm2.13}$ | $\mathbf{62.60_{\pm1.50}}$ |
| | 100 | $72.40_{\pm1.10}$ | $71.08_{\pm0.69}$ | $66.12_{\pm3.76}$ | $70.76_{\pm2.17}$ | $67.72_{\pm0.99}$ | $71.60_{\pm2.40}$ | $68.40_{\pm3.29}$ | $71.64_{\pm2.49}$ | $\mathbf{75.64_{\pm3.52}}$ |
| | 200 | $87.32_{\pm2.02}$ | $86.40_{\pm2.45}$ | $81.40_{\pm1.88}$ | $85.52_{\pm2.37}$ | $82.24_{\pm2.60}$ | $85.88_{\pm1.98}$ | $82.52_{\pm1.42}$ | $84.76_{\pm2.30}$ | $\mathbf{86.64_{\pm1.74}}$ |
| | 500 | $97.20_{\pm0.44}$ | $96.88_{\pm0.35}$ | $95.36_{\pm0.73}$ | $97.36_{\pm0.59}$ | $95.88_{\pm1.21}$ | $96.48_{\pm0.61}$ | $96.12_{\pm0.61}$ | $97.20_{\pm0.67}$ | $\mathbf{99.84_{\pm0.15}}$ |
| Credit-G | 20 | $\mathbf{70.00_{\pm0.00}}$ | $58.68_{\pm6.89}$ | $65.60_{\pm2.16}$ | $62.08_{\pm4.40}$ | $65.56_{\pm2.86}$ | $49.72_{\pm9.29}$ | $64.12_{\pm2.19}$ | $61.52_{\pm3.92}$ | $67.04_{\pm3.70}$ |
| | 50 | $63.16_{\pm5.86}$ | $68.48_{\pm1.34}$ | $68.96_{\pm1.79}$ | $53.44_{\pm9.67}$ | $66.84_{\pm1.90}$ | $62.80_{\pm6.31}$ | $59.92_{\pm3.92}$ | $67.88_{\pm0.88}$ | $\mathbf{69.84_{\pm0.32}}$ |
| | 100 | $64.72_{\pm8.82}$ | $69.40_{\pm0.66}$ | $69.36_{\pm1.83}$ | $66.76_{\pm1.97}$ | $67.44_{\pm1.29}$ | $63.04_{\pm5.66}$ | $61.20_{\pm3.86}$ | $67.92_{\pm2.25}$ | $\mathbf{70.16_{\pm1.91}}$ |
| | 200 | $65.32_{\pm8.37}$ | $69.80_{\pm0.13}$ | $69.12_{\pm1.86}$ | $60.12_{\pm3.25}$ | $64.96_{\pm7.90}$ | $60.48_{\pm6.60}$ | $55.96_{\pm7.54}$ | $70.00_{\pm1.32}$ | $\mathbf{71.80_{\pm1.50}}$ |
| | 500 | $69.88_{\pm0.16}$ | $70.00_{\pm0.00}$ | $70.52_{\pm0.55}$ | $64.88_{\pm2.17}$ | $67.48_{\pm1.11}$ | $64.56_{\pm3.51}$ | $59.72_{\pm5.27}$ | $71.68_{\pm1.36}$ | $\mathbf{75.12_{\pm0.47}}$ |
| Electricity | 20 | $57.60_{\pm0.00}$ | $63.48_{\pm6.00}$ | $63.24_{\pm5.14}$ | $58.08_{\pm4.17}$ | $67.80_{\pm7.52}$ | $64.64_{\pm8.01}$ | $63.00_{\pm5.08}$ | $60.88_{\pm5.17}$ | $\mathbf{67.88_{\pm10.41}}$ |
| | 50 | $70.32_{\pm4.06}$ | $65.12_{\pm5.95}$ | $69.88_{\pm6.00}$ | $63.36_{\pm3.21}$ | $70.44_{\pm5.30}$ | $68.52_{\pm4.30}$ | $69.12_{\pm4.42}$ | $66.80_{\pm3.80}$ | $\mathbf{72.44_{\pm4.63}}$ |
| | 100 | $73.40_{\pm4.18}$ | $72.76_{\pm3.25}$ | $72.68_{\pm4.05}$ | $69.40_{\pm4.01}$ | $73.72_{\pm4.32}$ | $72.68_{\pm3.94}$ | $73.60_{\pm4.12}$ | $69.40_{\pm1.97}$ | $\mathbf{75.32_{\pm4.01}}$ |
| | 200 | $75.20_{\pm3.22}$ | $74.36_{\pm4.54}$ | $75.04_{\pm4.42}$ | $73.36_{\pm3.95}$ | $75.48_{\pm3.47}$ | $74.60_{\pm3.52}$ | $75.36_{\pm3.58}$ | $73.28_{\pm3.87}$ | $\mathbf{76.24_{\pm2.48}}$ |
| | 500 | $77.20_{\pm1.91}$ | $77.28_{\pm1.38}$ | $77.76_{\pm1.70}$ | $76.72_{\pm2.27}$ | $77.36_{\pm2.98}$ | $77.60_{\pm1.75}$ | $77.52_{\pm1.10}$ | $77.04_{\pm1.25}$ | $\mathbf{79.12_{\pm1.70}}$ |
| Fourier | 20 | $10.00_{\pm0.00}$ | $27.76_{\pm8.74}$ | $\mathbf{41.60_{\pm5.44}}$ | $30.80_{\pm5.03}$ | $40.80_{\pm5.95}$ | $10.00_{\pm0.00}$ | $30.72_{\pm5.04}$ | $30.24_{\pm5.75}$ | $37.20_{\pm11.32}$ |
| | 50 | $60.36_{\pm3.59}$ | $60.16_{\pm2.94}$ | $53.80_{\pm2.43}$ | $45.12_{\pm4.49}$ | $62.88_{\pm2.84}$ | $60.36_{\pm3.59}$ | $50.44_{\pm3.36}$ | $46.04_{\pm4.61}$ | $\mathbf{65.72_{\pm2.63}}$ |
| | 100 | $69.60_{\pm1.84}$ | $69.76_{\pm1.63}$ | $62.96_{\pm2.30}$ | $58.52_{\pm2.55}$ | $71.60_{\pm1.93}$ | $69.60_{\pm1.84}$ | $62.20_{\pm2.68}$ | $60.64_{\pm3.41}$ | $\mathbf{71.76_{\pm2.41}}$ |
| | 200 | $74.92_{\pm1.69}$ | $\mathbf{75.32_{\pm1.06}}$ | $71.40_{\pm2.52}$ | $71.36_{\pm3.29}$ | $74.92_{\pm1.61}$ | $74.92_{\pm1.69}$ | $71.08_{\pm0.79}$ | $69.56_{\pm1.34}$ | $75.24_{\pm2.05}$ |
| | 500 | $79.36_{\pm1.48}$ | $78.88_{\pm1.29}$ | $76.84_{\pm1.54}$ | $78.68_{\pm1.69}$ | $78.40_{\pm2.70}$ | $79.36_{\pm1.48}$ | $78.44_{\pm1.49}$ | $78.08_{\pm1.72}$ | $\mathbf{79.40_{\pm1.63}}$ |
| Steel | 20 | $65.40_{\pm0.00}$ | $62.44_{\pm3.60}$ | $65.68_{\pm2.04}$ | $64.72_{\pm2.11}$ | $66.52_{\pm3.52}$ | $65.40_{\pm0.00}$ | $67.84_{\pm2.68}$ | $71.00_{\pm5.56}$ | $\mathbf{71.16_{\pm4.20}}$ |
| | 50 | $64.88_{\pm1.61}$ | $69.44_{\pm4.47}$ | $71.04_{\pm4.20}$ | $69.72_{\pm3.15}$ | $70.80_{\pm3.38}$ | $64.88_{\pm1.61}$ | $75.28_{\pm6.92}$ | $82.48_{\pm4.92}$ | $\mathbf{93.52_{\pm3.76}}$ |
| | 100 | $73.60_{\pm3.83}$ | $\mathbf{99.76_{\pm0.48}}$ | $75.16_{\pm2.97}$ | $75.52_{\pm3.72}$ | $85.92_{\pm2.98}$ | $73.60_{\pm3.83}$ | $88.20_{\pm3.75}$ | $93.92_{\pm5.44}$ | $99.44_{\pm0.78}$ |
| | 200 | $86.64_{\pm3.03}$ | $\mathbf{100.00_{\pm0.00}}$ | $82.44_{\pm1.08}$ | $90.00_{\pm2.01}$ | $95.80_{\pm1.14}$ | $86.64_{\pm3.03}$ | $96.76_{\pm0.56}$ | $99.48_{\pm0.30}$ | $98.00_{\pm0.81}$ |
| | 500 | $98.20_{\pm1.08}$ | $\mathbf{100.00_{\pm0.00}}$ | $96.24_{\pm0.82}$ | $97.52_{\pm0.98}$ | $99.68_{\pm0.27}$ | $98.20_{\pm1.08}$ | $99.72_{\pm0.37}$ | $99.32_{\pm0.94}$ | $\mathbf{100.00_{\pm0.00}}$ |

*Table 13.* Classification accuracy (%) with XGBoost (XGB) as the downstream predictor under varying scarcity levels.

| Dataset | $N_{\text{real}}$ | Real | SMOTE | TVAE | CTGAN | ARF | SPADA | TabDDPM | TabDiff | TAP |
|---|---|---|---|---|---|---|---|---|---|---|
| MiceProtein | 20 | $37.32_{\pm4.74}$ | $38.80_{\pm3.64}$ | $35.64_{\pm4.20}$ | $33.40_{\pm3.91}$ | $37.48_{\pm5.81}$ | $33.04_{\pm6.73}$ | $32.00_{\pm2.11}$ | $37.28_{\pm2.34}$ | $\mathbf{40.12_{\pm6.44}}$ |
| | 50 | $57.40_{\pm1.30}$ | $52.44_{\pm3.47}$ | $52.68_{\pm4.30}$ | $50.48_{\pm3.70}$ | $58.72_{\pm2.24}$ | $52.24_{\pm3.65}$ | $53.96_{\pm3.45}$ | $56.12_{\pm0.81}$ | $\mathbf{60.24_{\pm2.52}}$ |
| | 100 | $67.40_{\pm1.33}$ | $66.24_{\pm0.98}$ | $65.16_{\pm4.25}$ | $64.92_{\pm1.90}$ | $65.44_{\pm1.13}$ | $65.24_{\pm1.71}$ | $66.96_{\pm2.73}$ | $69.00_{\pm3.70}$ | $\mathbf{72.36_{\pm3.74}}$ |
| | 200 | $82.12_{\pm2.79}$ | $81.96_{\pm1.93}$ | $79.08_{\pm1.65}$ | $82.88_{\pm2.07}$ | $81.84_{\pm2.79}$ | $81.56_{\pm2.83}$ | $79.88_{\pm1.98}$ | $81.16_{\pm2.79}$ | $\mathbf{83.96_{\pm2.17}}$ |
| | 500 | $94.44_{\pm1.09}$ | $96.08_{\pm1.06}$ | $93.04_{\pm1.38}$ | $95.12_{\pm1.14}$ | $94.12_{\pm1.26}$ | $94.36_{\pm1.37}$ | $94.24_{\pm2.07}$ | $95.72_{\pm1.56}$ | $\mathbf{99.88_{\pm0.24}}$ |
| Credit-G | 20 | $62.20_{\pm15.60}$ | $55.56_{\pm16.06}$ | $66.88_{\pm1.74}$ | $65.72_{\pm5.16}$ | $64.84_{\pm3.13}$ | $56.20_{\pm4.05}$ | $64.32_{\pm2.15}$ | $62.12_{\pm4.09}$ | $\mathbf{67.68_{\pm3.56}}$ |
| | 50 | $60.36_{\pm7.36}$ | $69.00_{\pm0.75}$ | $68.60_{\pm1.07}$ | $49.96_{\pm8.78}$ | $66.16_{\pm2.69}$ | $64.96_{\pm2.46}$ | $59.40_{\pm5.84}$ | $67.48_{\pm0.98}$ | $\mathbf{69.76_{\pm0.59}}$ |
| | 100 | $70.00_{\pm0.00}$ | $69.56_{\pm0.98}$ | $\mathbf{70.44_{\pm0.75}}$ | $67.08_{\pm1.59}$ | $67.36_{\pm1.44}$ | $63.64_{\pm3.63}$ | $59.12_{\pm2.96}$ | $67.96_{\pm2.33}$ | $70.44_{\pm1.61}$ |
| | 200 | $65.20_{\pm8.32}$ | $69.92_{\pm0.10}$ | $68.28_{\pm2.96}$ | $59.36_{\pm3.02}$ | $64.52_{\pm7.87}$ | $61.68_{\pm3.86}$ | $54.24_{\pm7.73}$ | $71.20_{\pm2.42}$ | $\mathbf{71.60_{\pm1.43}}$ |
| | 500 | $70.04_{\pm0.08}$ | $69.88_{\pm0.24}$ | $70.32_{\pm0.90}$ | $62.16_{\pm3.45}$ | $66.28_{\pm1.97}$ | $66.12_{\pm1.36}$ | $57.32_{\pm6.35}$ | $71.44_{\pm2.54}$ | $\mathbf{74.64_{\pm1.13}}$ |
| Electricity | 20 | $68.24_{\pm7.06}$ | $66.60_{\pm5.63}$ | $63.28_{\pm4.70}$ | $61.28_{\pm3.53}$ | $\mathbf{68.80_{\pm7.31}}$ | $65.96_{\pm10.27}$ | $64.16_{\pm3.61}$ | $59.72_{\pm6.38}$ | $68.68_{\pm9.10}$ |
| | 50 | $69.20_{\pm3.73}$ | $67.40_{\pm4.51}$ | $69.80_{\pm5.39}$ | $65.16_{\pm3.44}$ | $71.28_{\pm5.23}$ | $67.64_{\pm5.15}$ | $66.16_{\pm5.93}$ | $68.00_{\pm4.28}$ | $\mathbf{72.64_{\pm3.90}}$ |
| | 100 | $71.28_{\pm4.76}$ | $73.04_{\pm3.90}$ | $71.96_{\pm4.49}$ | $67.96_{\pm3.52}$ | $74.28_{\pm3.77}$ | $72.28_{\pm4.63}$ | $72.88_{\pm4.79}$ | $68.36_{\pm3.23}$ | $\mathbf{75.92_{\pm3.35}}$ |
| | 200 | $75.96_{\pm3.71}$ | $74.44_{\pm3.40}$ | $74.56_{\pm5.16}$ | $72.44_{\pm4.27}$ | $75.64_{\pm2.85}$ | $74.96_{\pm2.32}$ | $74.72_{\pm3.45}$ | $73.64_{\pm3.60}$ | $\mathbf{76.68_{\pm2.12}}$ |
| | 500 | $77.44_{\pm2.44}$ | $77.68_{\pm1.23}$ | $77.28_{\pm1.74}$ | $76.16_{\pm2.73}$ | $77.68_{\pm2.49}$ | $77.72_{\pm1.85}$ | $77.56_{\pm1.78}$ | $76.16_{\pm1.02}$ | $\mathbf{79.84_{\pm1.36}}$ |
| Fourier | 20 | $41.04_{\pm4.56}$ | $\mathbf{42.12_{\pm3.26}}$ | $40.36_{\pm6.81}$ | $27.28_{\pm4.15}$ | $40.96_{\pm6.93}$ | $41.04_{\pm4.56}$ | $28.12_{\pm5.20}$ | $29.12_{\pm6.50}$ | $36.00_{\pm10.80}$ |
| | 50 | $52.52_{\pm4.33}$ | $59.00_{\pm1.82}$ | $52.80_{\pm2.81}$ | $41.36_{\pm4.55}$ | $62.96_{\pm3.59}$ | $52.52_{\pm4.33}$ | $48.76_{\pm5.78}$ | $44.84_{\pm3.74}$ | $\mathbf{64.56_{\pm1.84}}$ |
| | 100 | $65.44_{\pm1.89}$ | $67.88_{\pm1.87}$ | $63.52_{\pm0.96}$ | $54.88_{\pm1.51}$ | $70.68_{\pm1.55}$ | $65.44_{\pm1.89}$ | $62.08_{\pm2.69}$ | $59.76_{\pm3.01}$ | $\mathbf{70.76_{\pm2.73}}$ |
| | 200 | $72.80_{\pm0.72}$ | $74.44_{\pm0.64}$ | $70.88_{\pm1.92}$ | $70.16_{\pm3.75}$ | $75.28_{\pm1.93}$ | $72.80_{\pm0.72}$ | $69.64_{\pm0.62}$ | $69.52_{\pm2.35}$ | $\mathbf{75.56_{\pm1.35}}$ |
| | 500 | $78.04_{\pm1.57}$ | $\mathbf{78.72_{\pm2.17}}$ | $76.12_{\pm1.11}$ | $76.76_{\pm1.42}$ | $77.96_{\pm2.17}$ | $78.04_{\pm1.57}$ | $77.12_{\pm0.48}$ | $77.08_{\pm1.43}$ | $78.20_{\pm1.45}$ |
| Steel | 20 | $65.24_{\pm2.87}$ | $68.48_{\pm5.57}$ | $66.16_{\pm1.92}$ | $64.84_{\pm1.76}$ | $65.60_{\pm3.16}$ | $65.24_{\pm2.87}$ | $66.48_{\pm4.03}$ | $70.32_{\pm4.07}$ | $\mathbf{73.44_{\pm2.10}}$ |
| | 50 | $69.88_{\pm3.99}$ | $88.20_{\pm5.68}$ | $69.36_{\pm2.07}$ | $69.32_{\pm2.38}$ | $70.40_{\pm3.95}$ | $69.88_{\pm3.99}$ | $76.76_{\pm7.83}$ | $86.04_{\pm4.13}$ | $\mathbf{93.72_{\pm3.73}}$ |
| | 100 | $87.04_{\pm3.60}$ | $98.28_{\pm2.33}$ | $73.48_{\pm2.77}$ | $73.36_{\pm2.91}$ | $83.64_{\pm3.12}$ | $87.04_{\pm3.60}$ | $90.08_{\pm4.83}$ | $95.04_{\pm5.00}$ | $\mathbf{100.00_{\pm0.00}}$ |
| | 200 | $98.88_{\pm1.29}$ | $\mathbf{100.00_{\pm0.00}}$ | $83.84_{\pm1.16}$ | $89.08_{\pm3.22}$ | $94.72_{\pm1.34}$ | $98.88_{\pm1.29}$ | $97.20_{\pm0.44}$ | $99.08_{\pm1.08}$ | $99.96_{\pm0.08}$ |
| | 500 | $\mathbf{100.00_{\pm0.00}}$ | $\mathbf{100.00_{\pm0.00}}$ | $94.64_{\pm1.08}$ | $96.52_{\pm1.42}$ | $99.56_{\pm0.29}$ | $\mathbf{100.00_{\pm0.00}}$ | $99.48_{\pm0.41}$ | $99.84_{\pm0.15}$ | $\mathbf{100.00_{\pm0.00}}$ |

*Table 14.* Classification accuracy (%) with $k$-Nearest Neighbors (KNN) as the downstream predictor under varying scarcity levels.

| Dataset | $N_{\text{real}}$ | Real | SMOTE | TVAE | CTGAN | ARF | SPADA | TabDDPM | TabDiff | TAP |
|---|---|---|---|---|---|---|---|---|---|---|
| | 20 | $30.72_{\pm3.39}$ | $32.53_{\pm2.38}$ | $33.24_{\pm1.13}$ | $30.88_{\pm3.37}$ | $32.72_{\pm3.97}$ | $35.52_{\pm3.46}$ | $30.60_{\pm3.41}$ | $36.44_{\pm2.10}$ | $\mathbf{42.44_{\pm2.95}}$ |
| | 50 | $45.72_{\pm4.65}$ | $45.72_{\pm4.65}$ | $43.60_{\pm2.37}$ | $45.72_{\pm4.65}$ | $44.64_{\pm5.17}$ | $47.12_{\pm2.04}$ | $45.72_{\pm4.62}$ | $46.88_{\pm4.45}$ | $\mathbf{54.52_{\pm5.91}}$ |
| MiceProtein | 100 | $58.08_{\pm2.73}$ | $58.08_{\pm2.73}$ | $56.24_{\pm1.04}$ | $58.04_{\pm2.73}$ | $57.52_{\pm3.01}$ | $54.16_{\pm1.81}$ | $58.04_{\pm2.71}$ | $58.16_{\pm2.82}$ | $\mathbf{62.32_{\pm3.78}}$ |
| | 200 | $72.80_{\pm1.56}$ | $72.80_{\pm1.56}$ | $71.96_{\pm0.82}$ | $72.80_{\pm1.56}$ | $72.08_{\pm1.69}$ | $70.52_{\pm1.72}$ | $72.80_{\pm1.56}$ | $72.60_{\pm1.48}$ | $\mathbf{74.52_{\pm1.64}}$ |
| | 500 | $92.40_{\pm2.01}$ | $92.40_{\pm2.01}$ | $92.44_{\pm2.04}$ | $92.40_{\pm2.01}$ | $92.64_{\pm1.84}$ | $92.12_{\pm1.98}$ | $92.40_{\pm2.01}$ | $92.52_{\pm1.97}$ | $\mathbf{92.88_{\pm2.09}}$ |
| | 20 | $66.40_{\pm3.97}$ | $65.68_{\pm3.73}$ | $66.72_{\pm3.34}$ | $66.56_{\pm3.60}$ | $64.64_{\pm2.57}$ | $59.84_{\pm6.03}$ | $63.12_{\pm2.07}$ | $62.92_{\pm3.07}$ | $\mathbf{69.68_{\pm0.85}}$ |
| | 50 | $67.88_{\pm1.61}$ | $67.68_{\pm1.62}$ | $67.08_{\pm1.04}$ | $67.20_{\pm1.20}$ | $67.76_{\pm1.26}$ | $67.76_{\pm2.27}$ | $66.44_{\pm1.82}$ | $67.80_{\pm2.30}$ | $\mathbf{69.48_{\pm1.04}}$ |
| Credit-G | 100 | $67.80_{\pm2.30}$ | $67.80_{\pm2.28}$ | $68.12_{\pm1.95}$ | $66.40_{\pm1.93}$ | $67.56_{\pm2.78}$ | $69.16_{\pm1.59}$ | $66.52_{\pm1.80}$ | $67.24_{\pm1.83}$ | $\mathbf{71.48_{\pm1.62}}$ |
| | 200 | $69.72_{\pm0.73}$ | $69.64_{\pm0.77}$ | $\mathbf{71.08_{\pm1.10}}$ | $68.04_{\pm1.11}$ | $69.44_{\pm1.03}$ | $72.20_{\pm1.23}$ | $69.60_{\pm1.38}$ | $69.52_{\pm2.00}$ | $71.40_{\pm1.67}$ |
| | 500 | $71.40_{\pm1.63}$ | $71.36_{\pm1.59}$ | $72.52_{\pm1.03}$ | $69.76_{\pm1.56}$ | $70.56_{\pm1.34}$ | $\mathbf{72.96_{\pm1.25}}$ | $71.28_{\pm1.56}$ | $71.20_{\pm2.07}$ | $72.92_{\pm0.74}$ |
| | 20 | $68.32_{\pm6.51}$ | $68.28_{\pm6.52}$ | $63.76_{\pm4.81}$ | $59.56_{\pm3.57}$ | $67.64_{\pm5.41}$ | $67.96_{\pm8.10}$ | $63.64_{\pm7.08}$ | $58.12_{\pm9.17}$ | $\mathbf{69.32_{\pm8.60}}$ |
| | 50 | $68.16_{\pm3.45}$ | $68.12_{\pm3.41}$ | $67.00_{\pm1.81}$ | $62.52_{\pm3.44}$ | $69.36_{\pm4.56}$ | $\mathbf{69.56_{\pm4.70}}$ | $67.00_{\pm3.69}$ | $66.00_{\pm3.36}$ | $67.96_{\pm4.97}$ |
| Electricity | 100 | $70.52_{\pm1.45}$ | $70.52_{\pm1.45}$ | $70.16_{\pm3.80}$ | $66.00_{\pm1.74}$ | $72.36_{\pm2.02}$ | $71.20_{\pm1.32}$ | $70.20_{\pm1.40}$ | $67.16_{\pm1.59}$ | $\mathbf{71.92_{\pm2.09}}$ |
| | 200 | $71.56_{\pm1.74}$ | $71.52_{\pm1.75}$ | $73.56_{\pm2.37}$ | $69.40_{\pm1.87}$ | $73.52_{\pm2.32}$ | $71.72_{\pm1.62}$ | $71.24_{\pm1.44}$ | $70.24_{\pm3.87}$ | $\mathbf{73.80_{\pm3.15}}$ |
| | 500 | $73.28_{\pm1.73}$ | $73.28_{\pm1.73}$ | $74.20_{\pm2.60}$ | $72.84_{\pm2.41}$ | $73.80_{\pm1.57}$ | $73.36_{\pm1.66}$ | $73.16_{\pm1.68}$ | $73.00_{\pm2.22}$ | $\mathbf{75.64_{\pm2.16}}$ |
| | 20 | $35.56_{\pm5.47}$ | $35.56_{\pm5.47}$ | $40.44_{\pm5.71}$ | $35.36_{\pm6.34}$ | $36.48_{\pm3.15}$ | $35.56_{\pm5.47}$ | $35.52_{\pm5.42}$ | $39.88_{\pm10.18}$ | $\mathbf{40.48_{\pm7.49}}$ |
| | 50 | $52.80_{\pm1.24}$ | $52.80_{\pm1.24}$ | $49.52_{\pm2.35}$ | $52.88_{\pm1.36}$ | $54.80_{\pm1.48}$ | $52.80_{\pm1.24}$ | $52.80_{\pm1.24}$ | $53.20_{\pm2.12}$ | $\mathbf{57.32_{\pm2.58}}$ |
| Fourier | 100 | $60.24_{\pm1.67}$ | $60.24_{\pm1.67}$ | $60.12_{\pm2.23}$ | $60.12_{\pm1.76}$ | $60.84_{\pm1.54}$ | $60.24_{\pm1.67}$ | $60.24_{\pm1.67}$ | $60.40_{\pm2.24}$ | $\mathbf{63.32_{\pm2.32}}$ |
| | 200 | $68.24_{\pm1.24}$ | $68.24_{\pm1.24}$ | $66.60_{\pm1.15}$ | $68.24_{\pm1.24}$ | $68.16_{\pm1.62}$ | $68.24_{\pm1.24}$ | $68.24_{\pm1.24}$ | $68.12_{\pm1.32}$ | $\mathbf{69.60_{\pm0.63}}$ |
| | 500 | $74.60_{\pm1.81}$ | $74.60_{\pm1.81}$ | $73.68_{\pm1.90}$ | $74.60_{\pm1.81}$ | $74.52_{\pm1.71}$ | $74.60_{\pm1.81}$ | $74.60_{\pm1.81}$ | $74.48_{\pm1.82}$ | $\mathbf{74.86_{\pm1.55}}$ |
| | 20 | $72.04_{\pm4.41}$ | $72.04_{\pm4.41}$ | $70.76_{\pm4.10}$ | $70.76_{\pm4.33}$ | $70.56_{\pm3.35}$ | $72.04_{\pm4.41}$ | $71.32_{\pm5.13}$ | $78.64_{\pm4.71}$ | $\mathbf{79.00_{\pm4.33}}$ |
| | 50 | $82.80_{\pm3.06}$ | $82.80_{\pm3.06}$ | $72.80_{\pm1.18}$ | $82.76_{\pm3.96}$ | $79.24_{\pm3.30}$ | $82.80_{\pm3.06}$ | $83.52_{\pm2.88}$ | $85.08_{\pm3.52}$ | $\mathbf{88.52_{\pm1.59}}$ |
| Steel | 100 | $90.40_{\pm0.66}$ | $90.40_{\pm0.66}$ | $77.60_{\pm1.77}$ | $90.28_{\pm0.55}$ | $88.76_{\pm0.85}$ | $90.40_{\pm0.66}$ | $90.44_{\pm0.77}$ | $92.36_{\pm1.25}$ | $93.76_{\pm1.32}$ |
| | 200 | $93.36_{\pm1.18}$ | $93.36_{\pm1.18}$ | $85.48_{\pm1.90}$ | $93.36_{\pm1.05}$ | $94.20_{\pm1.57}$ | $93.36_{\pm1.18}$ | $93.28_{\pm1.19}$ | $94.60_{\pm1.09}$ | $94.60_{\pm1.29}$ |
| | 500 | $96.60_{\pm0.36}$ | $96.60_{\pm0.36}$ | $93.12_{\pm1.01}$ | $96.68_{\pm0.39}$ | $\mathbf{97.04_{\pm0.50}}$ | $96.60_{\pm0.36}$ | $96.60_{\pm0.36}$ | $96.76_{\pm0.71}$ | $96.92_{\pm0.63}$ |

*Table 15.* Classification accuracy (%) with a multilayer perceptron (MLP) as the downstream predictor under varying scarcity levels.

| Dataset | $N_{\text{real}}$ | Real | SMOTE | TVAE | CTGAN | ARF | SPADA | TabDDPM | TabDiff | TAP |
|---|---|---|---|---|---|---|---|---|---|---|
| | 20 | $45.08_{\pm4.96}$ | $44.53_{\pm7.52}$ | $39.24_{\pm5.37}$ | $35.48_{\pm2.34}$ | $40.08_{\pm5.55}$ | $42.92_{\pm4.28}$ | $39.20_{\pm5.55}$ | $41.32_{\pm3.16}$ | $\mathbf{48.80_{\pm3.99}}$ |
| | 50 | $64.00_{\pm3.36}$ | $\mathbf{66.12_{\pm3.22}}$ | $52.36_{\pm3.76}$ | $55.40_{\pm4.60}$ | $59.24_{\pm4.24}$ | $56.60_{\pm2.73}$ | $60.92_{\pm3.26}$ | $58.72_{\pm2.15}$ | $64.32_{\pm3.39}$ |
| MiceProtein | 100 | $77.24_{\pm2.70}$ | $77.64_{\pm2.53}$ | $67.00_{\pm2.15}$ | $73.36_{\pm2.24}$ | $69.88_{\pm2.38}$ | $74.20_{\pm0.83}$ | $75.96_{\pm2.82}$ | $73.28_{\pm3.37}$ | $\mathbf{77.82_{\pm3.33}}$ |
| | 200 | $93.12_{\pm2.33}$ | $93.40_{\pm1.86}$ | $84.24_{\pm2.99}$ | $90.88_{\pm1.48}$ | $88.84_{\pm2.44}$ | $90.24_{\pm2.85}$ | $\mathbf{92.56_{\pm1.85}}$ | $88.96_{\pm2.29}$ | $91.40_{\pm2.11}$ |
| | 500 | $99.12_{\pm0.30}$ | $\mathbf{99.52_{\pm0.39}}$ | $96.80_{\pm1.13}$ | $98.36_{\pm0.73}$ | $97.56_{\pm0.90}$ | $98.84_{\pm0.20}$ | $98.76_{\pm0.80}$ | $98.48_{\pm0.45}$ | $99.08_{\pm0.39}$ |
| | 20 | $63.36_{\pm4.70}$ | $59.24_{\pm5.61}$ | $64.16_{\pm3.16}$ | $65.16_{\pm3.66}$ | $59.28_{\pm6.75}$ | $57.72_{\pm3.85}$ | $58.00_{\pm4.95}$ | $61.00_{\pm3.18}$ | $\mathbf{67.80_{\pm3.23}}$ |
| | 50 | $65.24_{\pm2.33}$ | $65.04_{\pm2.89}$ | $67.44_{\pm1.84}$ | $65.36_{\pm1.23}$ | $62.84_{\pm2.90}$ | $64.80_{\pm4.75}$ | $59.04_{\pm3.42}$ | $65.56_{\pm3.10}$ | $\mathbf{69.84_{\pm0.23}}$ |
| Credit-G | 100 | $66.84_{\pm1.16}$ | $67.12_{\pm1.25}$ | $65.76_{\pm2.15}$ | $63.92_{\pm2.45}$ | $64.32_{\pm1.59}$ | $64.08_{\pm2.13}$ | $62.56_{\pm1.39}$ | $65.72_{\pm1.71}$ | $\mathbf{71.36_{\pm1.23}}$ |
| | 200 | $67.32_{\pm0.41}$ | $67.72_{\pm2.14}$ | $67.84_{\pm1.11}$ | $63.68_{\pm2.48}$ | $66.44_{\pm2.92}$ | $65.32_{\pm2.10}$ | $65.00_{\pm3.82}$ | $67.56_{\pm1.76}$ | $\mathbf{70.80_{\pm2.38}}$ |
| | 500 | $70.84_{\pm0.69}$ | $71.64_{\pm1.27}$ | $70.96_{\pm0.91}$ | $69.68_{\pm2.81}$ | $71.12_{\pm1.22}$ | $70.64_{\pm0.77}$ | $69.64_{\pm1.58}$ | $71.40_{\pm0.70}$ | $\mathbf{74.16_{\pm1.33}}$ |
| | 20 | $66.68_{\pm4.80}$ | $60.40_{\pm5.12}$ | $66.80_{\pm5.18}$ | $59.68_{\pm5.47}$ | $66.80_{\pm4.53}$ | $66.44_{\pm7.04}$ | $57.20_{\pm8.29}$ | $61.52_{\pm8.58}$ | $\mathbf{70.36_{\pm8.16}}$ |
| | 50 | $66.76_{\pm4.33}$ | $63.60_{\pm4.05}$ | $67.36_{\pm5.07}$ | $64.00_{\pm1.56}$ | $70.12_{\pm6.22}$ | $69.00_{\pm4.84}$ | $64.56_{\pm4.91}$ | $68.28_{\pm4.11}$ | $\mathbf{71.88_{\pm4.41}}$ |
| Electricity | 100 | $72.40_{\pm4.19}$ | $62.32_{\pm6.15}$ | $73.36_{\pm5.04}$ | $66.56_{\pm2.64}$ | $73.96_{\pm3.19}$ | $71.76_{\pm5.60}$ | $70.92_{\pm3.38}$ | $69.48_{\pm3.17}$ | $\mathbf{76.28_{\pm3.55}}$ |
| | 200 | $74.80_{\pm4.58}$ | $67.40_{\pm4.05}$ | $73.32_{\pm3.86}$ | $72.20_{\pm3.88}$ | $74.20_{\pm4.91}$ | $73.80_{\pm5.34}$ | $75.00_{\pm2.70}$ | $71.68_{\pm5.00}$ | $\mathbf{76.32_{\pm3.18}}$ |
| | 500 | $75.84_{\pm2.82}$ | $71.52_{\pm3.31}$ | $75.52_{\pm3.01}$ | $75.16_{\pm2.60}$ | $76.04_{\pm2.65}$ | $74.92_{\pm3.31}$ | $77.16_{\pm2.74}$ | $75.60_{\pm3.31}$ | $\mathbf{76.56_{\pm2.96}}$ |
| | 20 | $45.16_{\pm2.77}$ | $\mathbf{47.44_{\pm2.72}}$ | $42.24_{\pm4.40}$ | $27.60_{\pm4.06}$ | $40.00_{\pm3.44}$ | $45.16_{\pm2.77}$ | $30.32_{\pm4.34}$ | $29.04_{\pm4.86}$ | $45.44_{\pm4.16}$ |
| | 50 | $57.96_{\pm1.50}$ | $61.56_{\pm2.01}$ | $53.40_{\pm2.49}$ | $43.24_{\pm2.38}$ | $60.72_{\pm3.06}$ | $57.96_{\pm1.50}$ | $48.20_{\pm3.56}$ | $46.80_{\pm2.96}$ | $\mathbf{62.24_{\pm1.17}}$ |
| Fourier | 100 | $64.92_{\pm2.17}$ | $\mathbf{68.76_{\pm1.70}}$ | $60.32_{\pm1.57}$ | $54.00_{\pm1.66}$ | $68.08_{\pm0.95}$ | $64.92_{\pm2.17}$ | $63.20_{\pm2.62}$ | $55.52_{\pm3.02}$ | $66.68_{\pm2.90}$ |
| | 200 | $71.68_{\pm1.78}$ | $\mathbf{74.68_{\pm1.74}}$ | $68.80_{\pm2.59}$ | $66.92_{\pm2.62}$ | $73.24_{\pm1.97}$ | $71.68_{\pm1.78}$ | $70.88_{\pm1.06}$ | $66.00_{\pm3.27}$ | $73.76_{\pm1.94}$ |
| | 500 | $77.48_{\pm1.63}$ | $\mathbf{78.84_{\pm1.49}}$ | $74.12_{\pm1.80}$ | $75.28_{\pm2.12}$ | $77.76_{\pm2.13}$ | $77.48_{\pm1.63}$ | $77.36_{\pm1.31}$ | $73.60_{\pm1.87}$ | $77.84_{\pm1.52}$ |
| | 20 | $75.72_{\pm4.64}$ | $78.12_{\pm4.99}$ | $63.52_{\pm3.25}$ | $65.12_{\pm5.71}$ | $71.36_{\pm1.99}$ | $75.72_{\pm4.64}$ | $72.00_{\pm4.50}$ | $77.92_{\pm3.06}$ | $\mathbf{78.44_{\pm2.98}}$ |
| | 50 | $88.92_{\pm2.90}$ | $91.92_{\pm1.65}$ | $79.40_{\pm4.08}$ | $82.04_{\pm4.54}$ | $74.64_{\pm2.40}$ | $88.92_{\pm2.90}$ | $91.76_{\pm3.73}$ | $92.04_{\pm2.89}$ | $\mathbf{97.52_{\pm1.37}}$ |
| Steel | 100 | $96.64_{\pm1.81}$ | $98.60_{\pm1.16}$ | $85.84_{\pm1.43}$ | $90.20_{\pm2.44}$ | $87.84_{\pm1.50}$ | $96.64_{\pm1.81}$ | $98.08_{\pm0.79}$ | $97.48_{\pm2.55}$ | $\mathbf{99.48_{\pm0.43}}$ |
| | 200 | $99.12_{\pm0.41}$ | $99.60_{\pm0.33}$ | $91.56_{\pm1.49}$ | $95.60_{\pm0.72}$ | $96.68_{\pm1.42}$ | $99.12_{\pm0.41}$ | $99.48_{\pm0.35}$ | $99.48_{\pm0.43}$ | $\mathbf{99.84_{\pm0.15}}$ |
| | 500 | $99.76_{\pm0.23}$ | $99.84_{\pm0.15}$ | $97.24_{\pm0.87}$ | $98.96_{\pm0.62}$ | $99.24_{\pm0.34}$ | $99.76_{\pm0.23}$ | $99.80_{\pm0.22}$ | $99.88_{\pm0.10}$ | $\mathbf{99.88_{\pm0.16}}$ |

*Table 16.* Regression Root Mean Squared Error (RMSE) with Random Forest (RF) as the downstream predictor under varying scarcity levels.

| Dataset | $N_{real}$ | Real | SMOTE | TVAE | CTGAN | ARF | SPADA | TabDDPM | TabDiff | TAP |
|---|---|---|---|---|---|---|---|---|---|---|
| Ailerons | 20 | $0.917_{\pm0.176}$ | $1.053_{\pm0.221}$ | $1.024_{\pm0.235}$ | $1.118_{\pm0.220}$ | $\mathbf{0.907_{\pm0.181}}$ | $1.019_{\pm0.145}$ | $0.983_{\pm0.238}$ | $1.005_{\pm0.209}$ | $0.912_{\pm0.175}$ |
| | 50 | $0.723_{\pm0.150}$ | $0.820_{\pm0.130}$ | $0.870_{\pm0.204}$ | $0.873_{\pm0.150}$ | $0.738_{\pm0.153}$ | $0.782_{\pm0.122}$ | $0.770_{\pm0.144}$ | $0.920_{\pm0.202}$ | $\mathbf{0.699_{\pm0.135}}$ |
| | 100 | $0.551_{\pm0.079}$ | $0.655_{\pm0.087}$ | $0.663_{\pm0.092}$ | $0.754_{\pm0.079}$ | $0.561_{\pm0.090}$ | $0.608_{\pm0.096}$ | $54.798_{\pm108.343}$ | $0.656_{\pm0.079}$ | $\mathbf{0.533_{\pm0.071}}$ |
| | 200 | $0.514_{\pm0.054}$ | $0.575_{\pm0.064}$ | $0.589_{\pm0.058}$ | $0.629_{\pm0.066}$ | $0.526_{\pm0.052}$ | $0.522_{\pm0.049}$ | $14.533_{\pm27.967}$ | $0.551_{\pm0.075}$ | $\mathbf{0.507_{\pm0.055}}$ |
| | 500 | $0.454_{\pm0.040}$ | $0.502_{\pm0.041}$ | $0.502_{\pm0.040}$ | $0.552_{\pm0.084}$ | $0.465_{\pm0.038}$ | $0.456_{\pm0.036}$ | $49.334_{\pm97.739}$ | $0.472_{\pm0.041}$ | $\mathbf{0.448_{\pm0.038}}$ |
| Insurance | 20 | $0.883_{\pm0.331}$ | $0.866_{\pm0.239}$ | $0.969_{\pm0.266}$ | $1.228_{\pm0.221}$ | $1.198_{\pm0.217}$ | $1.425_{\pm0.170}$ | $0.917_{\pm0.318}$ | $1.040_{\pm0.198}$ | $\mathbf{0.832_{\pm0.349}}$ |
| | 50 | $0.893_{\pm0.321}$ | $0.850_{\pm0.294}$ | $0.914_{\pm0.340}$ | $1.228_{\pm0.205}$ | $1.083_{\pm0.204}$ | $1.541_{\pm0.187}$ | $0.973_{\pm0.409}$ | $0.811_{\pm0.269}$ | $\mathbf{0.603_{\pm0.224}}$ |
| | 100 | $0.598_{\pm0.070}$ | $0.606_{\pm0.073}$ | $0.708_{\pm0.085}$ | $1.328_{\pm0.178}$ | $0.794_{\pm0.149}$ | $1.351_{\pm0.092}$ | $0.601_{\pm0.071}$ | $0.581_{\pm0.059}$ | $\mathbf{0.468_{\pm0.073}}$ |
| | 200 | $0.630_{\pm0.036}$ | $0.623_{\pm0.031}$ | $0.665_{\pm0.037}$ | $1.218_{\pm0.124}$ | $0.693_{\pm0.056}$ | $1.180_{\pm0.169}$ | $0.661_{\pm0.073}$ | $0.527_{\pm0.045}$ | $\mathbf{0.436_{\pm0.020}}$ |
| | 500 | $0.636_{\pm0.019}$ | $0.643_{\pm0.020}$ | $0.619_{\pm0.052}$ | $1.189_{\pm0.076}$ | $0.734_{\pm0.031}$ | $0.786_{\pm0.212}$ | $0.775_{\pm0.174}$ | $0.452_{\pm0.103}$ | $\mathbf{0.447_{\pm0.022}}$ |

*Table 17.* Regression Root Mean Squared Error (RMSE) with LightGBM (LGBM) as the downstream predictor under varying scarcity levels.

| Dataset | $N_{real}$ | Real | SMOTE | TVAE | CTGAN | ARF | SPADA | TabDDPM | TabDiff | TAP |
|---|---|---|---|---|---|---|---|---|---|---|
| Ailerons | 20 | $1.194_{\pm0.209}$ | $1.128_{\pm0.242}$ | $1.036_{\pm0.219}$ | $1.130_{\pm0.177}$ | $0.908_{\pm0.189}$ | $1.094_{\pm0.147}$ | $1.131_{\pm0.169}$ | $0.991_{\pm0.160}$ | $\mathbf{0.891_{\pm0.178}}$ |
| | 50 | $0.822_{\pm0.135}$ | $0.771_{\pm0.142}$ | $0.892_{\pm0.194}$ | $0.915_{\pm0.164}$ | $0.724_{\pm0.147}$ | $0.779_{\pm0.093}$ | $0.779_{\pm0.181}$ | $0.913_{\pm0.204}$ | $\mathbf{0.658_{\pm0.127}}$ |
| | 100 | $0.559_{\pm0.078}$ | $0.563_{\pm0.076}$ | $0.665_{\pm0.096}$ | $0.750_{\pm0.057}$ | $0.546_{\pm0.086}$ | $0.567_{\pm0.094}$ | $107.009_{\pm212.828}$ | $0.661_{\pm0.076}$ | $\mathbf{0.497_{\pm0.062}}$ |
| | 200 | $0.519_{\pm0.045}$ | $0.526_{\pm0.051}$ | $0.581_{\pm0.060}$ | $0.645_{\pm0.068}$ | $0.505_{\pm0.057}$ | $0.512_{\pm0.051}$ | $632.574_{\pm652.954}$ | $0.565_{\pm0.067}$ | $\mathbf{0.484_{\pm0.052}}$ |
| | 500 | $0.449_{\pm0.028}$ | $0.468_{\pm0.027}$ | $0.505_{\pm0.031}$ | $0.559_{\pm0.074}$ | $0.446_{\pm0.036}$ | $0.451_{\pm0.027}$ | $632.574_{\pm652.954}$ | $0.473_{\pm0.042}$ | $\mathbf{0.438_{\pm0.032}}$ |
| Insurance | 20 | $1.158_{\pm0.204}$ | $1.184_{\pm0.184}$ | $1.072_{\pm0.200}$ | $1.260_{\pm0.257}$ | $1.196_{\pm0.206}$ | $1.436_{\pm0.224}$ | $1.108_{\pm0.341}$ | $1.066_{\pm0.185}$ | $\mathbf{0.921_{\pm0.489}}$ |
| | 50 | $1.182_{\pm0.332}$ | $0.853_{\pm0.257}$ | $0.915_{\pm0.277}$ | $1.232_{\pm0.213}$ | $1.078_{\pm0.216}$ | $1.385_{\pm0.195}$ | $1.052_{\pm0.324}$ | $0.824_{\pm0.268}$ | $\mathbf{0.624_{\pm0.227}}$ |
| | 100 | $0.695_{\pm0.134}$ | $0.672_{\pm0.125}$ | $0.694_{\pm0.142}$ | $1.358_{\pm0.156}$ | $0.831_{\pm0.121}$ | $1.277_{\pm0.269}$ | $0.730_{\pm0.134}$ | $0.582_{\pm0.058}$ | $\mathbf{0.492_{\pm0.103}}$ |
| | 200 | $0.565_{\pm0.043}$ | $0.649_{\pm0.026}$ | $0.638_{\pm0.048}$ | $1.235_{\pm0.112}$ | $0.682_{\pm0.040}$ | $1.258_{\pm0.138}$ | $0.679_{\pm0.187}$ | $0.558_{\pm0.055}$ | $\mathbf{0.433_{\pm0.018}}$ |
| | 500 | $0.616_{\pm0.009}$ | $0.619_{\pm0.012}$ | $0.632_{\pm0.047}$ | $1.207_{\pm0.073}$ | $0.717_{\pm0.040}$ | $0.733_{\pm0.151}$ | $0.737_{\pm0.100}$ | $0.450_{\pm0.015}$ | $\mathbf{0.448_{\pm0.084}}$ |

*Table 18.* Regression Root Mean Squared Error (RMSE) with XGBoost (XGB) as the downstream predictor under varying scarcity levels.

| Dataset | $N_{real}$ | Real | SMOTE | TVAE | CTGAN | ARF | SPADA | TabDDPM | TabDiff | TAP |
|---|---|---|---|---|---|---|---|---|---|---|
| Ailerons | 20 | $1.064_{\pm0.134}$ | $1.132_{\pm0.186}$ | $1.098_{\pm0.254}$ | $1.176_{\pm0.232}$ | $\mathbf{0.907_{\pm0.188}}$ | $1.028_{\pm0.131}$ | $1.023_{\pm0.207}$ | $1.035_{\pm0.156}$ | $0.923_{\pm0.177}$ |
| | 50 | $0.747_{\pm0.136}$ | $0.873_{\pm0.152}$ | $0.905_{\pm0.172}$ | $0.976_{\pm0.175}$ | $0.733_{\pm0.157}$ | $0.804_{\pm0.109}$ | $0.810_{\pm0.133}$ | $0.941_{\pm0.189}$ | $\mathbf{0.703_{\pm0.143}}$ |
| | 100 | $0.580_{\pm0.082}$ | $0.668_{\pm0.073}$ | $0.678_{\pm0.072}$ | $0.814_{\pm0.032}$ | $0.551_{\pm0.080}$ | $0.627_{\pm0.076}$ | $271.171_{\pm541.083}$ | $0.665_{\pm0.080}$ | $\mathbf{0.546_{\pm0.082}}$ |
| | 200 | $0.555_{\pm0.049}$ | $0.571_{\pm0.076}$ | $0.595_{\pm0.052}$ | $0.702_{\pm0.065}$ | $\mathbf{0.517_{\pm0.052}}$ | $0.562_{\pm0.060}$ | $191.099_{\pm244.975}$ | $0.597_{\pm0.079}$ | $0.518_{\pm0.057}$ |
| | 500 | $0.483_{\pm0.027}$ | $0.499_{\pm0.027}$ | $0.530_{\pm0.032}$ | $0.615_{\pm0.089}$ | $0.472_{\pm0.029}$ | $0.475_{\pm0.037}$ | $366.519_{\pm152.681}$ | $0.499_{\pm0.029}$ | $\mathbf{0.472_{\pm0.028}}$ |
| Insurance | 20 | $0.917_{\pm0.530}$ | $\mathbf{0.831_{\pm0.268}}$ | $1.026_{\pm0.241}$ | $1.353_{\pm0.223}$ | $1.162_{\pm0.258}$ | $1.463_{\pm0.166}$ | $0.966_{\pm0.429}$ | $1.153_{\pm0.189}$ | $0.907_{\pm0.408}$ |
| | 50 | $0.879_{\pm0.346}$ | $0.839_{\pm0.294}$ | $0.933_{\pm0.316}$ | $1.318_{\pm0.201}$ | $1.121_{\pm0.199}$ | $1.584_{\pm0.175}$ | $0.955_{\pm0.417}$ | $0.883_{\pm0.274}$ | $\mathbf{0.643_{\pm0.221}}$ |
| | 100 | $0.579_{\pm0.047}$ | $0.564_{\pm0.055}$ | $0.744_{\pm0.058}$ | $1.432_{\pm0.174}$ | $0.755_{\pm0.128}$ | $1.456_{\pm0.212}$ | $0.587_{\pm0.057}$ | $0.640_{\pm0.078}$ | $\mathbf{0.488_{\pm0.094}}$ |
| | 200 | $0.633_{\pm0.049}$ | $0.628_{\pm0.042}$ | $0.720_{\pm0.041}$ | $1.285_{\pm0.130}$ | $0.725_{\pm0.043}$ | $1.323_{\pm0.203}$ | $0.758_{\pm0.216}$ | $0.603_{\pm0.093}$ | $\mathbf{0.454_{\pm0.038}}$ |
| | 500 | $0.629_{\pm0.005}$ | $0.689_{\pm0.049}$ | $0.717_{\pm0.135}$ | $1.235_{\pm0.067}$ | $0.758_{\pm0.105}$ | $0.824_{\pm0.228}$ | $0.834_{\pm0.193}$ | $0.504_{\pm0.188}$ | $\mathbf{0.480_{\pm0.013}}$ |

*Table 19.* Regression Root Mean Squared Error (RMSE) with $k$-Nearest Neighbors (KNN) as the downstream predictor under varying scarcity levels.

| Dataset | $N_{real}$ | Real | SMOTE | TVAE | CTGAN | ARF | SPADA | TabDDPM | TabDiff | TAP |
|---|---|---|---|---|---|---|---|---|---|---|
| Ailerons | 20 | $\mathbf{0.994_{\pm0.239}}$ | $0.994_{\pm0.239}$ | $1.026_{\pm0.215}$ | $1.006_{\pm0.215}$ | $0.982_{\pm0.200}$ | $1.118_{\pm0.159}$ | $1.003_{\pm0.238}$ | $1.030_{\pm0.208}$ | $1.031_{\pm0.234}$ |
| | 50 | $0.869_{\pm0.179}$ | $0.869_{\pm0.179}$ | $0.926_{\pm0.178}$ | $0.892_{\pm0.161}$ | $0.886_{\pm0.207}$ | $0.966_{\pm0.162}$ | $0.873_{\pm0.179}$ | $0.904_{\pm0.156}$ | $\mathbf{0.855_{\pm0.197}}$ |
| | 100 | $0.696_{\pm0.100}$ | $0.696_{\pm0.100}$ | $0.728_{\pm0.090}$ | $0.707_{\pm0.092}$ | $\mathbf{0.692_{\pm0.096}}$ | $0.764_{\pm0.088}$ | $0.696_{\pm0.100}$ | $0.710_{\pm0.092}$ | $0.705_{\pm0.093}$ |
| | 200 | $0.690_{\pm0.072}$ | $0.690_{\pm0.072}$ | $0.716_{\pm0.069}$ | $0.693_{\pm0.070}$ | $\mathbf{0.680_{\pm0.081}}$ | $0.723_{\pm0.072}$ | $0.690_{\pm0.072}$ | $0.704_{\pm0.088}$ | $0.693_{\pm0.089}$ |
| | 500 | $0.620_{\pm0.056}$ | $0.620_{\pm0.056}$ | $0.634_{\pm0.055}$ | $0.621_{\pm0.057}$ | $0.616_{\pm0.059}$ | $0.643_{\pm0.043}$ | $0.620_{\pm0.056}$ | $0.620_{\pm0.070}$ | $\mathbf{0.615_{\pm0.055}}$ |
| Insurance | 20 | $0.925_{\pm0.279}$ | $0.925_{\pm0.279}$ | $1.002_{\pm0.233}$ | $1.226_{\pm0.262}$ | $1.027_{\pm0.259}$ | $1.416_{\pm0.206}$ | $0.925_{\pm0.279}$ | $1.010_{\pm0.218}$ | $\mathbf{0.879_{\pm0.380}}$ |
| | 50 | $0.794_{\pm0.259}$ | $0.794_{\pm0.259}$ | $0.837_{\pm0.264}$ | $1.215_{\pm0.222}$ | $0.873_{\pm0.231}$ | $1.322_{\pm0.123}$ | $0.794_{\pm0.259}$ | $0.847_{\pm0.264}$ | $\mathbf{0.657_{\pm0.226}}$ |
| | 100 | $0.593_{\pm0.117}$ | $0.593_{\pm0.117}$ | $0.660_{\pm0.128}$ | $1.158_{\pm0.157}$ | $0.653_{\pm0.101}$ | $0.975_{\pm0.087}$ | $0.593_{\pm0.117}$ | $0.624_{\pm0.069}$ | $\mathbf{0.561_{\pm0.119}}$ |
| | 200 | $0.544_{\pm0.042}$ | $0.544_{\pm0.042}$ | $0.591_{\pm0.038}$ | $0.925_{\pm0.113}$ | $0.594_{\pm0.051}$ | $0.646_{\pm0.069}$ | $0.544_{\pm0.042}$ | $0.580_{\pm0.061}$ | $\mathbf{0.514_{\pm0.037}}$ |
| | 500 | $0.492_{\pm0.022}$ | $0.492_{\pm0.022}$ | $0.511_{\pm0.021}$ | $0.764_{\pm0.058}$ | $0.532_{\pm0.018}$ | $0.515_{\pm0.024}$ | $0.492_{\pm0.022}$ | $0.493_{\pm0.008}$ | $\mathbf{0.484_{\pm0.023}}$ |

