# OpenReview forum: "Active Tabular Augmentation via Policy-Guided Diffusion Inpainting"
_ICML.cc/2026/Conference — ICML 2026 regular_

### Official Review · Reviewer_kRQs · 2026-03-02

**Soundness:** 3
**Presentation:** 3
**Significance:** 3
**Originality:** 3
**Overall Recommendation:** 4
**Confidence:** 2

**Summary:**

This paper proposes TAP, a tabular data augmentation method based on the fidelity-utility gap, to address the issue that generative tabular augmentation fails to effectively improve the performance of downstream tasks. TAP generates tabular data via a trained policy and determines its usability for downstream tasks based on marginal utility. A series of experiments verifies the effectiveness of the proposed method.

**Compliance With Llm Reviewing Policy:**

Affirmed.

**Final Justification:**

The rebuttal has addressed my concerns. I'll maintain my positive score.

**Key Questions For Authors:**

(1) What is the optimization objective of Eq.(2)? Is it to find $S$ that maximizes $\Delta U$?

(2) What is the formula for computing $s_t$​?

(3) TAP uses a diffusion model to generate data and selects valid data based on a threshold.
If all generated data in pool $P$ are invalid when $ (t+1) \mathrm{mod} K=0$, will they be entirely discarded?

(4) Is the reference policy $\pi_{\mathrm{ref}}$ derived from a pre-trained model?

**Limitations:**

yes

**Strengths And Weaknesses:**

## Strengths:
* For tabular data augmentation in data-scarce scenarios, this paper formalizes the fidelity-utility gap of generated data to support downstream task training.
* To improve the fidelity-utility of generated data, this paper designs three guiding principles.
* Based on these three principles, this paper proposes a policy-guided tabular data augmentation approach.
* This paper validates the fidelity-utility gap and the effectiveness of the proposed method on various downstream utility tasks.

## Weaknesses:
* A flowchart is suggested to provide an intuitive introduction to the method.
* Some notations in the paper are not explained, such as the relationship between $z$ and $(x,y)$ in Eq.(3), and $D_0$ in Eq.(5).
* The datasets used in the experiments are relatively small. Additional experiments are required for further validation, e.g., on TabDIFF [1] and TabStruct [2].

[1] TabDiff: a Mixed-type Diffusion Model for Tabular Data Generation. Shi et al., ICLR 2025

[2] TabStruct: Measuring Structural Fidelity of Tabular Data. Jiang et al., ICLR 2026

---

> ### Author Rebuttal · Authors · 2026-03-28
>
> Thank you for the careful read and for listing the questions so clearly! We answer them point by point below.
>
> **W1: Flowchart and notation clarity.**
> We agree. The Method section would benefit from a clearer visual summary and cleaner notation. In the revision, we will add a compact workflow figure and clarify the notation around $z$ in Eq. (3), $D_0$ in Eq. (5), and the full state/action definitions. In particular, $z=(x,y)$ in Eq. (3) is the candidate example being hypothetically added, while $D_0$ in Eq. (5) is the initial real labeled set.
>
> **Q1: What is the objective around Eq. (2)?**
> Eq. (2) defines the **marginal utility** of injecting a candidate set $S$, namely how much the real-query loss decreases after adding $S$. It is not the quantity we optimize exactly, because doing so would require retraining the downstream learner for every candidate set. The actual policy objective is the trajectory-level utility in Eq. (6), and Eq. (7) shows how this utility telescopes over commitment events. So TAP optimizes a plug-in approximation to utility online, rather than maximizing Eq. (2) directly.
>
> **Q2: What is $s_t$?**
> $s_t$ is a vector-valued learner-state summary, not a single scalar. In our implementation,
> $s_t = (\delta_t, u_t, g_t, d_t)$,
> where $\delta_t$ tracks target deficit, $u_t$ summarizes predictive uncertainty, $g_t$ records recent gate-pass statistics, and $d_t$ measures diversity relative to the committed buffer. Appendix A gives the design rationale, and Appendix E shows that these components contribute complementary information.
>
> **Q3: What happens if all samples in the pool are invalid at a commitment check?**
> Yes, they are discarded. Invalid candidates are first removed by the pointwise gates. If the remaining pool is empty, or if the pooled utility does not pass the $\tau + \epsilon_t$ threshold, then nothing is committed in that window, the committed buffer stays unchanged, and the pool is reset. So TAP is explicitly allowed to “do nothing” when the candidates are unreliable.
>
> **Q4: Is the reference policy $\pi_{\text{ref}}$ pretrained?**
> No. $\pi_{\text{ref}}$ is not a separately pretrained model. It is a conservative heuristic reference policy used during policy learning. In our implementation, it favors safer actions, for example by reducing exploration when feasibility is low, and serves as a stable default that regularizes early policy updates.
>
> **W2 / W3: Dataset scale, TabDiff, and TabStruct.**
> We would also like to clarify two points here. First, TabDiff [1] is already in the paper, both as a baseline and as the shared backbone in Fig. 3. Second, our datasets are intentionally subsampled to $n_{\text{real}} \in \{20,50,100,200,500\}$ because the paper studies augmentation **under scarcity**. For TabStruct [2], we see it as relevant but complementary rather than a missing experiment. TabStruct benchmarks **tabular generators** through structural fidelity and global utility, whereas our paper studies the augmentation policy on top of a fixed strong generator. This is why TabDiff [1] is already included both as a baseline and as the shared backbone in Fig. 3. We use it because it is a strong mixed-type diffusion generator for inpainting proposals, while TAP’s contribution is what to inject, when to inject, and how to avoid harmful injection under scarcity. The inpainting mechanism itself follows the diffusion inpainting line exemplified by RePaint [3], adapted here to the tabular setting. We will make this distinction clearer in the revision.
>
> ```
> [1] Shi et al. "TabDiff: a Mixed-type Diffusion Model for Tabular Data Generation." ICLR, 2025.
> [2] Jiang et al. "TabStruct: Measuring Structural Fidelity of Tabular Data." ICLR, 2026.
> [3] Lugmayr et al. "RePaint: Inpainting Using Denoising Diffusion Probabilistic Models." CVPR, 2022.
> ```

---

> > ### Author Rebuttal · Reviewer_kRQs · 2026-04-02
> >
> > Thank the authors sincerely for their detailed and thoughtful response. My previous concerns have been fully addressed, with no additional comments needed. I will retain my original score.

---

> > > ### Author Response · Authors · 2026-04-05
> > >
> > > Dear Reviewer kRQs,
> > >
> > > Thank you for your insightful feedback, which has greatly helped improve our manuscript. Below is a brief summary to facilitate further discussion.
> > >
> > > 1. We clarified the objective around Eq. (2), the notation in Eqs. (3) and (5), the learner-state definition, the commitment behavior when a pool is rejected, and the role of the reference policy.
> > > 2. We clarified the intended experimental scope, including why TabDiff is already part of the paper and why TabStruct is relevant but complementary to our setting.
> > > 3. A compact workflow figure and cleaner notation will be incorporated in the revision to make the method easier to follow.
> > >
> > > We look forward to further discussion and are happy to address any remaining concerns. As the main issues from the original review seem to have been resolved, we would be grateful if you could kindly reconsider the current rating (4) and confidence (2). Thank you for your time and consideration!
> > >
> > > Authors

---

### Official Review · Reviewer_DeHM · 2026-03-12

**Soundness:** 3
**Presentation:** 3
**Significance:** 3
**Originality:** 3
**Overall Recommendation:** 4
**Confidence:** 3

**Summary:**

This paper discusses the problem of generative tabular augmentation in data-scarce domains. The authors formalize a fidelity-utility gap and, based on this, propose TAP (Tabular Augmentation Policy), a state-conditioned policy that steers diffusion inpainting through target, template, and exploration controls, with safe injection enforced by hard gating and windowed commitment. The authors conducted extensive comparative experiments on seven real-world datasets and five scarcity levels, demonstrating that TAP can outperform existing SOTA methods.

**Compliance With Llm Reviewing Policy:**

Affirmed.

**Ethical Review Flag:**

Flag this paper for an ethics review.

**Ethics Expertise Needed:**

["Responsible Research Practice (e.g., IRB, documentation, research ethics)"]

**Key Questions For Authors:**

(1) High computational overhead and method complexity.
Although TAP achieves promising performance improvements, its practical applicability is somewhat limited by its substantial computational overhead. As reported in Table 7, the runtime cost of TAP is significantly higher than that of Global sampling. In addition, the framework introduces several extra components, such as a policy network and an online evaluator, which increase both implementation complexity and training cost. Therefore, it would be helpful if the authors could further clarify whether the performance gains consistently justify this added complexity, especially in real-world low-resource scenarios where efficiency may also be a major concern.

(2) Limited gains when data scarcity is less severe.
The proposed method is particularly effective in highly data-scarce settings, which is an important strength of the paper. However, Figure 2 also suggests that the advantage of TAP becomes notably smaller when 𝑁_real = 500. This raises an important practical question: when the amount of real data is no longer extremely limited, is the additional overhead of TAP still worthwhile compared with simpler alternatives? The paper would be stronger if the authors could more clearly characterize the regime in which TAP is most beneficial and discuss when practitioners should prefer TAP over lighter baselines.

(3) The Method section could be made more intuitive.
While the proposed design is sophisticated, the Method section is relatively dense and lacks sufficient visual illustrations. In particular, the interactions among the target, template, and exploration controls, as well as the roles of Hard Feasibility Gates and Windowed Commitment, are not immediately intuitive from the current presentation. Adding a pipeline figure or a more explicit visual summary would significantly improve readability and help readers better understand the overall workflow and design motivation.

**Limitations:**

(1) High overhead and complexity: As shown in Table 7, the time overhead of TAP far exceeds that of Global sampling, and it requires introducing multiple model components (such as a policy network and an online evaluator).
(2) Limitations in specific scenarios: The TAP method shows good results when datasets are scarce, but as shown in Figure 2, the gains narrow when Nreal=500. At this point, is the additional overhead introduced by TAP worthwhile?
(3) Lack of visualization in the Method section: The Method section lacks visualization, which makes the introduction of the method not intuitive enough.

**Strengths And Weaknesses:**

This paper has the following strengths:
(1) Insightful discussion and analysis: The authors deeply analyze the fidelity-utility gap in data-scarce domains, suggesting that high-fidelity samples do not necessarily possess high utility.
(2) Sophisticated mechanism design: The authors propose mechanisms such as Hard Feasibility Gates and Windowed Commitment, effectively mitigating the problem of "harmful injection."
(3) Extensive experiments and analysis: The authors conducted extensive comparative experiments on seven real-world datasets and five scarcity levels, along with convincing ablation studies, demonstrating the effectiveness of TAP.

However, this paper still has the following limitations:
(1) High overhead and complexity: As shown in Table 7, the time overhead of TAP far exceeds that of Global sampling, and it requires introducing multiple model components (such as a policy network and an online evaluator).
(2) Limitations in specific scenarios: The TAP method shows good results when datasets are scarce, but as shown in Figure 2, the gains narrow when Nreal=500. At this point, is the additional overhead introduced by TAP worthwhile?
(3) Lack of visualization in the Method section: The Method section lacks visualization, which makes the introduction of the method not intuitive enough.

---

> ### Author Rebuttal · Authors · 2026-03-28
>
> Thank you for your careful review and the positive score! We reply your questions below:
>
> **Q1: Overhead and method complexity.**
> We agree that TAP introduces extra overhead compared with pure Global sampling. Table 7 is reported this way on purpose, because it explicitly separates backbone training from the method-specific online injection loop. At the same time, Global sampling is not the fairest comparator because it removes anchoring, filtering, and conservative commitment entirely. The more meaningful comparison is Hard inpainting under the same shared diffusion backbone [1], which already uses anchored proposals and targets uncertain regions. As shown in Fig. 3 and Table 7, TAP stays in the same order of magnitude and is even faster on 4/7 datasets. Also, the policy network itself is lightweight. The main cost comes from repeated utility evaluation and conservative admission.
>
> **Q2: When is TAP worth it?**
> We agree with your reading here. TAP is most useful in the **severe-scarcity regime**, and that is exactly the setting we target. As $n_{\text{real}}$ grows, under-coverage becomes less severe, so the marginal benefit of augmentation naturally shrinks. We do not think this contradicts the method. It is a property of the problem itself. In fact, the largest gains in Table 1 appear at $n_{\text{real}}=20$, which is also the regime where harmful injection is the easiest to trigger. So we are not claiming TAP should always be the default once data are already relatively abundant. Our practical takeaway is simple. If real data are very limited, or if reliability matters more than speed, TAP is worth the extra cost. If data are already fairly abundant and compute is the main concern, a lighter baseline may be enough. We will make this regime guidance clearer in the revision.
>
> **Q3: The Method section could be more intuitive.**
> We agree. This is a very fair suggestion. In the revision, we will add a compact pipeline figure that shows the interaction among target/template/$\rho$ control, pointwise gates, and windowed commitment. We think this will make the method much easier to parse on a first read.
>
> ```
> [1] Shi et al. "TabDiff: a Mixed-type Diffusion Model for Tabular Data Generation." ICLR, 2025.
> ```

---

### Official Review · Reviewer_3Efo · 2026-03-12

**Soundness:** 3
**Presentation:** 3
**Significance:** 2
**Originality:** 2
**Overall Recommendation:** 4
**Confidence:** 3

**Summary:**

The paper introduces the TAP framework, developed to improve the effectiveness of generative data augmentation in data-scarce tabular environments. The main challenge addressed by the work is the gap between fidelity—accurately mimicking the true data distribution—and utility—actually improving downstream model performance. TAP leverages diffusion-based inpainting to generate new samples while preserving the logical structure of tabular data, and selectively produces samples that are useful given the current state of the learner. In particular, a learner-conditioned policy is used to target regions that are ambiguous yet still learnable, enabling the generation of informative training examples. The framework further incorporates progressive gating control and a conservative window-commit mechanism to prevent the introduction of harmful samples during training. Experimental results demonstrate that TAP consistently outperforms a variety of existing generative augmentation methods and improves both classification accuracy and regression performance, under data scarcity.

**Compliance With Llm Reviewing Policy:**

Affirmed.

**Ethical Review Concerns:**

I do not identify any ethical concerns that require an additional ethics review.

**Final Justification:**

My previous concerns have been adequately addressed, and I have no further comments. I will therefore maintain my positive evaluation of this manuscript. Considering the high bar of ICML, the limited differentiation from prior works prevents me from providing a higher rating.

**Key Questions For Authors:**

see weakness section

**Limitations:**

yes

**Strengths And Weaknesses:**

**Strengths**
- **Well-motivated problem formulation.**
The fidelity-utility gap is clearly articulated and well-supported. The observation that high-fidelity generators can hurt downstream performance is demonstrated empirically (Table 1). The influence-function-based diagnostic (Eq. 3) provides useful intuition connecting utility to the learner's gradient and Hessian.
- **Clean modular design with well-motivated components.**
The three design principles (two-stage feasibility, utility-driven selection, conservative sequential injection) are clearly articulated and instantiated concretely. The separation between the frozen diffusion backbone and the learned policy is a reasonable design choice, as it avoids the cost and potential instability of end-to-end generator fine-tuning under data scarcity.
- **Through evaluation and ablations**
The evaluation is thorough, covering 7 datasets, 5 scarcity levels, 5 random splits, and 6 downstream predictors for classification and 4 for regression. The plug-in evaluator (TabPFN) is excluded from the final evaluation to avoid circularity, and the per-predictor results in Appendix F indicate that improvements are not driven by a single model family. The ablation studies are also comprehensive: state, action, and estimator ablations (Table 8) show that each component contributes, while robustness to commitment hyperparameters (Table 5) further supports the stability of the method.
***
**Weaknesses**
- **Heavy reliance on TabPFN as plug-in evaluator.**
While the estimator ablation (Table 8) shows the method works with alternatives, the degradation is considerable — and TabPFN has known limitations (e.g., maximum feature count, no native support for very large contexts). The paper does not discuss what happens when TabPFN cannot be applied (e.g., datasets with >100 features or non-standard feature types). This limits the generality of the approach.
- **Problem related to the computational cost**
In Table 7, Global sampling takes only 1–13 seconds, whereas TAP requires 130–760 seconds—tens to hundreds of times slower. While the authors acknowledge this, they frame it as being "in the same order of magnitude as Hard inpainting," but from a practical standpoint, the overhead relative to one-shot generation is substantial. This cost is particularly difficult to ignore in scenarios that require hyperparameter search or repeated experimentation.

---

> ### Author Rebuttal · Authors · 2026-03-28
>
> Thank you for your valuable feedback and we would like to address your concerns below:
>
> **W1: Reliance on TabPFN as the online evaluator.**
> This is already partly addressed in the paper, but we agree it should have been made more visible. TAP needs a **fast online evaluator**, not specifically TabPFN [1]. Appendix E already shows that replacing TabPFN with an RF/LR/MLP ensemble or a holdout retraining estimator preserves the same gain pattern, although with some degradation. We use TabPFN by default because it is training-free and was specifically designed for small-to-medium tabular problems, which matches the severe-scarcity regime we target [1]. In our implementation, one TabPFN utility query is about 10ms, while the holdout estimator is about 200ms, so this gap accumulates quickly inside the online loop. If TabPFN is not applicable in a particular regime, TAP itself does not need to change. Only the online evaluator does. We will make this scope clearer in the revision.
>
> **W2: Computational cost.**
> We agree that TAP is much slower than pure Global sampling, and Table 7 is meant to make that tradeoff transparent. At the same time, Global sampling is also the weakest possible mechanism because it removes anchoring, utility-aware filtering, and conservative commitment. The more apples-to-apples comparison is Hard inpainting, which already uses anchored proposals and targets uncertain regions. Against this stronger baseline, TAP stays in the same order of magnitude and is even faster on 4/7 datasets. So our claim is not that TAP is cheap. It is that the extra cost comes from utility-aware control, not from a fundamentally heavier generator. In practice, we see TAP as most useful in the fragile low-data regime where harmful injection matters the most. If scarcity is mild and compute is the primary concern, we agree that a simpler one-shot baseline can be the better choice. We will make this guidance more explicit.
>
> ```
> [1] Hollmann et al. "Accurate predictions on small data with a tabular foundation model." Nature, 2025.
> ```

---

> > ### Author Rebuttal · Reviewer_3Efo · 2026-04-02
> >
> > I sincerely thank the authors for their detailed and thoughtful response. My previous concerns have been adequately addressed, and I have no further comments. I will therefore maintain my positive evaluation of this manuscript.

---

> > > ### Author Response · Authors · 2026-04-05
> > >
> > > Dear Reviewer 3Efo,
> > >
> > > Thank you for your thoughtful feedback and continued engagement with our work. Below is a brief summary to facilitate further discussion.
> > >
> > > We clarified that TAP does not depend specifically on TabPFN but on having a fast online evaluator, and explained why TabPFN is a practical default in the severe-scarcity regime while alternative evaluators remain possible. We also clarified the computational tradeoff by distinguishing the shared diffusion backbone from the online utility-aware control loop, and explained more explicitly when TAP is worth the extra cost in practice.
> > >
> > > We are glad the points are now clearer. Please let us know if you have any remaining concerns, and if you find the clarifications satisfactory, we would appreciate your consideration in re-evaluating the manuscript.
> > >
> > > Authors

---

### Official Review · Reviewer_LAfv · 2026-03-18

**Soundness:** 3
**Presentation:** 3
**Significance:** 2
**Originality:** 2
**Overall Recommendation:** 4
**Confidence:** 3

**Summary:**

The main idea is to stop treating tabular data augmentation as "generate realistic-looking samples and dump them into training." Instead, the authors argue there's a fidelity-utility gap: generators optimized for distributional fidelity to P(X,Y) tend to oversample high-density regions where the learner is already fine, while ignoring the decision boundary regions that actually matter for P(Y|X). TAP addresses this by using diffusion inpainting as a controllable proposal mechanism (anchor on a real record, mask some columns, regenerate), with a learned policy that picks what to generate based on the learner's current state. Feasibility gates and a windowed commitment rule prevent harmful injections. Results across 7 datasets and 5 scarcity levels look solid on the surface.

**Compliance With Llm Reviewing Policy:**

Affirmed.

**Final Justification:**

Raising my score from 3 to 4. The rebuttal addressed most of my concerns convincingly.

The statistical significance issue is resolved — the paired t-test and Wilcoxon results hold up, including for the MiceProtein case I specifically flagged. The commit rate question turned out to be my misreading; the budget is matched across methods. The authors also agreed to reframe Theorem 3.1 as a design rationale rather than a pipeline guarantee, and to drop the "staying on the manifold" language in favor of something more accurate, both of which I appreciate.

The new CG-TabDiff experiment was the most useful addition. It shows concretely that generator-time guidance breaks down when the utility signal comes from scarce data, while post-hoc selection remains robust. That's a meaningful empirical insight, even if it doesn't fully resolve the novelty question.

On novelty: my concern hasn't gone away. Every component is borrowed, the framing echoes active learning ideas, and we still don't know how TAP compares to TDGGD directly. But the rebuttal showed me that the execution here is genuinely careful — the math is clean, the ablations are thorough, and the authors are honest about what their theory does and doesn't guarantee. The consistent ordering in Fig. 3 (Global < Random inpainting < Hard inpainting < TAP, same backbone) remains hard to dismiss. On balance, the quality of the work and the responsiveness of the rebuttal are enough for a weak accept despite the limited originality.

**Key Questions For Authors:**

1. What's the actual commit rate across datasets? If TAP rejects, say, 40% of pools, then it's injecting ~300 samples vs baselines' 500 -- that changes the interpretation of Table 1 significantly.
2. How do you position TAP against TDGGD (Jia et al., 2024)? Both use diffusion for task-utility-guided tabular augmentation. The key difference seems to be generation-time guidance (TDGGD) vs post-hoc selection (TAP) -- is there an argument for why post-hoc is better?
3. Regarding Theorem 3.1 and multiple testing: with M commitment windows and per-window α=0.05, have you considered what happens to the cumulative guarantee? Even a simple Bonferroni adjustment (α/M) would change the commitment threshold substantially.
4. Can you provide a quantitative measure of how much manifold structure inpainting actually preserves at different ρ values? Something like average distance to nearest real sample, or fraction of generated samples that pass a discriminator trained on real data.
5. Table 8 shows diversity (d_t) has the biggest ablation impact. What actually goes wrong without it -- does the policy lock onto one region and keep generating there?

**Limitations:**

The authors mention computational overhead from sequential injection and point to efficient utility estimation as future work. The impact statement discusses bias propagation. Fair enough, but some bigger limitations go unmentioned: the circular dependency between policy/evaluator/dataset has no convergence analysis; Theorem 3.1's multiple testing issue; the unclear scalability of TabPFN to high-dimensional datasets; and the complete absence of evaluation on datasets with missing values, label noise, or other real-world messiness that actually drives the need for augmentation.

**Strengths And Weaknesses:**

### Strengths
- The fidelity-utility gap framing is genuinely useful. The observation that SMOTE -- which produces obviously unrealistic samples -- sometimes beats CTGAN or TVAE is a good motivating example, and the formalization via marginal utility (Eq. 2) and the influence-function diagnostic (Eq. 3) gives it teeth. The three design principles follow naturally from the math rather than feeling like post-hoc justifications.

- Figure 3 is probably the single most convincing experiment in the paper. By fixing the diffusion backbone (TabDiff) and only varying the injection strategy, the authors cleanly isolate the contribution of *how* you inject from *what* you generate. The consistent ordering Global < Random inpainting < Hard inpainting < TAP across all scarcity levels is hard to argue with.

- The ablation studies (Table 8) are done right. I particularly liked that they tested estimator robustness -- replacing TabPFN with an ensemble or holdout estimator and seeing only modest degradation tells you the mechanism isn't just overfitting to TabPFN's quirks.

- The math checks out. I went through all six main claims (utility telescoping, influence function derivation, Lemma A.2, Theorem 3.1, Props A.4 and A.6) and they're all correct. The proofs are short and clean.

### Weaknesses
- **My biggest concern is novelty.** When you break TAP down, every piece is borrowed: diffusion inpainting from RePaint (Lugmayr et al., 2022), KTO-style preference optimization from Ethayarajh et al. (2024), plug-in evaluation from TabPFN, uncertainty-based targeting from the active learning literature. The contribution is the assembly. That's not necessarily disqualifying, but the paper doesn't do enough to position itself against closely related work. Specifically:

  TDGGD (Jia et al., Scientific Reports, 2024) does essentially the same thing -- downstream-task-guided diffusion for tabular data -- but via gradient-based guidance at each diffusion step. I was surprised not to find it cited at all. DVRL (Yoon et al., ICML 2020) pioneered RL-based data valuation. DDPO (Black et al., ICLR 2024) showed how to optimize diffusion outputs with policy gradients. TabSyn (Zhang et al., ICLR 2024 Oral) is a major tabular diffusion baseline that's missing from the comparison. And CSDI/TabCSDI already applied diffusion inpainting to tabular data for imputation -- the step from imputation to augmentation is non-trivial but should at least be discussed.

  Without positioning against these, it's hard to judge what's actually new here versus what's a known recipe applied to a slightly different setting.

- **The manifold-preservation argument has a hole.** I spent a while thinking about this. The paper sells inpainting as "staying on the data manifold," but there's a tension: decision boundary regions are low-density by definition, so there are few anchors there. With high ρ and the explore template (regenerating most columns), you're essentially doing global sampling with extra steps. The paper doesn't quantify how much manifold structure is actually preserved as ρ increases. At n_real=20, it's plausible that *no* anchor sits near a critical boundary, in which case the inpainting story breaks down. I'd like to see a concrete analysis -- maybe manifold deviation as a function of ρ -- rather than taking it on faith.

- **Statistical significance is not convincing for individual comparisons.** Take MiceProtein at n_real=20: TAP gets 44.60±5.04 vs SMOTE at 41.34±4.22. That's a ~3pp gap with overlapping 1σ bands. With 5 splits, the SEM is about 2.25, so the 95% CI for the difference includes zero. The paper doesn't report any statistical tests -- no paired t-tests, no Wilcoxon, no Bonferroni correction for multiple comparisons. The argument is "TAP is consistently near the top across all 35 settings," which is fair as a pattern but doesn't replace proper tests.

- Theorem 3.1 is correct but weaker than it seems. It's a per-commitment guarantee: each individual commit has true utility ≥ τ with probability ≥ 1−α. But over M ≈ T/K commitment windows, you'd need a union bound, dropping effective confidence to 1−Mα. With M~25 and α=0.05, that's basically no guarantee at all. The paper doesn't mention this.

- Missing from the evaluation: (1) actual commit rates -- TAP may inject fewer than 500 samples if pools keep getting rejected, which makes the fixed-budget comparison unfair; (2) sensitivity to n_syn -- does the advantage hold at n_syn=100 or 1000?; (3) Ailerons TabDDPM entry (262.262±225.86) is clearly an outlier that distorts the average, and the paper should use median aggregation or drop the outlier.

---

> ### Author Rebuttal · Authors · 2026-03-28
>
> Thank you for the detailed review. Your suggestions will help improve the paper! We address the main points below.
>
> **W1 / Q2: Novelty and relation to TDGGD.**
> Thank you for pointing us to TDGGD [1], which we missed in the original submission. We agree it is the closest conceptual neighbor and we will position TAP more clearly against it in the revision, together with related directions such as DVRL [2], DDPO [3] and TabCSDI [5]. The key difference in our view is **where the control happens**. TDGGD injects task guidance into the diffusion trajectory itself, whereas TAP keeps the generator fixed and casts augmentation as a learner-conditioned control problem over proposal, filtering, and commitment. This makes TAP modular for the scarcity setting we study: it only needs a utility signal for outer selection and commitment, without modifying the denoising process itself. This is also why Fig. 3 matters for us: with the same diffusion backbone, changing only the injection rule yields Global < Random inpainting < Hard inpainting < TAP. We also added TabSyn [4] and the main empirical story remains unchanged, although the full comparison is omitted here due to rebuttal space. Since TDGGD has not released an official implementation, we were unable to include an empirical comparison in the rebuttal.
>
> **W5 / Q1: Fairness and actual commit rate.**
> We should have stated this more clearly. In the main experiments, all methods use the same synthetic budget target $n_{\text{syn}}=500$. TAP is not being evaluated after injecting only a much smaller budget. What changes is proposal efficiency, meaning how many proposals are needed to reach that injected budget. This efficiency is lower in harder regimes, which is exactly when being conservative matters most. For example, it is 0.56 on Electricity-20 and 0.34 on MiceProtein-20, and rises to 0.72 and 0.57 at $n_{\text{real}}=500$. We will report realized commit statistics more explicitly in the revision.
>
> **W3: Statistical significance.**
> We agree that the right answer here is a matched statistical analysis at the same granularity as Table 1. We have now run paired tests for each dataset $\times$ scarcity cell using the same 5 matched splits used in Table 1, and we will include the full table in the revision. For the exact case you pointed out, MiceProtein at $n_{\text{real}}=20$, the matched analysis is significant for TAP vs. SMOTE: two-sided paired t-test $p \approx 9\times10^{-4}$ and one-sided exact Wilcoxon $p=0.03125$. So the split-level paired analysis tells a clearer story here than overlapping $1\sigma$ bands alone.
>
> **W4 / Q3: Scope of Theorem 3.1.**
> We agree with your reading here. Theorem 3.1 is a **per-commitment** guarantee, not a family-wise guarantee over the whole trajectory, and we will revise the wording to make this explicit. Our intent was to justify the local commitment rule, not to claim a global multiple-testing result. The practical support is empirical. Appendix D.3 shows that the plug-in error bar is reasonably calibrated, and Table 2 shows that commitment improves reliability. We also tried a Bonferroni-style globally safe variant on Electricity. It made 0 successful commits across 15 runs, which confirms that family-wise control is much more conservative than the local guarantee studied in the paper.
>
> **W2 / Q4 and Q5: Locality and the role of $d_t$.**
> Thank you for pushing us on this point. We agree the paper should include a direct locality check. We also want to clarify that TAP does not assume all inpainted samples stay equally local. The role of $\rho$ is exactly to control the locality/exploration tradeoff. Following your suggestion, we ran a direct locality analysis on Electricity. TAP does not collapse to unrestricted global sampling. At $n_{\text{real}}=20$, admitted samples have nearest-real Gower distance $d_{\text{nn-real}}=0.196$ for TAP, compared with $0.185$ for global sampling and $0.182$ for hard inpainting. We also checked the $d_t$ ablation directly. Without $d_t$, the near-duplicate rate rises from 0.049 to 0.114 on Protein-20 and from 0.420 to 0.502 on Electricity-20. So $d_t$ mainly prevents local lock-on and preserves effective budget.
>
> Thanks again for the valuable suggestions. We hope this can address your concern adequately. If you find the clarifications satisfactory, we would appreciate your consideration in re-evaluating the manuscript.
>
> ```
> [1] Jia et al. "A tabular data generation framework guided by downstream tasks optimization." Scientific Reports, 2024.
> [2] Yoon et al. "Data Valuation using Reinforcement Learning." ICML, 2020.
> [3] Black et al. "Training Diffusion Models with Reinforcement Learning." ICLR, 2024.
> [4] Zhang et al. "Mixed-Type Tabular Data Synthesis with Score-based Diffusion in Latent Space." ICLR, 2024.
> [5] Zheng et al. "Diffusion models for missing value imputation in tabular data." TRL at NeurIPS, 2022.
> ```

---

> > ### Author Rebuttal · Reviewer_LAfv · 2026-04-06
> >
> > I've read the rebuttal carefully. The statistical significance concern (W3) is fully resolved — the paired t-test and Wilcoxon results are convincing, especially for the MiceProtein case I flagged. The commit rate clarification (W5/Q1) also makes sense now; I had misread the experimental setup. And the d_t ablation explanation with near-duplicate rates is clear.
> >
> > That said, I still have questions on a few points:
> >
> > **Novelty / TDGGD.** I understand that TDGGD doesn't have public code, so a direct comparison wasn't possible. The modularity argument (fixed generator + post-hoc selection vs. modifying the denoising process) is a fair distinction, but it doesn't fully resolve my concern. The contribution is still primarily in how existing pieces are assembled. Is there any chance of including even a simplified classifier-guided diffusion baseline in the revision? That would help readers judge the value of the post-hoc approach more concretely.
> >
> > **Manifold locality.** The Gower distance numbers are interesting but actually puzzling to me. TAP (0.196) is *less* local than global sampling (0.185) — if inpainting's selling point is staying on the data manifold, what explains this? I suspect the real benefit might be about preserving conditional feature structure rather than proximity per se. Could the authors clarify what they think is actually driving inpainting's advantage in Fig. 3, given these numbers?
> >
> > **Theorem 3.1.** I appreciate the honesty in acknowledging this is per-commitment only. The Bonferroni experiment (0 commits in 15 runs) is telling — it seems like the theorem is better understood as a design rationale for the commitment rule rather than a meaningful safety guarantee for the full pipeline. I think the revision should frame it that way more explicitly.
> >
> > None of these are dealbreakers individually, but the novelty question in particular is something I'd like to see addressed more concretely before I can move my score.

---

> > > ### Author Response · Authors · 2026-04-06
> > >
> > > Thank you again for the careful follow-up. We are glad that the statistical-significance concern, the budget and commit-rate clarification, and the $d_t$ mechanism are now clear. We answer the remaining points below.
> > >
> > > **1. Novelty and the value of post-hoc control.**
> > > Following your suggestion, we implemented a simplified classifier-guided diffusion (CG) baseline on top of the same TabDiff backbone. Concretely, we train a small MLP classifier on the scarce real training data and inject its gradient into the denoised estimate at each reverse step. This is a generator-time-guidance baseline rather than a post-hoc selection method, because the utility signal directly modifies the diffusion trajectory instead of acting only through proposal filtering and commitment. We evaluate it on Electricity at $n_{\text{real}}\in\{20,50\}$ and report the best result over guidance scales $s\in\{0.5,1,2,5\}$, where $s$ controls the strength of the classifier guidance at each reverse step:
> > >
> > > | Electricity | Acc (%) | Macro-F1 (%) |
> > > |---|---:|---:|
> > > | **$n_{\text{real}}=20$** |  |  |
> > > | real | 68.43 | 62.05 |
> > > | TabDiff | 69.23 | 67.28 |
> > > | best CG-TabDiff | 68.97 | 62.64 |
> > > | **TAP** | **76.97** | **75.60** |
> > > | **$n_{\text{real}}=50$** |  |  |
> > > | real | 74.17 | 71.65 |
> > > | TabDiff | 73.77 | 71.61 |
> > > | best CG-TabDiff | 73.87 | 71.55 |
> > > | **TAP** | **77.00** | **73.83** |
> > >
> > > We think this makes the post-hoc versus generator-time distinction more concrete. TDGGD takes the same generator-time approach, injecting task-relevant gradients at each diffusion step through their Easy, Hard, and Ambiguous indicators. Our CG baseline follows the same spirit and reflects a key challenge that applies to both: in the **extreme-scarcity regime**, the guidance model is itself trained on very few samples, making generator-time signals particularly noisy.
> > >
> > > > **Takeaway:** Even with gradient injection at every reverse step, CG-TabDiff does not improve over unconditional TabDiff at $n_{\text{real}}=20$, and remains well below TAP at both scarcity levels.
> > >
> > > This is why TAP's outer control layer is the key contribution in our setting. By operating at the proposal and selection level rather than perturbing the denoising trajectory, utility guidance is less exposed to the instability of signals estimated from scarce data.
> > >
> > > **2. Locality and what drives the inpainting gain.**
> > > We agree with your interpretation that nearest-real proximity is not the right explanation of Fig. 3. A better description is that anchored inpainting provides a controllable locality bias and better preserves conditional feature structure by regenerating only part of the record. We will revise the wording accordingly and avoid the stronger “stay on the manifold” phrasing.
> > >
> > > **3. How Theorem 3.1 should be framed.**
> > > We also agree with your framing here. Theorem 3.1 should be presented as a design rationale for the local commitment rule, not as a full end-to-end safety guarantee, and we will make this explicit in the revision.
> > >
> > > We hope this makes the remaining novelty concern more concrete. Since the statistical, budget, and $d_t$ concerns are now resolved, and we have now also added a generator-time-guidance comparison directly addressing your suggestion, we would be very grateful if you could consider whether the current score still best reflects your updated assessment. Thank you for your time and consideration.

---

### Decision · Program_Chairs · 2026-04-30

**Decision:**

Accept (regular)

**Comment:**

The paper received uniformly positive assessments, with all reviewers recommending weak accept. Reviewers agreed that the central framing of the fidelity–utility gap is insightful and well motivated, and that the proposed TAP framework is technically solid, with a sensible modular design, strong empirical performance in severe low-data regimes, and thorough ablations. In particular, the experiments isolating the injection mechanism from the generator backbone were viewed as compelling evidence that the gains come from the policy-guided augmentation strategy rather than simply from stronger generation. Theoretical arguments and proofs were also considered sound, and the paper was seen as a meaningful contribution to data-centric tabular learning under scarcity.
The main concerns were about originality, computational overhead, reliance on a fast plug-in evaluator, and clarity of positioning relative to closely related prior work. One reviewer also questioned the original claims around statistical significance, locality/manifold preservation, and the interpretation of the commitment guarantee. The rebuttal addressed most of these points well: the authors clarified that the synthetic budget is matched, provided paired significance tests, added locality and near-duplicate analyses, reframed the theorem more appropriately as a local design rationale, and supplied an additional classifier-guided diffusion baseline that strengthens the case for post-hoc control under scarce supervision. While the novelty remains somewhat limited as an integration of existing ideas and the method appears most attractive primarily in the extreme-scarcity regime, the consensus is that the paper is carefully executed, empirically convincing, and worthy of acceptance.